



# The AirGAM 2022r1 air quality trend and prediction model

Sam-Erik Walker[1], Sverre Solberg[1], Philipp Schneider[1], Cristina Guerreiro[1]

[1]Norwegian Institute for Air Research (NILU), Kjeller, Norway

*Correspondence to*: Sam-Erik Walker (sew@nilu.no)

**Abstract.** This paper presents the AirGAM 2022r1 model – an air quality trend and prediction model developed at the Norwegian Institute for Air Research (NILU) in cooperation with the European Environment Agency (EEA) over 2017-2021. AirGAM is based on nonlinear regression GAM – Generalized Additive Models – capable of estimating trends in daily measured pollutant concentrations at air quality monitoring stations, discounting for the effects of trends and time variations in corresponding meteorological data. The model has been developed primarily for the compounds $NO_2$, $O_3$, $PM_{10}$ and $PM_{2.5}$.

Meteorological input data consist of temperature, wind speed and direction, planetary boundary layer height, relative and absolute humidity, cloud cover and precipitation over the period considered. The exact set of meteorological variables used in the model depends on the compound selected for analysis. In addition to meteorological variables introduced in the model as covariates, i.e. explanatory variables for the concentration levels, the model also incorporates time variables such as day of the week, day of the year, and overall time, related to the model's trend term. The trend analysis is performed at each station

separately. Thus, the model only considers the temporal features of concentrations and meteorology at a station, not any spatial correlations or dependencies between stations. AirGAM is implemented using the R language for statistical computing and, in particular, the GAM package `mgcv`. In the model, meteorological and time covariates are represented and estimated as smooth nonlinear functions of the corresponding variables. Thus, the trend term is defined and estimated as a smooth nonlinear function of time over the period selected for analysis. Once fitted to training data, the model may be used as a prediction tool capable

of predicting air pollutant concentrations for new sets of meteorological and time data which are not in the training set – e.g. for cross-validation or forecasting purposes. The model does not explicitly use emissions or background concentrations – these are sought to be implicitly represented through the estimated nonlinear relations between meteorology, time and concentrations. In addition to meteorology-adjusted trends, the program also produces unadjusted trends – i.e., trends based on the same regression set-up but only including the time covariates. Both types of trends can be output in the same run,

making it possible to compare them. Ideally, the meteorology-adjusted trend will show the trend in concentration mainly due to changes in emissions or physio-chemical processes not induced by changes in meteorology. AirGAM has been developed and tested primarily in trend studies based on measurement data hosted by EEA, including the Airbase data (before 2013) and the Air Quality e-Reporting (AQER) data from 2013 and onwards. Still, the model is general and could be applied in other regions with other input data. The EEA data provide daily or hourly surface measurements at individual monitoring stations

in Europe. For input meteorological data, we extract time-series from the gridded meteorological re-analysis (ERA5) provided





by the European Centre for Medium-Range Weather Forecast (ECMWF) for each monitoring station. The paper presents results with the model for all Airbase/AQER stations in Europe from the latest EEA trend study for 2005-2019.

## 1 Introduction

The atmospheric level of pollutants at a given site and time is determined by the emissions, meteorology, and various physio-
chemical conditions (vegetational uptake, solar radiation, etc.). The evaluation of emission abatement protocols relies on long-term trends in measured air pollutant concentrations. These analyses are complicated by the influence of year-to-year variations in meteorology. Although the measurements are based on daily or hourly data throughout the year, seasonal anomalies in the weather conditions could significantly alter the annual statistics used in the trend calculations, such as, e.g. the 2003 and 2018 heatwaves affecting the surface ozone levels in Europe (Logan et al., 2012; Sicard et al., 2013, Simpson et al., 2014, Diaz et
al., 2020; Johansson et al., 2020). State-of-the-art CTMs (Chemical Transport Models) simulating the physio-chemical processes in the atmosphere is the common tool to meet this challenge. However, applying CTMs for very long periods in a multi-scenario approach could be costly and time-consuming.

Furthermore, the analyses of trends become non-trivial when there are significant discrepancies between the CTM calculations
and the measured level of pollutants. One background for our development of the AirGAM model – which is based on statistical regression generalized additive modelling or GAM (Hastie and Tibshirani, 1990; Wood, 2017) – was a statement in the 2013 Air Quality Report (EEA, 2013): "… there is a discrepancy between the past reductions in emissions of $O_3$ precursor gases in Europe and the change in observed average $O_3$ concentrations in Europe". This raised the question of whether the discrepancy was due to errors in the emission data, lack of performance by the CTMs or simply a result of the uncertainty of the data.

A large number of scientific papers have shown that statistical-based models focussed on the normalisation or removal of the impact of meteorological anomalies are valuable tools that could complement the CTMs when designed carefully (e.g. Thompson et al., 2001; Ordóñez et al., 2005; Camalier et al., 2007; Zheng et al., 2007; Chan and Vet, 2010; Davis et al., 2011; Grange et al., 2018; Fix et al., 2018; Otero et al., 2018; Grange and Carslaw, 2019; Pernak et al., 2019). A variety of names
and types of these statistical models have been used for the assessment of long-term atmospheric data, like random forest (RF) models (e.g. Grange et al., 2018; Grange and Carslaw, 2019, Pernak et al., 2019), boosted regression trees (Carslaw, 2021), gradient boosting techniques (Barré et al., 2021; Keller et al., 2021, Petetin et al., 2020),  as well as generalized additive models (Ordóñez et al., 2020) as used in this work. Note that standard trend estimation techniques, such as e.g. curve fitting, smoothing methods (moving average), or robust methods such as, e.g. Theil-Sen estimation etc., can be used to estimate trends in time-
series of concentrations but cannot account for trends in or impact of the corresponding meteorology. For this, regression-based methods are needed. An excellent recent overview of scientific issues and statistical methods for trend analysis in atmospheric time series is given in Cheng et al. (2021).





The initial development of the AirGAM model (Solberg et al., 2018b) was based on a statistical method that was used by the
US-EPA (Environmental Protection Agency) routinely for assessing surface ozone trends, adjusting for the inter-annual
influence of changing meteorology (Camalier et al., 2007). Subsequently, the model has been gradually refined and extended
for $NO_2$, $PM_{10}$ and $PM_{2.5}$ (Solberg et al., 2018a; Solberg et al. 2019; Solberg et al., 2021a).

## 1.1 The AirGAM model

AirGAM is a model for estimating trends in daily measured pollutant concentrations at one or more monitoring stations over
a given period by adjusting for trends and time variations in corresponding meteorological data. It is based on nonlinear
regression GAM modelling and has been developed primarily for the compounds $NO_2$, $O_3$, $PM_{10}$ and $PM_{2.5}$. Meteorological
data consist of temperature, wind speed and direction, planetary boundary layer height, relative and absolute humidity, cloud
cover and precipitation. The exact set of meteorological variables used in the model depends on the compound selected for
analysis. In addition to meteorological variables introduced as covariates, i.e. explanatory variables for the concentrations, the
model also uses time variables as covariates such as day of the week, day of the year (seasonality), and total time (days) over
the period; the latter of which is associated with the model's trend term. The trend analysis is performed at each station
separately. Thus, the model only considers the temporal features of concentrations and meteorology at a station, not spatial
correlations or dependencies between stations.

The model is implemented using the R language for statistical computing (R Core Team, 2021) and, in particular, the GAM
(Generalized Additive Modelling) statistical modelling package `mgcv` (Wood, 2017). The program also uses the air pollution
data analysis package `openair` (Carslaw and Ropkins, 2012; Carslaw, 2019) for analysis and plotting purposes and the
`sandwich` package (Zeileis, 2004) for some statistical calculations. Using the GAM regression approach, the relationships
between concentrations and meteorological and time covariates are represented and estimated as smooth nonlinear functions
of the variables. Thus, the trend term is defined and estimated as a smooth nonlinear function of time (days) over the period
selected for analysis.

In GAM modelling, the eventual nonlinear relations between the response (concentration) and covariates need not be known
in advance. Still, they will, in a sense, be uncovered as part of the estimation procedure. Further, regularisation by penalising
variability (wiggliness) of each nonlinear relation helps identify a more generalizable model and avoid overfitting. This
represents one of the essential advantages of using a GAM model. Other standard regression model approaches such as multiple
linear regression (MLRs) with linear or polynomial terms or generalized linear models (GLMs) incorporating only linear
relationships between the meteorology and time covariates and the concentrations cannot model these dependencies with
sufficient flexibility and accuracy since they usually are of a more complex and unspecified nonlinear form.





Once fitted to training data, the model may be used as a prediction tool capable of predicting air pollutant concentrations for new sets of meteorological and time data which are not in the training set – e.g. for cross-validation or forecasting purposes. The model's predictive capability is evaluated with associated plots using several deterministic and probabilistic model evaluation metrics. A leave-one-year-out cross-validation procedure is incorporated in AirGAM and is usually performed
automatically as part of the model run.

The model has been mainly developed for trend studies based on the air quality (AQ) measurement data hosted by the European Environmental Agency (EEA), including the Airbase data (before 2013) and the AQER data (Air Quality e-Reporting) from 2013 and onwards. The EEA data provide daily or hourly surface concentrations at individual monitoring stations. For the
input meteorological data, we extract time-series from the gridded meteorological re-analysis data (ERA5) provided by ECMWF for each monitoring station (Hersbach et al., 2018; Hersbach et al., 2020). Figure 1 shows a schematic of the data flow of AirGAM.

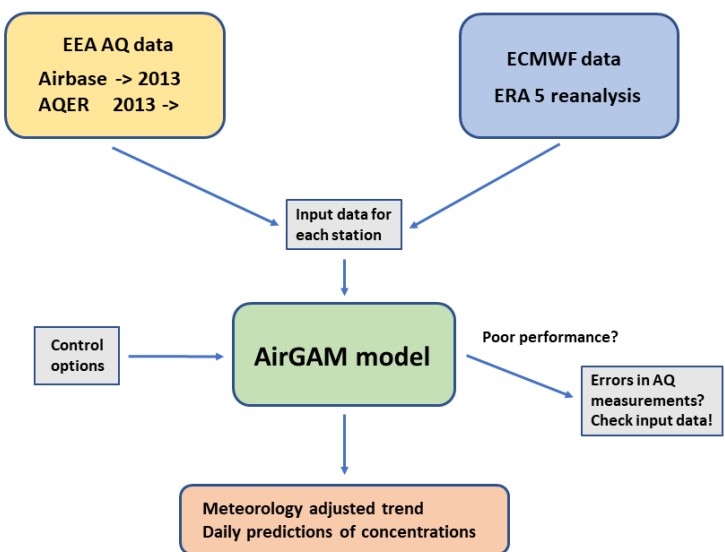

**Figure 1.** AirGAM data flow scheme.

In addition to concentrations and meteorology, the program reads several control options for the model run. Another feature of AirGAM is that it may sometimes check for errors in the air quality data. Poor performance of the model, e.g. low correlations between observed and model-predicted concentrations from cross-validation, we have often found associated with dubious measurement data.






AirGAM does not explicitly use emissions or background concentrations – these are sought to be implicitly represented through the estimated nonlinear relations between the concentrations and the meteorology and time variables. In addition to meteorology-adjusted trends, the program may also produce unadjusted trends – i.e., trends based on the same regression set-up but only including the time covariates. Both types of trends can be output in the same run, making it possible to compare them.

The model estimates trends over a user-defined period, from a minimum of two years and upwards. For each year, the user may select the whole year; or a sub-part of the year, e.g. only winter months (say October-March), summer months (say April-September), or any user-defined interval of months for the trend analysis. Usually, only a single set of smooth relations between concentrations and the covariates is estimated from the data in the model. However, it is possible to operate with different groups of estimated smooth relations for different parts of the year (or sub-year) if needed, e.g. one set for the winter, say October-March, and another for the summer, say April-September. This latter capability of the model is typically necessary for modelling $O_3$ and $PM_{2.5}$ using data for the whole year since the relationships usually are different in the wintertime than in the summer.

## 1.2 Predictions in the Covid-year 2020

The conceptual idea behind a statistical model as AirGAM is that the model is "trained" to find patterns between various input data (local temperature, wind speed, mixing height etc.) and the daily level of pollutants ($NO_2$, $O_3$ etc.) for a given training period. Based on these patterns, the model can predict pollutant levels outside the training period. The main advantage compared to CTMs is that no assumptions on emissions are needed. Thus, the exceptional conditions experienced during the Covid-19 lockdown in 2020 offered a perfect task for such statistical models. We applied the AirGAM model for EEA's AQER (Air Quality e-Reporting) data of $NO_2$ (EEA, 2020; Solberg et al., 2021b) during the first lockdown in Europe (March-July 2020) and trained the model on data for the previous five years (2015-2019). The difference between the AirGAM predictions ("business-as-usual") and the observed $NO_2$ levels could then be related to the impact of the lockdown on mobility (road transport, aviation etc.). Compared to gridded models as CTMs, statistical models could be applied directly for urban stations. We found that the AirGAM model performed fine for most sites while performing more poorly for a minor number of stations, partly explained by inconsistent measurement data. After aggregating all traffic sites (urban and suburban) for individual countries, the results showed good compliance between predicted and observed daily $NO_2$ levels.

The predictive capabilities of the AirGAM model come in addition to the application for long-term trend assessments. The experience from applying AirGAM specifically for a Covid-19 analysis (Solberg et al., 2021b) was that the model performed very well for $NO_2$ at the urban and suburban background and traffic sites. In contrast, the performance was lower at rural locations as expected since the $NO_2$ levels outside the urban areas are less determined by local meteorological conditions. The performance was lower for $O_3$ and PM, which is also expected since secondary formation and long-range transport are more



important processes for these compounds. Such processes are only indirectly captured by AirGAM. These results do not imply
that the AirGAM model and other statistical tools are unfit for O₃ and PM assessments, just that the model performance is
somewhat lower than for a primary pollutant such as NO₂.

### 1.3 Outline of the paper

First, in Sect. 2, we overview the statistical GAM methodology implemented in the AirGAM model and the details from its
numerical implementation. Sections 3-5 describe the input data files to the model, how to run the model on Windows and
Linux, and the result files. Section 6 gives results from a recent EEA trend study for 2005-2019 based on using all available
Airbase/AQER stations in Europe. Section 7 compares GAM with the RF method in the R package `rmweather`. In Sect. 8,
we shortly discuss how AirGAM can also be used as a tool for data quality investigations. Finally, Sect. 9 contains a summary
and conclusions. Appendix A describes how to download and install R, the necessary R packages, and the AirGAM model for
Windows and Linux, respectively. Appendix B gives an overview of the model's warning and error codes and messages.
Appendix C contains a list of some possible questions and answers (Q & A) regarding the model.

### 2 Model formulation and implementation

#### 2.1 GAM models

In statistics, a GAM model (Hastie and Tibshirani, 1990; Wood, 2017) is a nonlinear regression model linking expected values
$\mu_i$ of a given response variable $Y_i$ to one or more explanatory variables $x_{ij}$ through the following relations:


$$g(\mu_i) = \beta_0 + \sum_{j=1}^{p} \beta_j(x_{ij}); \quad \mu_i = E(Y_i), \tag{1}$$

where $\beta_0$ is a constant (the intercept), and where $\beta_j(\cdot)$, for $j = 1, ..., p$, represents smooth functions of the covariates $x_{ij}$,
with $p$ the number of covariates. In our implementation of this model for air quality analysis, the response variable $Y_i$ in Eq.
(1) represents a daily average (NO₂ or PM) or maximum 8-hour running mean (O₃) concentration at day number $i$ at a given
site, while $x_{ij}$ represent the values of the explanatory variables, for $j = 1, ..., p$, at the same location and day. These consist of
various meteorological variables such as temperature, wind, etc., and time variables such as day of the week, day of the year,
etc. The meteorological covariates depend on the air pollutant being modelled, as shown in Table 1.

In Eq. (1), $g(\cdot)$ is a function (the link function) that links the statistically expected value of the response variable $Y_i$, i.e. $\mu_i$,
to the covariates $x_{ij}$. Also, $Y_i$ is assumed to have a definite probability distribution, the response distribution, with mean $\mu_i$





and variance $V_i$. Further, in Eq. (1), each $\beta_j$ is a smooth function of $x_{ij}$, and not simply a constant to be multiplied with $x_{ij}$ as in multiple linear regression (MLR) or generalized linear regression models (GLMs). Thus, GAM models represent an extension of these models for regression. GAMs are also more flexible than MLR models (but not GLMs) since the mean

value $\mu_i$ is related to the covariates through a given link function $g(\mu_i)$, which need not be the identity function $g(\mu) = \mu$.

Since the relation between air pollution and meteorology is generally nonlinear, MLR models or GLMs cannot naturally model this relationship. Only a nonlinear model, such as a GAM, capable of fitting nonlinear relations between an air pollutant and a set of meteorological covariates, will have a chance of succeeding in this regard. It is then vital to choose a "right" set of

meteorological covariates for each type of air pollutant to be modelled. Besides, various time covariates will also be needed.

Note that emissions, background concentrations, physio-chemical processes and depositions have deliberately been left out from the design of such a regression model, even though we know that air pollutants are closely linked to and primarily determined by these factors and processes in addition to meteorology. The idea is to see how far we can use meteorology and

time data to model air pollutants. Limitations depend on the compound and type of data. Over the last four years, the current model has been developed for $NO_2$, $O_3$, $PM_{10}$ and $PM_{2.5}$. This development has resulted in a set of meteorological and time covariates found to model and predict concentrations of these compounds well (Table 1).

For $NO_2$, $PM_{10}$ and $PM_{2.5}$, we apply a log link $g(\mu) = \log \mu$ and Gamma distributions as response distributions. This is

because these compounds generally have a somewhat more extensive range of concentration variations than $O_3$, with the variance of $Y_i$, i.e. $V_i$, typically proportional to $\mu_i^2$. For such variables, it is usual practice in GAM modelling to select a logarithmic link function and a distribution potentially skewed to the right, such as a Gamma, as a response distribution for $Y_i$ (Wood, 2017). This was also applied in the previous trend studies (Solberg et al., 2018a; 2018b; 2019).

For $O_3$, we apply an identity link $g(\mu) = \mu$ and Normal distribution as a response distribution. This choice is because $O_3$ has a relatively small range of concentration variations where the variance of $Y_i$, i.e. $V_i$, does not change very much with the mean $\mu_i$. Thus the response distribution is well represented with a symmetric distribution such as a Normal.

The selection of input variables has been made by combining a priori knowledge of the main physio-chemical processes and

experience during the model development. Extensive research in previous work with the model (Solberg et al., 2018a; 2018b; 2019) resulted in meteorological and time variables used, as presented in Table 1. Absolute humidity is introduced as a variable for $O_3$ since the gas-phase reaction O'D + $H_2O$ → 2OH is the main production path for OH in the atmosphere and since OH,



in turn, is decisive for the O₃-formation. For PM and NO₂, we used relative humidity to reflect the importance of wet deposition and cloudiness.


**Table 1.** List of meteorological and time variables used in the AirGAM model (Eq. (1)) for various compounds. The short names refer to those used in the text and graphics files in Sect. 5.

|  | Meteorological variable | Short name | Unit | Used by compound |
|---|---|---|---|---|
| $x_1$ | Daily mean temperature at 2 m | `temp` | °C | All except O₃ |
|  | Daily temperature at 2 m at 18 UT | `temp` | °C | O₃ |
| $x_2$ | Daily mean wind speed at 10 m | `ws` | m s⁻¹ | All |
| $x_3$ | Daily mean wind direction at 10 m | `wd` | ° | All |
| $x_4$ | Daily mean planetary boundary layer height | `pblh` | m | All |
| $x_5$ | Daily mean relative humidity | `rh` | % | All except O₃ |
|  | Daily absolute humidity at 18 UT | `h2o` | g kg⁻¹ | O₃ |
| $x_6$ | Medium height cloud cover | `mcc` | % | All |
| $x_7$ | Daily total precipitation | `prec` | mm day⁻¹ | PM₁₀ and PM₂.₅ |
| $x_8$ | Weekday number | `dayofweek` | day | All |
| $x_9$ | Day number in the year or sub-part of the year | `dayofyear` | day | All |
| $x_{10}$ | Continuous-time in fraction of years (0.0 at the start of the period). This is the trend term. | `years` | year | All |

In the model, the trend term is represented as a smooth function of time ($x_{10} = t$ ) rather than as a straight line as in some previous

studies (Solberg et al., 2018a, 2018b). The main reason for this choice is for the model to be better prepared for trend studies over extended periods. It is less relevant to represent the whole trend over the entire period as a straight line in such cases.

Since meteorological variables are included in this GAM model to explain the expected ($\mu_i$) and observed ($Y_i$) concentrations of air pollutants, at each time point $t_i$, the estimated trend $\beta_{10}(t)$ in Eq. (1) will represent a so-called meteorology-adjusted

trend, i.e. a trend discounting for the effects of trends or time variations in these meteorological variables over the period selected for the analysis. This represents the main output from AirGAM.

In addition to meteorology-adjusted trends as produced by the model described above, AirGAM may also estimate so-called unadjusted trends. These are trends produced by the same GAM regression model set-up as above, but only including the time

covariates $x_8$-$x_{10}$, i.e., removing all the meteorological covariates $x_1$-$x_7$.  Both trends can be produced individually and output





together from the same run, making it possible to compare them. These two models will be called the meteorology-adjusted and unadjusted models in the following.

Note that in AirGAM, we only use GAM models with covariates in a purely additive form, as shown by Eq. (1). Thus, no interactions between the variables are used, such as e.g. multiplications between variables, defining 2-dimensional smooth functions, etc. This makes the models easy to interpret since the estimated nonlinear functions encode each independent variable's contribution to the predicted concentration. Sect. 7 compares our GAM models to RF models that incorporate interactions between the variables. We show that our GAM approach produce models with predictive performance on par with this method. Thus, we argue that a purely additive model seems sufficient to build models with a good predictive performance, at least for the data analysed in this paper.

AirGAM handles each station individually; therefore, it does not consider any nearby station data when estimating trends at a given station. Thus, only the temporal features of the data are evaluated at each station and not any spatial features such as e.g. spatial correlations or trends. If there are many stations with data, and the period selected for the analysis is extensive, e.g. more than ten years with data, the amount of CPU time needed may be extensive. The program has been constructed to be run in parallel in a semi-automatic fashion in these situations, concurrently handling blocks of stations (see Sects. 3.1.5 and 4.1.1). Thus, it might be better to run the model in parallel on computers with multiple CPUs in these cases.

### 2.1.1 Calculation of physically interpretable trend curves

AirGAM outputs trend curves as plots and data values to various result files described in Sects. 5.1.1-5.1.2 and 5.1.9-5.1.10. To interpret a change in the trend level between two arbitrary time-points in these plots and data files as a change in the expected concentration levels between the same two time points under certain well-defined conditions, it is essential to adjust the raw trend given by the estimated trend function $\beta_{10}(t)$ from Eq. (1) into a physically interpretable trend curve $y_{\text{trend}}(t)$.

This section describes how this is done for the various compounds and the meteorology-adjusted and unadjusted models.

First, we focus on the meteorology-adjusted model. For compounds such as $O_3$, where we apply an identity link $g(\mu) = \mu$ in Eq. (1), the expected concentration at the time $t$ is given by

$$\mu(t) = \beta_0 + \sum_{j=1}^{p-1} \beta_j\left(x_j(t)\right) + \beta_p(t) = A(t) + B(t) ,\qquad(2)$$

with $A(t) = \beta_0 + \sum_{j=1}^{p-1} \beta_j\left(x_j(t)\right)$ and $B(t) = \beta_p(t) = \beta_{\text{trend}}(t)$. Here $A(t)$ is the contribution to the expected concentration





at the time $t$ from meteorology and the time variables day of week and day of the year, while $B(t)$ is the contribution to the expected concentration at the time $t$ from the trend term. In this case, the physically interpretable trend curve is sought defined as the function


$$y_{\text{trend}}(t) = A + B(t), \tag{3}$$

with $A$ determined so that a difference in trend values between two arbitrary time points, say $t_1$ and $t_2$, can be interpreted as the difference in the expected concentrations between these two time points, i.e., we need to have


$$y_{trend}(t_2) - y_{trend}(t_1) = A + B(t_2) - (A + B(t_1)) = B(t_2) - B(t_1) =$$
$$\mu(t_2) - \mu(t_1) = A(t_2) + B(t_2) - A(t_1) - B(t_1), \tag{4}$$

which holds for all values of $A$ as long as $A(t_1) = A(t_2)$, i.e. as long as the impact of meteorology and the two time variables is the same for the two time points. Under this assumption, Eq. (3) can be used as a physically interpretable trend curve for

any value of $A$. However, it is natural to set $A$ so that the trend curve values have the same time average over the period as the actual observations, i.e. $\overline{y}_{\text{trend}} = \overline{Y}$. Since the smooth $\beta_{\text{trend}}$-function from GAM always averages to zero over the time points of the trend estimation, this corresponds to setting $A = \overline{Y}$. Changes in the trend curve values can then be interpreted as changes in the expected concentrations if we assume the same impact of meteorology and day of week and day of the year for the two time points, i.e. $A(t_1) = A(t_2)$. Note that the actual values $A(t_1)$ and $A(t_2)$ will usually differ.


The same development as above also holds in the case of the unadjusted GAM model. There are only two time variables, day of week and day of the year, in the $A(t)$ function in this case. Thus, Eq. (3), with $A = \overline{Y}$, can be used as a physically interpretable trend curve, where changes in the trend curve values can be interpreted as changes in the expected concentrations, if we assume that $A(t_1) = A(t_2)$, i.e. as long as the impact of, in this case, only the day of week and day of the year, is the

same for the two time points.

Next, we apply the log-link function $g(\mu) = \log \mu$ to the more complicated GAM models for NO$_2$ and PM. Again, we focus first on the meteorology-adjusted model. The expected concentration level at time $t$ is now given by

$$\mu(t) = \exp\left\{\beta_0 + \sum_{j=1}^{p-1} \beta_j(x_j(t)) + \beta_p(t)\right\} = A(t)B(t), \tag{5}$$



with $A(t) = \exp\left\{\beta_0 + \sum_{j=1}^{p-1} \beta_j\left(x_j(t)\right)\right\}$ and $B(t) = \exp\left\{\beta_p(t)\right\} = \exp\left\{\beta_{trend}(t)\right\}$. Here again, $A(t)$ is the contribution to the

expected concentration at the time $t$ from meteorology and the time variables day of week and day of the year, while $B(t)$ is

the contribution to the expected concentration at the time $t$ from the trend term, both as factors in this case. Now a physically

interpretable trend curve is sought defined as a function which in this case, reads

$$y_{\text{trend}}(t) = A \cdot B(t), \tag{6}$$

with $A$ determined so that again a difference in trend values between two arbitrary time points, say $t_1$ and $t_2$, can be

interpreted as the difference in expected concentrations between these two time points, i.e., we need to have

$$y_{trend}(t_2) - y_{trend}(t_1) = A \cdot B(t_2) - A \cdot B(t_1) = A \cdot \left(B(t_2) - B(t_1)\right) =$$
$$\mu(t_2) - \mu(t_1) = A(t_2) \cdot B(t_2) - A(t_1) \cdot B(t_1), \tag{7}$$

which holds as long as $A = A(t_1) = A(t_2)$, i.e., as long as the impact of meteorology and the two time variables is the same

for these two time points and $A$ is equal to this value.

Again, it is natural to set $A$ so that the trend curve values have the same time average over the period as the actual observations,

i.e. $\bar{y}_{\text{trend}} = \bar{Y}$. Thus, $A = \bar{Y}/\bar{B}$ where $\bar{B}$ is the average of the $B(t)$ values over the period of trend estimation (note that this

latter average is a positive number due to the exponentiation in Eq. (5)). Changes in the trend curve values can then be

interpreted as changes in the expected concentrations if we assume this average impact of meteorology and day of week and

day of the year for the two time points, i.e. $A(t_1) = A(t_2) = A$. Note again that the actual values $A(t_1)$ and $A(t_2)$ will usually

differ and also differ from $A$.

To better understand the impact factor $A$ in the previous paragraph, we give a formula for the expected value of this quantity.

Since the conditioned observed concentrations $Y_i \mid \mu_i$ have expected values $\mu_i = \mu(t_i)$, from Eq. (1), we have

$$EA = \frac{E\bar{Y}}{\bar{B}} = \frac{\bar{\mu}}{\bar{B}} = \frac{\dfrac{1}{N}\sum_{i=1}^{N} A(t_i) B(t_i)}{\dfrac{1}{N}\sum_{i=1}^{N} B(t_i)} = \sum_{i=1}^{N} w_i A(t_i), \tag{8}$$





with $w_i = B(t_i) \big/ \sum_{i=1}^{N} B(t_i)$. Thus, $EA$ represents a weighted average of the $A(t_i)$ factors with weights $w_i$ for $i = 1,...,N$,

where $N$ is the number of days for the trend estimation. If $\mathrm{var}(Y_i \mid \mu_i)$ is uniformly bounded above, i.e., $\mathrm{var}(Y_i \mid \mu_i) \leq V$ for

some $V > 0$ for all $i$, and $\mathrm{cor}(Y_{i+k} \mid \mu_{i+k}, Y_i \mid \mu_i)$ exponentially approaching zero as $k \to \infty$, which holds almost invariably

for air quality observations, we have from Serfling's strong law of large numbers applied to $Y_i - \mu_i$ (McFadden, 2000, Ch. 4,

p. 92) that $A - EA \to 0$ almost surely as $N \to \infty$. Thus, for large $N$, $A \approx EA$, and the average factor $A$ in the physically

interpretable trend curve will be close to a weighted average of the factors from the meteorological and time variables over the

period for trend estimation given by Eq. (8), with weights based on the meteorology-adjusted $B(t)$ values.

The same development as above also holds in the case of the unadjusted GAM model. There are only the two time variables,

day of week and day of the year, in the $A(t)$ factor function in this case. Again, for large $N$, $A$ will be close to a weighted

average of these factor values as in Eq. (8), but with weights based on the unadjusted $B(t)$ values.

## 2.2 Numerical implementation

The GAM model (Eq. (1)) is fitted using the R package `mgcv` (Wood, 2017). The model fitting is done independently for each

station for the selected period of trend estimation. All data are used first to estimate each station's meteorology-adjusted trend.

Then, cross-validation is performed by leaving out one year of data, training the GAM model on the remaining data, and

predicting the left-out year's concentrations. This is then repeated for each year in the trend estimation period or a specific

period selected for cross-validation. In the following sections, we describe the details of the implementation.

### 2.2.1 Solution methods

The GAM model fitting is done by applying the function `bam` in the `mgcv` library. If fitting using `bam` fails for some reason,

the `gam` routine in `mgcv` is used instead. These routines fit GAM models to data, but `bam` is generally much faster than `gam`,

essential for many stations with long data periods. The `bam` routine is, therefore, always tried first. It is run with the numerical

solution method fREML (`method=fREML`), short for fast restricted maximum likelihood. This is the default solution method

in `bam`. We use the default setting `discrete=FALSE` for all compounds in this routine, i.e. we apply no discretisation of

the covariates. We do not use parallel processing in the call to the `bam` routine, only applying the default setting

`cluster=NULL` and `nthreads=1` in this routine. Thus, each `bam` call only occupies the workload of a single CPU core.

This makes it easier to utilize a multi-core computer when running several R sessions concurrently, as described in Sects. 4.1.1

and 4.2.2. For `gam,` we apply the REML (restricted maximum likelihood) solution method (`method=REML`). The GAM

modelling community now favours these numerical solution methods rather than the older GCV (generalized cross-validation)

approach to model fitting GAMs, mainly due to improved numerical stability of estimating the penalty parameters.





The `bam` and `gam` routines in `mgcv` give robust and fast solutions to the regression equations – and consistent estimates of the

smooth nonlinear relations between the concentrations and meteorology and time covariates. However, these routines do not consider autocorrelations in the model residuals, leading to somewhat underestimated confidence regions for the smooth nonlinear functions, including the nonlinear trend term. To amend this, AirGAM contains an option (Sect. 3.3.14) to include an autoregressive time-series model of order 1 (AR(1)) for the model residuals to handle the autocorrelations. The model is then solved using the `gamm` routine in the `mgcv` package. However, this routine is much slower and slightly less robust than

the `bam`/`gam` routines. Therefore, when running for many stations, we prefer to ignore the autocorrelation, at least initially, and apply the `bam`/`gam` routines instead. Additional runs with the `gamm` routine may be performed for stations to obtain a more proper confidence region for the trend, e.g. to check whether it differs significantly from a zero function.

### 2.2.2 Automatic model selection

The `bam` and `gam` routines generally do an excellent job estimating the covariates' smooth functions by penalizing variations

in these functions. However, they can only penalize a covariate function to become a straight line, not zero. Thus, the standard penalization cannot delete unnecessary covariates from the model. However, using `select=TRUE` in the call to these routines, a further penalty to straight lines in the GAM is introduced. This may lead to straight lines becoming zero functions, in effect deleting covariates from the GAM. The advantage is that more parsimonious models with no unnecessary covariates can be found. Thus, it represents a form of automatic model selection being built-in as part of the solution method. It has been

found (Marra and Wood, 2011) that this approach in most cases leads to better model selections than more traditional approaches to this in regression, which is typically based on adding or removing individual covariates in a step-wise fashion. Thus, we apply `select=TRUE` in the calls to both `bam` and `gam`. Such automatic model selection is also of great value in our system since we usually have many stations and periods to model. Thus, applying the more traditional model selection approaches of step-wise adding and removing covariates would be cumbersome even if this were done automatically.

### 2.2.3 Number of basis functions

Important user input in GAM modelling defines each smooth covariate's number of basis functions. These numbers represent the maximum model complexity allowed in the model for each smooth covariate. The precise value of each number is not vital. Still, they should be large enough to accommodate the GAM to correctly identify the smooth relationship between the response and each covariate. A setting somewhat higher than needed is usually not a problem since the penalization in the

GAM will usually take care of that and appropriately reduce the degree of variability (wiggliness) of the function curve. It will, however, increase the computing time. For the meteorological covariates and the time covariate `dayofyear`, we have found it sufficient to operate with ten basis functions (`k=10`) which is the default setting for smooth covariates in the calls to the `bam` and `gam` routines. For the `dayofweek` variable, seven basis functions are used, which is the maximum since this





variable can take only seven different discrete values (1, 2, …, 7) corresponding to Monday-Sunday. However, our system
still handles this variable as a continuous "time of the week" variable.

Defining the number of basis functions `k_years` for the trend variable `years` is more delicate. Ideally, it should be defined
high enough so that the `gam.check` routine in `mgcv` returns with the empirical number of degrees of freedom for this term
somewhat lower, say 0.5-1 lower, than the theoretical number of degrees of freedom `k_years` minus 1. This means trying
several values and choosing, say, the smallest fulfilling the above criterion. The large number of stations typically used in our
studies makes searching for the best value in each case somewhat intractable. Instead, we have chosen to introduce a simple
empirical rule for this, introducing a basis function every three years, which seems to work well for our studies since it captures
the main features of and more long-term variations in the trend quite well in most cases. Thus, our current formula used in
AirGAM as a *default* for the number of basis functions for the trend term is


$$k_{\text{years}} = \max\left(2, \left[n_{\text{years}}/3\right]\right),\qquad\qquad(9)$$

where $n_{\text{years}}$ is the total number of years for the trend analysis, and $[\cdot]$ means rounding to the nearest integer. The maximum
operator in Eq. (9) ensures that there are always at least two basis functions in the trend term irrespective of years. The formula
is only used if the user defines `k_years` as missing, i.e. the R-value NA, upon input. It is conservative, typically leading to
fewer basis functions than the ideal number. If the user is interested in more details and short-term variations in the trend,
`k_years` can be set to a higher value upon input as described in Sect. 3.3.11.

### 2.2.4 Choice of basis function

As recommended by Wood (2017), we have speeded up the AirGAM model by applying cubic regression splines (`bs=cr`) as
basis functions rather than the default thin-plate splines (`bs=tp`) in both `bam` and `gam`. This is done for all covariates, except
for the wind direction, which contains circular data, where cyclic cubic regression splines (`bs=cc`) are used as basis functions.
This ensures equal covariate function values for angles close to 0 and 360 °.

### 2.2.5 Standard deviation of regression trend coefficients

The `vcovHAC` routine in the R package `sandwich` (Zeileis, 2004) is used to calculate the standard deviation of the linear
regression trend coefficient `beta.linreg` as output to the `<station>_gam.coef_<ya>-<yb>.csv` files as described
in Sect. 5.1.5. It is also used to calculate the corresponding p-value `p.linreg` in this file.





### 2.3 Smooth function and trend uncertainty

Smooth covariate functions are estimated by the AirGAM model using data for the whole period of trend estimation. In particular, there is such a smooth curve representing the estimated trend. These smooth curves come with an estimated
uncertainty representing a 95 % confidence region for each curve. These regions are depicted as the grey-shaded areas around each of the curves described in Sects. 5.1.1-5.1.3. Note that these 95 % regions do not necessarily correspond to 95 % confidence intervals pointwise, i.e., for each point of the curve or covariate value, but rather as an average across the curve, over all covariate values (Nychka, 1988; Marra and Wood, 2012).

### 2.4 Model prediction uncertainty

The AirGAM model performs predictions of daily concentrations at stations based on the actual meteorological and time covariates for each day for each left-out year in the cross-validation calculations. These predictions come with an associated statistical uncertainty which it is essential to estimate correctly and communicate to the user. It is crucial to consider this uncertainty when comparing the model predictions with observations to accurately interpret how well the model performs to predict concentrations not used as part of the training. Our implementation focuses on estimating a 95 % probability prediction
interval (credibility interval) associated with each point prediction. The prediction intervals are displayed as grey-shaded areas around the prediction curves in the time-series plots of observed and model calculated values described in Sect. 5.1.6.

The 95 % prediction intervals are defined as intervals of 95 % probability for the unconditional (compound) distribution of predicted concentrations. These cannot be expressed analytically, so a Monte Carlo approach is needed. Two different
procedures are used: One for $O_3$, where the conditional response distribution is Gaussian, and another for $NO_2$ and PM, where it is a Gamma. We will now shortly describe these.

For $O_3$, the procedure is as follows. At day number $i$, $N$ samples of expected values $\hat{\mu}_{ij}$, $j = 1,...,N$, are drawn from a normal distribution with mean $\hat{\mu}_i$ and standard deviation $\hat{\sigma}_i$, where these two last values are obtained in the same way as in
the previous procedure. Next, the scale ($s$) parameter of the Normal conditional response distribution given the expected value $\hat{\mu}_{ij}$ is defined as $\hat{s} = \hat{s}_i$, with $\hat{s}_i$ the estimated scale or dispersion parameter obtained as the square root of the `sig2` output value from the `bam` and `gam` routines (Wood, 2017). Then, $N$ samples of predicted concentrations $\hat{y}_{ij}$ are obtained by a random draw from each of the $N$ Normal distributions, i.e., $\hat{y}_{ij} \sim N\left(\hat{\mu}_{ij}, \hat{s}_i\right)$, $j = 1,...,N$, representing samples from the unconditional (compound) response distribution. Finally, a 95 % prediction interval is again obtained for day number $i$ as the
interval between these predicted concentrations between the 0.025 and 0.975 sample quantiles.





For NO$_2$ and PM, the procedure is as follows. At day number $i$, $N$ samples of log-expected values $\log \hat{\mu}_{ij}$, $j = 1,...,N$, are drawn from a normal distribution with mean $\hat{\mu}_i$ and standard deviation $\hat{\sigma}_i$. These two last values correspond to the estimate of the expected value and standard error, respectively, of the linear predictor (Eq. (1)) at day number $i$. These values are

obtained from the `fit` and `se.fit` output values from the `predict.gam` routine in `mgcv`. Next, shape ($a$) and scale ($s$) parameters of the Gamma conditional response distribution given the expected value $\hat{\mu}_{ij}$ is defined as $\hat{a} = \hat{\phi}^{-1}$ and $\hat{s} = \hat{s}_{ij} = \hat{\mu}_{ij}/\hat{a}$, with $\hat{\phi}$ the estimated scale or dispersion parameter obtained as the `sig2` output value from the `bam` and `gam` routines (Wood, 2017). Then, $N$ samples of predicted concentrations $\hat{y}_{ij}$ are obtained by a random draw from each of the $N$ Gamma distributions, i.e., $\hat{y}_{ij} \sim \text{Gamma}\left(\hat{a}, \hat{s}_{ij}\right)$, $j = 1,...,N$, representing samples from the unconditional (compound)

response distribution. Finally, a 95 % prediction interval is obtained for the day number $i$ as the interval between the 0.025 and 0.975 sample quantiles of these predicted concentrations.

After testing with several values of $N$, 100 samples were found to give satisfactory results in defining the 95 % prediction intervals for all compounds, with a good trade-off between the accuracy of the final intervals and computational efforts. This

has thus been implemented in the model.

**3 Description of input to the model**

Input data to the AirGAM model consists of control variables defined in two files: (1) The main run script file; and (2) The model options file. Both of these files are placed in the `airgam` main directory. Further input to the model is read from files in the directory `<inp_dir>` which is the value of the input directory variable `inp_dir` as defined in the model options file.

The name of the options file is defined in the main run script but is usually called `airgam_options.txt`. An example of this file is given in the model's software distribution.

We also provide examples of main run script files for Windows and Linux in the software distribution. For Windows, the script file is called `airgam_run.bat` and is a Windows batch file. For Linux, it is called `airgam_run.sh,` a Bash shell script

file. It is also possible to run the model on a Linux cluster under a Slurm (Simple Linux Utility for Resource Management) job scheduler and workload manager (https://slurm.schedmd.com). For this, we provide a template Slurm batch file `airgam_run.sl`. The AirGAM model options file (`airgam_options.txt`) is the same for Windows and Linux.





Sects. 3.1 and 3.2 below describes the run script files for Windows and Linux, respectively. A description of the options file

is given in Sect. 3.3. Section 3.4 describes the input files as read from the `<inp_dir>` directory, containing station description

and data.

### 3.1 The main run script file for Windows

The Windows batch file `airgam_run.bat` contains control variables needed to run the AirGAM model for Windows. These

control variables are given in the form of a sequence of statements on the form `set <variable>=<value>`, where

`<variable>` is the variable to be set and `<value>` its value. With an initial setting of the variables, this script is included

in the model's software distribution. The variables to be set are described in the following subsections.

#### 3.1.1 The R script executable path

The first variable you need to set in this file is the `R_script` variable, which contains the path to the R script executable.

The initial setting of this variable in the software distribution as of this writing is `"C:\Program Files\R\R-`

`4.1.2\bin\i386\Rscript.exe"`. You need to set this variable to point to where the R script executable is installed on

your system. If the R script executable is already in your path when you log in to Windows, you may set

`R_script="Rscript.exe"`.

#### 3.1.2 Model version

The following variable in the file is `version`, which describes the AirGAM model version to run as given in the model R

script name. As of this writing, this variable is set to `2022r1,` and the model R script name is `airgam_2022r1.R`. The

variable `model` is automatically assigned to this latter value in the script.

#### 3.1.3 The work directory

Next, you set the working directory, i.e., the main directory under which all the input and output (results) directories and files

will be stored. The variable is `wrkdir,` and its initial value is `"C:\Users\xxx\Documents\My`

`documents\airgam"`. You need to change this to whatever is appropriate for you.

#### 3.1.4 The model options file

The following variable is `optfn`, the file name for the model options. Its initial value is `"airgam_options.txt"`.

#### 3.1.5 Number of blocks for parallel processing

The last variable in the batch file is `nb` which is short for the number of blocks of stations to split the calculations over when

performing parallel processing with the model R script. This variable's initial value is one, meaning we want to run one instance





of the program handling all stations. Suppose you run the model on a computer with multiple CPUs. In that case, you may wish to perform parallel processing utilizing several CPU cores to perform the GAM model calculations faster. E.g. if your computer has a CPU with four cores, you may wish to set nb=4. Four copies of the model R script will then be set up to run in parallel, handling roughly a quarter of the stations, each. Hyper-threading on Windows with Intel processors can also

effectively run two separate model R scripts on each core. Thus, you may wish to set nb between 5 and 8 to use this on a computer with four CPU cores. Section 4.1.1 contains a more detailed description of parallel processing on Windows.

## 3.2 The main run script file for Linux

The Bash shell script file airgam_run.sh contains all control variables needed to run the AirGAM model for Linux. The variables are defined in the shell script as a sequence of statements of the form <variable>=<value>, where

<variable> is the variable to be set and <value> its value. This script is included in the model's software distribution with an initial setting of the variables. The variables that can be set are the same as those in the Windows batch script file, and we thus refer the reader to Sect. 3.1 for a description of these.

### 3.2.1 Linux cluster

The AirGAM model can also be run parallel on a Linux cluster with multiple nodes and CPUs. The file airgam_run.sl is

a script similar to the airgam_run.sh file but contains Slurm job scheduling and workload managing directives to utilize parallelisation on such a system. See Sect. 4.2.1 for a description of this file and how to perform parallel processing with it.

## 3.3 The model options file

The model options file, usually called airgam_options.txt, contains the major set of control variables needed to run the AirGAM model. As stated above, this file is common to Windows and Linux. The file contains statements of the form

<variable>=<value>, where <variable> is the variable to be set and <value> is its value. This file is included in the model's software distribution with an initial setting of the variables.

Note that no single or double quotes should be used around the text strings on the left or right of the equal (=) sign in this file. However, you can have as many blank characters as you wish before and after the variable's name and before and after its

value. All values are read as text strings and converted to numerical or logical values as necessary by the program. Further, the file may contain any number of blank or empty lines or comment lines, the latter of which must start with the # character. Each <variable>=<value> line may also include a comment at the end, after a # character. The sequence of variables does not matter; you can freely permute this as you wish. The variables to be set are described in the following subsections. If a variable is commented out or removed from the options file, it will be given a default value as described in each subsection

below.





### 3.3.1 Input and output directories

The first two variables you need to set in this file are the program's directories for input and output (results). The defaults are `inp_dir=airgam_input` and `out_dir=airgam_results`. These input and output directories for the model can be defined using full paths or relative to your defined work directory, as described in Sect. 3.1.3. The program will create the output directory if it does not already exist.

### 3.3.2 The compound to run for and its unit

The following two variables are `comp` and `unit`, respectively, the compound the model will run for and its unit, i.e. the unit used for the concentrations. The default values are `no2` and `ugm-3`, respectively. For `comp`, you may use the values `no2`, `o3`, `pm10` and `pm2.5`. For `unit`, there are no specific legal values. It is only used as a text string in output plots to indicate the unit; thus, any value is permitted. However, it is interpreted and formatted by the routines in the openair package for some plots. Therefore, you may wish to stick to the conventions used by this package. E.g. use `ug/m3` or `ugm-3` to indicate concentrations in µgm$^{-3}$. Other possible strings are `ppm`, `umol/mol`, `ppb`, `nmol/mol`, etc.

### 3.3.3 The start and end year of the trend calculations

Then you need to define the start and end year of the trend calculations. This is done using the variables `year_a` and `year_b`, respectively. E.g. setting `year_a=2005` and `year_b=2019` defines the trend calculation period as 2005-2019. There are no defaults for these variables.

### 3.3.4 The start and end year of the cross-validations

Next, you need to define the start and end year of the cross-validation part of the calculations. This is done using the variables `year_c` and `year_d`. You may set these to the same values as `year_a` and `year_b`, respectively, default, or opt for a shorter cross-validation period. E.g., setting `year_c=2017` and `year_d=2018` means you will only perform cross-validation for the shorter period 2017-2018. If `year_c=year_d`, the cross-validation will be performed for a single year. If `year_c > year_d`, the cross-validation part of the calculations will be skipped entirely.

### 3.3.5 Sub-part of the year

Next, you need to set the sub-part of the year you will use for the trend calculations. This is done using the `subyear` variable as follows `subyear=mma-mmb`, where `mma` and `mmb` are three-letter abbreviations for the start- and end-month of the sub-part of the year. Valid values for `mma` and `mmb` are: `jan`, `feb`, `mar`, `apr`, `may`, `jun`, `jul`, `aug`, `sep`, `oct`, `nov` and `dec`. E.g., setting `subyear=nov-feb` means running the model only for November-February each year. You may also use the





values `winter`, `summer`, and `year`, which is short for `oct-mar`, `apr-sep` and `jan-dec`, respectively. The default is `jan-dec`. Usually, you will want to run the model for whole years, i.e. using `subyear=year` or `jan-dec`.

### 3.3.6 Seasonal conditioning

The control variable `use_season_cond` is a 0/1 logical variable of whether or not you want to use the season indicator strings as optionally given in the `season` column in the station data files. If `use_season_cond=0`, the default, then the season indicator strings are not used. This means that all dates with data throughout each year or sub-part of the year are considered to belong to a single "season", which indicates to the model that we only need to estimate a single set of smooth functions based on all the data. If `use_season_cond=1` but the `season` column is not present in the station data files; or exists but only contains a single type of value, e.g. `all` (the value in itself does not matter), a single set of smooth functions will again be estimated based on all data. However, suppose `use_season_cond=1` and the station data files contain season columns with at least two different values, e.g. `winter` and `summer`. In that case, conditional seasonal modelling will be turned on, and the model will estimate a set of smooth functions based on the data belonging to each unique value of the season string. E.g. in the above case, one set of smooth functions will be estimated for the winter period, e.g. from October-March, based on data given in the station data files with season=winter; and one for the summer period, e.g. from April-September, based on the data with `season=summer`. The user can freely choose the number of unique string values and their actual values. In AirGAM, the season string variable will be converted to a factor variable in R and used in a so-called "by"-construct for the smooth functions in the call to the GAM routines. However, this "by"-construct is only used for the meteorological variables and not for the time variables such as `dayofweek`, `dayofyear` and the trend. Thus, a single smooth function will always be output for the time variables. If you need to estimate different trends for each season, you will need to use the `subyear` control variable described in Sect. 3.3.5 and run AirGAM separately for each sub-part of the year thus defined.

Such individual modelling and estimation of the smooth functions of the meteorological variables depending on the season are often essential for specific compounds. The relationship between the concentration level and the meteorological variables will often differ for seasons, e.g. winter or summertime. This is true in particular for $O_3$ and $PM_{2.5}$. Note that it is also possible to use a four-season type of modelling with AirGAM by defining the season variable to have four different values, e.g. `winter`, `spring`, `summer` and `fall`, corresponding to, e.g. December-February, March-May, June-August and September-November. Likewise, it is possible to separate on an even finer basis, e.g. monthly, if need be. However, the run time will quickly increase with such finer partitioning. In practice, we have found that separation between winter and summer is often sufficient, at least for the compounds mentioned above.





### 3.3.7 Filename with static station data

Next comes the file's name with static station data, the variable being `statfn`. The default is `stations.csv`. It must be either a comma-separated value (CSV) file with file extension `.csv`; or a text file with one or more blank characters separating

the data, in which case the file extension must be `.txt`, e.g. `stations.txt`. The file lists all stations to be used in the calculations, and one such file needs to exist for each year with data. A description of these files' content and placement under the work directory is given in Sect. 3.4.

### 3.3.8 Data coverage percentages

Then comes two percentages for data coverage: The variables are `perc1` and `perc2`, respectively. The variable `perc1`

describes the percentage coverage needed for the data in each year or sub-part of the year to use this year in the trend calculations. E.g., setting `perc1=75` means that at least 75 % of the data needs to exist (not be missing) in any given year for that year to be included in the trend calculations. The variable `perc2` describes the percentage coverage of years fulfilling the previous criterion of non-missing data in a specific year or sub-part of a year to perform a trend calculation for a given station. E.g., `perc2=100` means that 100 % of the years needs to fulfil the data coverage criterion for individual years (controlled by

`perc1`) to perform the trend calculation. The default is 75 for both variables. Note that it is only possible to give these percentages as integers.

### 3.3.9 Meteorology-adjusted and unadjusted trend modelling

The following variables indicate whether you want to perform a meteorology-adjusted trend modelling, unadjusted trend modelling, or both. The variables to be set are `incl_metadj` and `incl_unadj`, respectively. You can specify either of

these to 1 if you want to include the corresponding type of trend. If both are set to 1, both kinds of trends will be estimated and output. If only one is set to 1, the indicated type will be output. If both are set to 0, no modelling will be performed, and neither of the trends will be produced, but the program will run through, reading in all station data. Thus, you may wish to use this as an initial quick test of your data setup. The default for these variables is 1.

If you select to model only unadjusted trends, no meteorological data are needed in the station data files, only observed concentrations. This makes it possible to quickly run the model at stations with only air quality observations and no meteorology.

### 3.3.10 Trend type

The following variable is the `trend_type`. It defines the kind of trend to use in the model. This variable can be set to

nonlinear, linear, or zero values. The default is `nonlinear,` which means the trend is modelled as a nonlinear smooth function. In this case, a cubic regression spline (bs="cr") is used for the trend term with penalty parameters determined





automatically by the GAM solution routines, i.e. `bam`, `gam` and `gamm`, using `method="REML"` in the calls to these routines. You may, however, use `linear` if you want to model the trend as a straight line and `zero` if you're going to run the model without a trend. When choosing `linear`, a thin plate regression spline (`bs="tp"`) is used instead for the trend term with

both penalty parameters set to a high value; currently, $10^2$ is used. This leads to a straight line for the trend (approximately). When choosing `zero`, a shrinkable version of the same thin plate regression spline (`bs="ts"`) is used for the trend term with its single penalty parameter set to a very high value; currently, $10^5$ is used to obtain a zero trend (again approximately). These settings are used in the calls to the GAM solution routines `bam` and `gam` in `mgcv`.

### 3.3.11 Number of basis functions for the trend term

The following variable is `k_years` which is the number of basis functions defined for the trend term, i.e. for the time variable `years` of the model. The default setting of this variable when the `trend_type` is `nonlinear` is NA, i.e. missing value, which means that the value will be calculated based on the number of years for the trend estimation using Eq. (9). This results in two basis functions (i.e. a straight line) for up to eight years, where the number of basis functions switches to three. For 10, 15, 20, 25 and 30 years of trend estimation, it corresponds to three, five, seven, eight, and ten basis functions to be used,

respectively. Introducing a basis function for the trend term every three years is often an appropriate setting if the focus is to investigate the main features and more long-term variations in the trend. If the user is interested in more details and short-term variations, it should be set to a higher value. If `trend_type` is `linear` or `zero,` only two basis functions are used for the trend term irrespective of the `k_years` value set.

### 3.3.12 Calling bam

Next is the variable `incl_bam`. This is set to 1 as default which means that a call to the `bam` routine in `mgcv` is always tried first. If the call to `bam` fails, the `gam` routine in `mgcv` will be called. The `bam` routine is usually much faster than `gam`. However, if you want to bypass `bam` and only call `gam,` you may set `incl_bam=0`.

### 3.3.13 Automatic model selection

The following variable is `incl_select`. This is set to 1 as default which means that automatic model selection will be

turned on in the calls to the `bam` and `gam` routines in `mgcv` via the `select=TRUE` setting in the calls to these routines. Usually, you will want to use this as part of the GAM modelling. However, if you want to exclude such automatic model selection, you may set `incl_select=0`, which sets `select=FALSE` in the calls to `bam` and `gam`.





### 3.3.14 Include a time-series AR(1) model for the residuals

Next comes the variable incl_ar1. This is set to 0 as default. If incl_ar1=1, an AR(1) model, i.e. a time-series
autoregressive model with a single one-day time lag, will be used for the residuals. In this case, the gamm routine in mgcv
will be used instead of bam and gam.

### 3.3.15 Robust predictions

Next is a variable rob_pred, which can turn on two robust predictions from the fitted GAM model. This variable's default
value is limcov. In this case, covariate values outside the interval of values encountered in the training data will be set to the
nearest covariate value in these data before being used in a prediction. In this way, we ensure that only covariate values within
the training data boundaries will be used for prediction. A second possibility is rob_pred=outmiss. In this case, if a
covariate value is outside the interval of values encountered during training, an additional analysis is performed to check
whether the corresponding predicted concentration is a potential outlier compared with the concentration values of the training
data as judged by a generalized box plot method (Bruffaerts et al., 2014). If so, the prediction will be set to a missing value
NA. If rob_pred=none, the robust prediction will not be performed.

### 3.3.16 GAM seed

The variable gam_seed defines the seed value used in AirGAM before calling the routines gam.check and k.check from
mgcv. It is also used before producing the 100 random samples from the unconditional (compound) response distribution
when creating a 95% prediction interval for each day in the leave-one-year-out cross-validation part. This ensures exact
reproducibility regarding output from the program. You can set this value to any positive whole number; the default is 1234.

### 3.3.17 Legend position on plots

The variable leg_pos can define the legends' vertical position in the program's time series output plots. E.g. you may use
top or bottom to place the legends at the top or bottom of the plots. Note, however, that the legends will always be placed
on the right of each plot. You may also use leg_pos= (an empty string) to put it in the right middle position. The default
value of this variable is top.

### 3.3.18 Autocorrelation results

If incl_acf=1, an analysis of the autocorrelation of the residuals is performed. This analysis checks to see to what degree
the residuals are dependent or not. Ideally, in a fitted GAM model, the residuals should be independent, i.e. all autocorrelation
values should be zero or close to zero. The default value of this variable is 0.





### 3.3.19 Concurvity analysis

If `incl_ccuv=1,` a concurvity analysis will be performed. This type of analysis checks to what degree the covariates are independent. Thus, concurvity is to GAM modelling as multicollinearity is multiple linear regression. However, concurvity also consider to what degree the covariates are nonlinearly independent. The default value of this variable is 0.

### 3.3.20 Conditional quantile plots

If `incl_cond_quant=1,` a conditional quantile plot of observations versus GAM predicted values will be produced. The routine `conditionalQuantile` in the `openair` package in R produces this plot. The default value of this variable is 0.

### 3.3.21 Taylor diagram plots

If `incl_taylor=1,` a Taylor diagram plot of observations versus GAM predicted values will be produced. The routine `TaylorDiagram` in `openair` produces this plot. The default value of this variable is 0.

### 3.3.22 Probabilistic evaluation results

This is controlled by the last five control variables in the model options file: `incl_pit_hist`, `incl_pit_ecdf`, `incl_marg_ecdf`, `incl_sharp` and `incl_crps`. By setting these variables to 1, one obtains PIT (Probability Integral Transform) histograms, PIT empirical CDFs, marginal CDFs, sharpness diagrams, and CRPS (Continuous Ranked Probability Score) plots and results, respectively, based on the observed and GAM model predicted values for each year of the cross-validation period. The default values of these variables are 0.

### 3.4 The input data directory and station data files

When the model is started from the run script, it reads its input from the data directory `<inp_dir>`, where `<inp_dir>` is the input directory as given in the options file. This input directory can either be provided with a full path or relative to the working directory of the run script.

The model output result files will be written to the directory `<out_dir>`, where `<out_dir>` is the output (results) directory as given in the options file. This directory can also be provided either using a full path or relative to the working directory of the run script. The result files in this directory are described in Sect. 5.

The input data directory must be organised with one or more sub-directories `<ccc>/<yyyy>` where `<ccc>` denotes the compound, e.g. `<ccc>=no2` and `<yyyy>` denotes the year with data, e.g. `<yyyy>=2005`. The `<ccc>` string must be the same as the compound string `comp` as given in the options file, e.g. `no2`, `o3`, `pm10` or `pm2.5`. There must be one such sub-


directory `<ccc>/<yyyy>` for each year `<yyyy>` in the period defined for the trend calculation in the options file (`year_a`-`year_b`).


In each sub-directory `<ccc>/<yyyy>`, there needs to be a single file with a list of all stations active for that year. This file's name is defined in the options file by the variable `statfn`, e.g. by default `statfn=stations.csv`. Each such station file is an ASCII file with one header line with field names and one or more subsequent lines with the following station data:

- Station EoI code (`name`), e.g. `EE0018Ah`[1]
- Station longitude (`lon`) in degrees (°)
- Station latitude (`lat`) in degrees (°)
- Station height above mean sea level (`z`) in m
- Station type (`type`), e.g. traffic, industrial or background
- Station area (`area`), e.g. urban, suburban or rural
- Country where the station is located (`country`)

The file must be either a comma-separated value (CSV) file with file extension `.csv`; or a text file where one or more blank characters separate the data, in which case the file extension must be `.txt`, e.g. `stations.txt`. In either case, the header field names must be exactly as given in the parenthesis above, without double quotes around the terms. The model actively
uses `name`, `type`, `area` and `country` values for file naming and plotting purposes. However, the other data may be used to create sub-sets of stations for specific purposes, e.g. if one only wishes to run for stations between certain latitudes or longitudes, below certain mean sea levels, or stations of a particular type, in certain types of areas or situated in a given country, etc. However, this must be done manually by the user. There are currently no filters built into the program to select sub-sets of stations automatically. Usually, stations are pre-screened for altitude, and only stations below a certain height above sea
level are used in the AirGAM model, e.g. only stations below 1000 m. This is based on the view that the model and the meteorological data are less appropriate for mountain stations.

We have used EEA's EoI codes for naming the stations. However, any station code or name could be used as long as the names in the station list file agree with the names of the individual data files (see below). Note that the EoI codes were the central
entity in Airbase until 2012, while the station local-id was introduced in AQER. To link the time-series across the Airbase/AQER databases, we used the Sampling Point Identifier (provided in both databases), a unique code referring to the combination of pollutant and monitoring stations. We added the letter 'h' or 'd' to the EoI code to distinguish hourly based data from daily ones when both types of measurements of the same compounds have been carried out at a station.

---

[1] Note that we added the letter 'h' or 'd' to the EoI code to distinguish between hourly and daily based data





When the model starts, it reads the station list files for each year and builds up a global list of stations internally. In this build-up, the program tolerates missing years. A station listed for a given year is added to the global list if it is not on the global list already. It is also checked if there is insufficient data coverage based on the number of years remaining until the last year compared with the `perc2` data coverage percentage defined in the options file. If so, the station is excluded. Otherwise, the station is accepted and added to the global list. When the global list is finally built, the model will consider each station in this

list, one at a time. Whether or not trend calculations and cross-validation analysis will be performed for a station will depend on the actual station data read and whether or not these meet the data coverage criteria defined by the `perc1` and `perc2` coverage percentages described in Sect. 3.3.8.

When the model performs calculations for a given station, it reads the station data. For each year `<yyyy>` the station data are

read from a separate file in the `<ccc>/<yyyy>` sub-directory. The name is `<station>_<ccc>_<yyyy>.csv` when it is a comma-separated values (CSV) file. It is also possible to use blank-separated values files, in which case the file names are the same but with the extension `.txt` instead. The program automatically detects which type of file name is present in each sub-directory. In either case, `<station>` must be the station EoI code name string, e.g. `EE0018Ah`, and `<ccc>` and `<yyyy>` must be the compound name and year, respectively.


Each station data file is an ASCII file with one header line with field names and one or more subsequent lines with daily station data of air quality, meteorology and optionally a season indicator string. Each line of data in this file consists of the following values:

- A date string (`date`) on the form `yyyy-mm-dd` (year-month-day)
- Observed concentration (`<ccc>`) of the given compound in $\mu g m^{-3}$
- Air temperature (`temp`) in °C
- Wind speed (`ws`) in $m s^{-1}$
- Wind direction (`wd`) in degrees (0-360 °)
- Planetary boundary layer height (`pblh`) in m
- Relative humidity (`rh`) in % (for all compounds other than $O_3$)
- Absolute humidity (`h2o`) in $g\ kg^{-1}$ dry air (for $O_3$)
- Medium height cloud cover (`mcc`) in %
- Precipitation (`prec`) in mm $day^{-1}$ (only for $PM_{10}$ and $PM_{2.5}$)
- Optional season indicator string


Note again that the header field names in these files must be as given in the parenthesis above. However, upper case letters in these names are allowed but converted to lower case internally in AirGAM. Here the header `<ccc>` means to use, e.g. the header-name `no2` for the observed concentrations if we run for $NO_2$. Again, there should be no double quotes in the header names. Missing data are denoted in these files with the two-letter value NA, standard for R missing data. The program tolerates



missing data values in the station data files in the sense that the model uses only whole rows with non-missing data. The program also accepts full missing years with data, i.e. the station data file need not exist for all years of the trend calculation.

Note again that if you opt for running only an unadjusted trend model with AirGAM (see Sect. 3.3.9), no meteorological data are needed in the station data file, only dates and air quality observations, and optionally, the season indicator strings.


Whether or not a trend calculation will be performed for a given station depends on this station's available data and the coverage percentages as read from the options file. As an absolute minimum, the program needs to have at least two years with data to run, i.e. at least a period with data in two different years. E.g. if the period 2005-2019 is chosen and a 75 % coverage of years is specified in the options file, one needs to have at least twelve years with data for a given compound available for a station

to perform trend calculation.

## 4 Running the model

### 4.1 Running the batch script for Windows

The simplest way to run the AirGAM model on Windows is to double-click on the batch script file `airgam_run.bat`, placed in the `airgam` directory. This will run the model based on the control variables set in this batch file and the model

options file, usually named `airgam_options.txt`. These are described in Sects. 3.1 and 3.3, respectively.

Alternatively, the user may start a command prompt window in Windows and navigate to the same `airgam` directory. The model may then be started by issuing the command `airgam_run.bat` in the command prompt window. However, note that for this to work, the `%%i` construction in the batch file must be replaced by `%i`. However, this second approach's advantage is

that the command prompt window will not disappear in the case of errors, and it is possible to see any eventual error messages from the batch run. In any case, the user should consult the model's log file after a model run (see below).

In either case, the model reads its input data from the `<inp_dir>` directory and writes its output in text and plot files to the `<out_dir>` directory. The content of these directories is described in Sects. 3 and 5, respectively. Status messages and any

warning or error messages are written to the program log-file `AirGAM_log.txt` in the sub-directory `main` of the `<out_dir>` directory. After a model run, this file should be inspected to check for status and any warnings or errors. A description of this file is given in Sect. 5.1.12. The warning and error messages are described in more detail in Appendix B.





### 4.1.1 Parallel processing

If you have a large number of stations, you may wish to split the number of stations into nb > 1 blocks of stations and to run

each block in parallel utilizing multiple CPUs or CPU cores concurrently. You need to set the control variable nb in the batch script file `airgam_run.bat` to your desired number of blocks. E.g. if you run on a Windows computer with four CPU cores, you may wish to set nb=4, or if you want to utilize hyper-threading running up to two processes per core, you may set nb to some number between five and eight, e.g. nb=7, so that you half a core available to other tasks.

When you start the batch script file `airgam_run.bat`, nb copies of the AirGAM R script will be started, each in a separate

run window by the last command in this batch file. Each copy of the R script will process its block of stations indicated by the variable ib in the call to the R script. This variable ranges from 1 to nb. Each R script copy will create the same global list of stations but only process the part indicated by the block number ib. E.g. if an R script copy receives the argument variable ib=3, only stations in the third block of the global list will be processed by this R script. In this way, nb copies of the R script will be run parallel on a Windows computer handling separate blocks of stations.

There will be no conflicts writing to the result files since station names are used as unique identifiers in these files. The only exception is result files containing all station results, i.e. the `AirGAM_*.csv` files and the `AirGAM_log.txt` file. To avoid conflicts when writing to these files, the results will instead be written to files with names `AirGAM<ib>_*.csv` and `AirGAM<ib>_log.txt`, thus a unique set of files for each block ib of stations when performing parallel processing. After all model runs are finished, all the separate `AirGAM`-files must be concatenated to one common set of `AirGAM_*.csv` and

`AirGAM_log.txt` files. This can be accommodated by using the script `airgam_cat.bat` in the `airgam` directory.

### 4.2 Running the shell script for Linux

Running the model on Linux is very similar to running it on Windows (see Sect. 4.1). The simplest way is to use the Linux Bash script file `airgam_run.sh` in the `airgam` directory. This will start and run the AirGAM model based on the variables defined in this file. These are described in Sect. 3.2.


As for Windows, the model reads input data from the directory `<inp_dir>` and writes output to the directory `<out_dir>`. The input and output (results) files are described in Sects. 3 and 5, respectively. Similarly, status and eventual warning or error messages are written to the program log-file `AirGAM_log.txt` in the sub-directory `main` of the `<out_dir>` directory. This file should be inspected to check for status and any warnings or errors from the model's run. The description of this file

is given in Sect. 5.1.12. The warning and error codes and messages are described in more detail in Appendix B.





### 4.2.1 Linux cluster

The model can also be run on a Linux cluster with multiple nodes and multiple CPUs per node. Such a cluster usually employs a system to submit jobs through a queue system and run parallel programs. Slurm is a very common job scheduler and workload manager for Linux clusters (https://slurm.schedmd.com). The file `airgam_run.sl` in the `airgam` directory provides a

Slurm batch script file template for running the model on a Linux cluster using Slurm. The file contains some `#SBATCH` Slurm directives for starting and running several parallel model instances. As for Windows (see Sect. 4.1.1), this is done by splitting the number of stations into $nb > 1$ blocks of stations to be run in parallel.

After you have decided on the number of station blocks `nb` you wish to run in parallel, you need to edit two lines of the

`airgam_run.sl` file. First, you need to edit the line `nb=<value>` to insert the total number of station blocks, e.g. `nb=20` if you want to use 20 blocks of stations. Next, you need to edit the line `#SBATCH -array=<ab>-<bb>` to edit the start and end indices of the blocks you wish to run in parallel. E.g. setting `#SBATCH -array=1-20` will run for station blocks number 1 to 20 in parallel using 20 CPUs. Finally, you submit the job simply by issuing the command `sbatch airgam_run.sl`.

## 5 Description of result files

The AirGAM model produces several graphics and text files for each station. These result files are placed in the two sub-directories `main` and `eval` of the output directory `<out_dir>/<ccc>_<ya>_<yb>_<ma>-<mb>`. Here `<out_dir>` is the output directory as set in the model options file `airgam_options.txt`, `<ccc>` the compound used (`no2`, `o3`, `pm10`, `pm2.5`), `<ya>` and `<yb>` the start and end year respectively of the period selected for the trend estimation, i.e. `year_a` and

`year_b` as defined in the options file, and `<ma>` and `<mb>` a three-letter abbreviation of the start and end month (`jan`, `feb`, …, `dec`), of the sub-part of the year used for the calculations, as defined by the variable `subyear` in the options file.

In the following subsections, each result file is described in more detail. We separate the main results, deterministic model evaluation results, and probabilistic model evaluation results. These three types of output are described in Sects. 5.1-5.3 below.

All main result files are described in Sect. 5.1, and these are written to the sub-directory `main` of the output directory. Deterministic and probabilistic evaluation files are described in Sects. 5.2 and 5.3 and are written to the `eval` sub-directory.

Overall, three types of files are being produced by the model:
- Plot files using the format PNG (Portable Networks Graphics) (`.png`)
- Text files of comma-separated (`.csv`) or blank separated (`.txt`) data with one header line with field names
- Text files with results in a more free-format style (`.txt`)



In describing the result files below, `<station>` will denote the station name acronym, and `<ya>` and `<yb>` the start and end year, respectively, for the period selected for the trend estimation. Further, `<yy>` will denote a specific year in the period chosen for cross-validation, with start and end year of cross-validation `<yc>` and `<yd>`, respectively. The cross-validation period is always the same or shorter than the period selected for the trend calculation. All plots are produced in high quality with a resolution of 300 dpi (dots per inches), a height of 2000 pixels, and a width of either 2000 or 4000 pixels, depending on the plot type.

Some results are being produced separately for the meteorology-adjusted and unadjusted GAM models. These files will include the `<adj>` specifier in the file name, with `<adj>=metadj` for the meteorology-adjusted model, and `<adj>=unadj` for the unadjusted model.

There is also a set of files containing specific results for all stations. They have file names on the form `AirGAM_*.csv` and `AirGAM_*.txt`, where the asterisk is replaced by an indication of the type of results. When performing parallel processing with the model, these files will be named `AirGAM<ib>_*.csv` and `AirGAM<ib>_*.txt` instead, where `<ib>` is the index of the block of stations to run for. These indices range from 1 to `nb`, where `nb` is the number of station blocks. After the parallel runs are finished, the user can use the script `run_cat.bat` on Windows or `run_cat.sh` on Linux, which both reside in the airgam main directory, to concatenate these into a set of common `AirGAM_*.csv/txt` files. After this concatenation operation, the user may wish to delete the individual `AirGAM<ib>_*.csv/txt` files.

As an example of the output in this section, we use the station EE0018Ah, a background station in an urban part of Tallin, Estonia. It represents the median station regarding cross-validation correlation results for $NO_2$, which means that half of all stations had a poorer correlation than this and half had a better for $NO_2$. Thus, the station should represent results at individual stations for $NO_2$.

## 5.1 Main results

Below we describe the most central result files from a run with the AirGAM model.

### 5.1.1 The estimated trend curve

The file name is `<station>_gam.trend_<adj>_<ya>_<yb>.png`. An example of this type of plot is shown in Fig. 2.



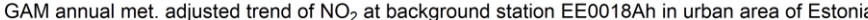

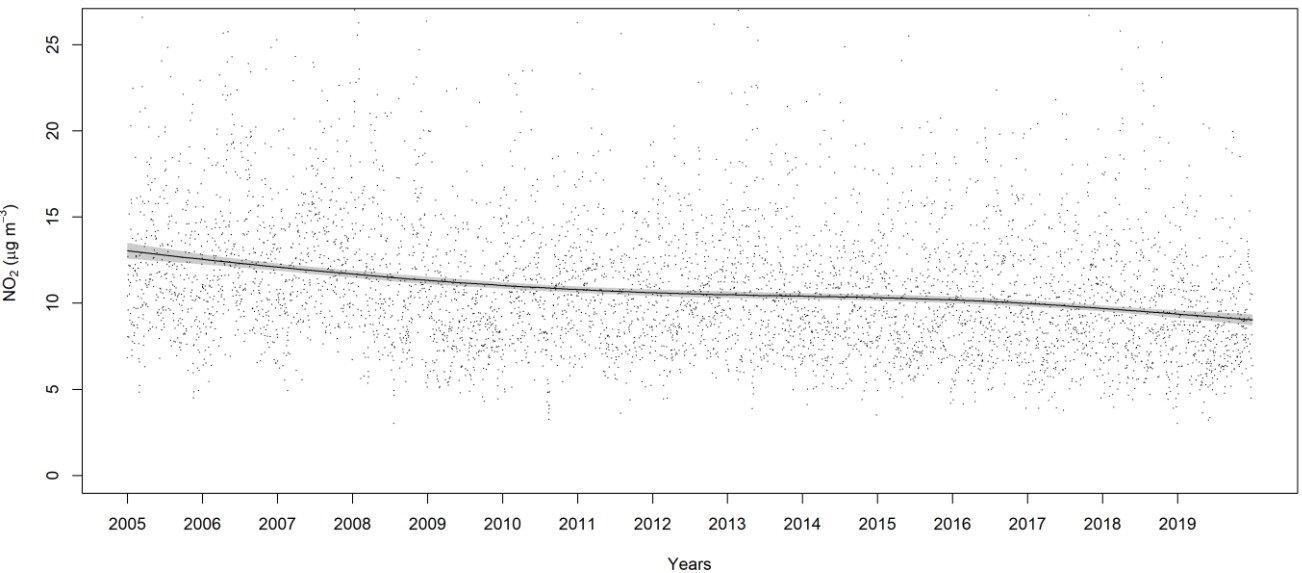

**Figure 2.** The meteorology-adjusted trend curve for $NO_2$ at station EE0018Ah for 2005-2019 (whole years). The units are year (x-axis) and $\mu gm^{-3}$ (y-axis).

This plot shows the smooth response function corresponding to the estimated meteorology-adjusted trend for $NO_2$ at station EE0018Ah ((`<station>`=EE0018Ah) over the period 2005-2019 (`<ya>`-`<yb>`). The dots in the plot are the partial residuals from fitting the current GAM model, i.e. residuals that would have been obtained if dropping this specific term from the model while leaving all other estimates unchanged. The grey shaded area represents a 95 % confidence region for the smooth trend curve. Note that this 95 % region do not necessarily correspond to 95 % confidence intervals pointwise, i.e., for each point on the curve or covariate value, but rather as an average across the curve, over all covariate values (Nychka, 1988; Marra and Wood, 2012). The confidence region and intervals are calculated by the `plot.gam` function in the `mgcv` package. In the call to this routine, we set the parameter `seWithMean=TRUE`; thus, the confidence region also includes the uncertainty about the overall mean. Work by Marra and Wood (2012) suggests that this setting results in improved coverage performance.

As shown from Fig. 2, the trend for $NO_2$ at station EE0018Ah decreases from 2005 to around 2011, where it becomes relatively flat up to 2015 before falling again slightly towards 2020. The trend declined from about 13 $\mu gm^{-3}$ in 2005 to approximately 10 $\mu gm^{-3}$ at the end of 2019, thus decreasing about 3 $\mu gm^{-3}$ over the 15 years.

### 5.1.2 The estimated trend data values

The file name is `<station>_gam.trend_<adj>_<ya>_<yb>.csv`. This is a comma-separated (CSV) text file containing the data used to produce the plot in the previous section. Each row of this file contains a time indicator (year value)




(`years`), followed by the trend curve value of the smooth response function for the trend (`trend`) and the lower and upper 95 % confidence region levels (`trend.025`, `trend.975`). The file always contains 100 values of the trend curve.

### 5.1.3 Smooth response functions plots

The file name is `<station>_gam.smooth_<adj>_<ya>_<yb>.png`. An example of this type of plot is shown in Fig. 3.


This panel of plots shows the smooth response function for each covariate of the meteorology-adjusted model for NO$_2$ at station EE0018Ah (`<station>=EE0018Ah`) based on the years 2005-2019 (`<ya>-<yb>`). Each response function describes an estimated smooth relationship (smooth curve) between the log of the concentrations and the corresponding covariate values from the GAM model regression.


Again, the dots in each plot are the partial residuals from fitting the current GAM model, i.e. residuals that would have been obtained if dropping the specific term from the model while leaving all other estimates unchanged. The grey shaded areas represent 95 % confidence regions for each smooth curve. Note again that these 95 % regions do not necessarily correspond to 95 % confidence intervals pointwise, i.e., for each point on the curve or covariate value, but rather as an average across the
curve, over all covariate values (Nychka, 1988; Marra and Wood, 2012). For these plots, we again set the parameter `seWithMean=TRUE` in the call to the routine `plot.gam` in `mgcv`. Hence, the intervals also include uncertainty about the overall mean, improving coverage performance. The last plot in the panel shows the smooth trend curve as described in Sect. 5.1.1.

As shown in Fig. 3, the concentrations of NO$_2$ at station EE0018Ah decreases with temperature (top left plot) up to about 0 °C and then increases with the temperature above this. For wind speed (top centre), the concentrations continuously decrease with wind speed which is natural. The concentrations vary quite a bit with wind direction (top right), with the lowest concentrations for wind directions from around 90° and highest from around 250-300°. Concentrations decrease with planetary boundary layer height (middle left), which is also natural, and also reduces but only slightly so with relative humidity (middle centre).
Medium cloud cover does not influence the concentration levels much (middle right). Concentrations are also relatively flat during weekdays (bottom left) except for a slight increase on Fridays but are lower during the weekend. The day of the year seems to influence concentrations in a sinusoidal pattern (bottom centre), with the lowest concentrations during summertime and highest during wintertime. The trend curve plot (bottom right) is the same as shown in Sect. 5.1.1 and is commented upon there. The estimated relations between concentrations of NO$_2$ and the meteorological and time covariates are typical for most
Airbase/AQER stations in Europe during the period 2005-2019. For the other compounds, there are different relations, although several patterns are similar, e.g. for wind speed, planetary boundary layer height and day of the week.



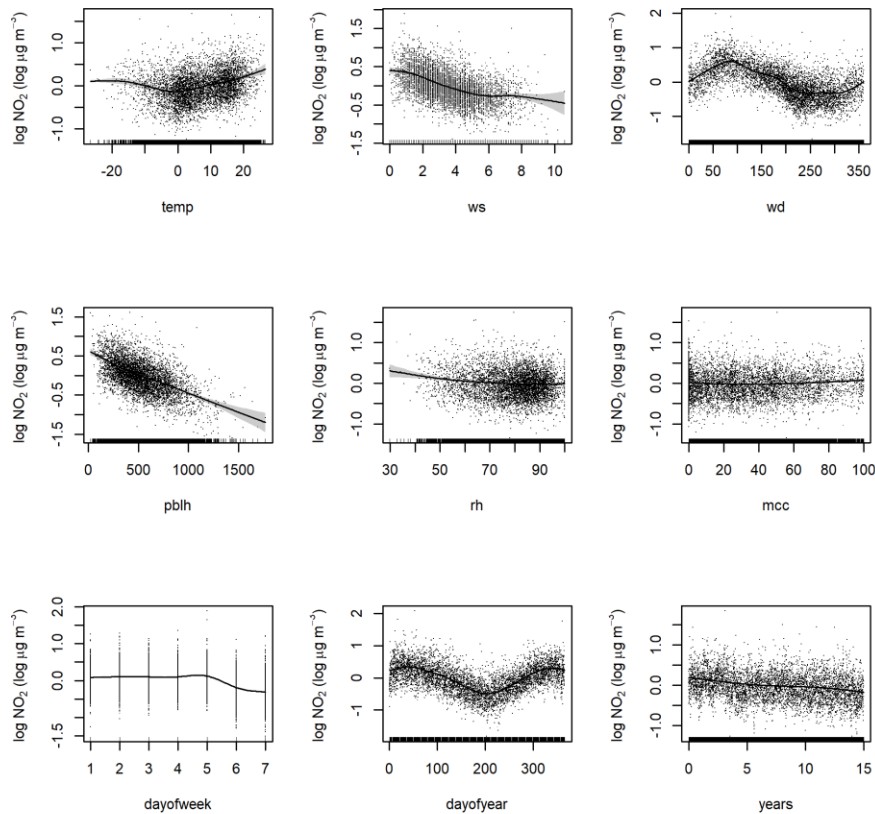

**Figure 3.** Smooth response functions for each covariate for NO$_2$ at station EE0018Ah for 2005-2019. The covariates (x-axes) units are described in Table 1. The unit on the y-axis is log $\mu$gm$^{-3}$.

If seasonal conditioning is used (`use_season_cond=1`), smooth functions will be estimated for each season and output to separate files. The file names will be `<station>_gam.smooth_<season>_<adj>_<ya>_<yb>.png`, with `<season>` the season string. These season strings are taken from the station data files. There will be one such file for each unique value of the season string with plots of the smooth response functions for the indicated season.

**5.1.4 Smooth response functions values**

The file name is `<station>_gam.smooth_<adj>_<ya>_<yb>.csv`. This is a comma-separated (CSV) text file containing the data used to produce the plots in the previous section. Each row of this file contains a row index (`i=1,2,…`), followed by the x- and y-coordinates of the smooth response functions for each covariate (`<cov>.x <cov>.y`) where `<cov>` is the name of the covariate. The files always contain 100 pairs of x- and y-coordinates for each smooth function. Again, if seasonal conditioning is used (`use_season_cond=1`), there will be one such text file for each season. The file

names will be `<station>_gam.smooth_<season>_<adj>_<ya>_<yb>.csv`, with `<season>` the season strings.





### 5.1.5 Regression coefficients and p-values

The file name is `AirGAM_gam.coef_<ya>_yb>.csv`. This is a CSV file containing the smooth response functions beta-coefficients and p-values plus some other results related to the fitted model based on all years used for the trend estimation (`<ya>-<yb>`). Note that this file is only being produced from the meteorology-adjusted model. The file is common to all
stations with a header line and one row of results per station.

Each row contains the station name acronym (`name`), beta coefficients for each covariate (`beta.<cov>`), corresponding p values (`p.<cov>`), GAM regression $R^2$ value (`r.sq`), deviance explained (`dev.expl`), Akaike information criterion (`aic`), a linear regression trend slope coefficient (`beta.linreg`), and its p-value (`p.linreg`). The `beta.<cov>` coefficients
are calculated for each covariate based on the smooth response function's slope between the 0.25 and 0.75 quantiles of the corresponding set of covariate values. The `p.<cov>` values are associated with a null hypothesis of an exactly zero response function for the corresponding covariate. They can be used to reject this null hypothesis in the same way as in linear regression. In addition to the GAM model, a simple linear regression model is run based on concentration ($O_3$) or log of concentration ($NO_2$ and PM) as response variable using only the time variable `years` as a covariate. The `beta.linreg` coefficient with
its p-value `p.linreg` corresponds to the slope coefficient from this linear regression. For $NO_2$ and PM, the slope is transformed back to the original scale.

For $NO_2$ at station EE0018Ah for 2005-2019, all p-values are close to zero ($< 3 \cdot 10^{-5}$), while $R^2$ and the deviance explained are 0.69 and 0.77, respectively. Finally, the linear regression slope is -0.21 $\mu gm^{-3}$ per year with a p-value of $3.4 \cdot 10^{-7}$.

### 5.1.6 Plots of observations and model predictions from cross-validation

The file name is `<station>_gam.pred_<yy>_<yy>.png`. An example of this type of plot is shown in Fig. 4.

The plot shows observed (black curve) and model-predicted (red curve) concentrations of $NO_2$ at station EE0018Ah (`<station>=EE0018Ah`) for 2019 (`<yy>=2019`). The model predictions are based on training the meteorology-adjusted
GAM model on all years for the trend estimation (2005-2019) except for the plotted year (2019). There is one such file being produced for each year `<yy>` of the leave-one-year-out cross-validation period (`<yc>-<yd>`). Here the start and end years for the cross-validation `<yc>` and `<yd>` can be different from `<ya>` and `<yb>` corresponding to a possible sub-period of the whole period defined for trend estimation. In this way, we show how well the model can predict concentrations left out from the training of the GAM model for each year of the cross-validation period.



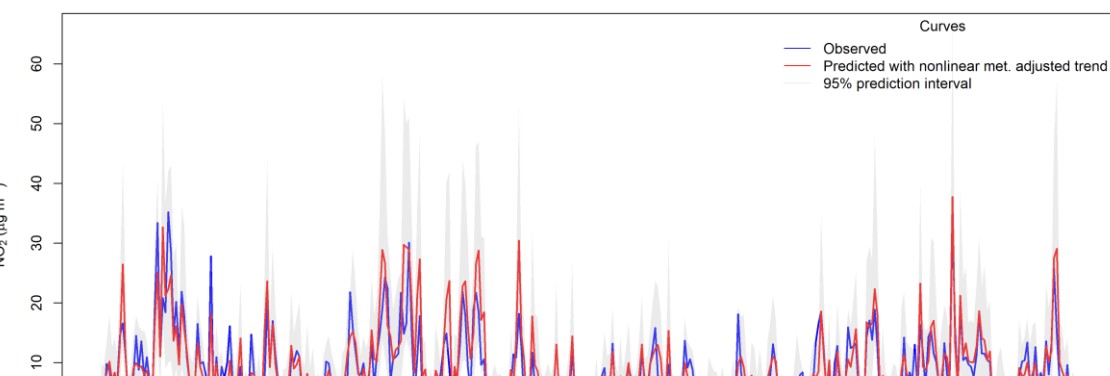

**Figure 4.** Observed (black curve) and model-predicted (red curve) concentrations of NO$_2$ at station EE0018Ah for 2019.

As shown in Fig. 4, there is quite a good correspondence between observed and predicted values for NO$_2$ at station EE0018Ah for the year 2019.

### 5.1.7 Values of observations and model predictions from cross-validation

The name of the file is `<station>_gam.pred_<yy>_<yy>.csv`. This is a CSV file containing the data used to produce the file in the previous section. Each row of the file contains the date (`yyyy-mm-dd`), followed by the observed and model-predicted concentrations for the station `<station>` for the left-out year `<yy>`.

### 5.1.8 Model linear predictor values

The name of the file is `<station>_gam.linpred_<ya>_<yb>.csv`. This is a CSV file containing observed and model linear predicted daily concentrations for the indicated station (`<station>`) using predictions from a fit to the data for all years or sub-parts of years used for the meteorology-adjusted trend estimation (`<ya>-<yb>`). Each row of the file contains the date (`yyyy-mm-dd`) and the following data: observed concentration (`linpred.obs`), predicted concentration (`linpred.pre`), the constant or intercept term (`beta0`), followed by the contribution to the predicted concentration from each smooth covariate response function for the covariate values for the current date (`term.<cov>`), where `<cov>` ranges over the set of covariate names. The sum of the covariates' contributions plus the constant term equals the predicted concentration value. It is important to note that the observations and predictions in this file are the concentrations on the scale of the GAM linear predictor. This means that the concentrations are on the original scale for O$_3$ (µgm$^{-3}$) and the logarithmic scale for NO$_2$, PM$_{10,}$ and PM$_{2.5}$ (log µgm$^{-3}$).





### 5.1.9 Plots of (sub-) annual and monthly averages and medians of observations and model predictions

The file names are `<station>_gam.aave_<ya>_<yb>.png` and `<station>_gam.mave_<ya>_<yb>.png`, for (sub) annual and monthly averages, with examples of plots shown in Fig. 5 and 6, respectively. For medians, the string `ave` in the filenames is replaced by `med`.

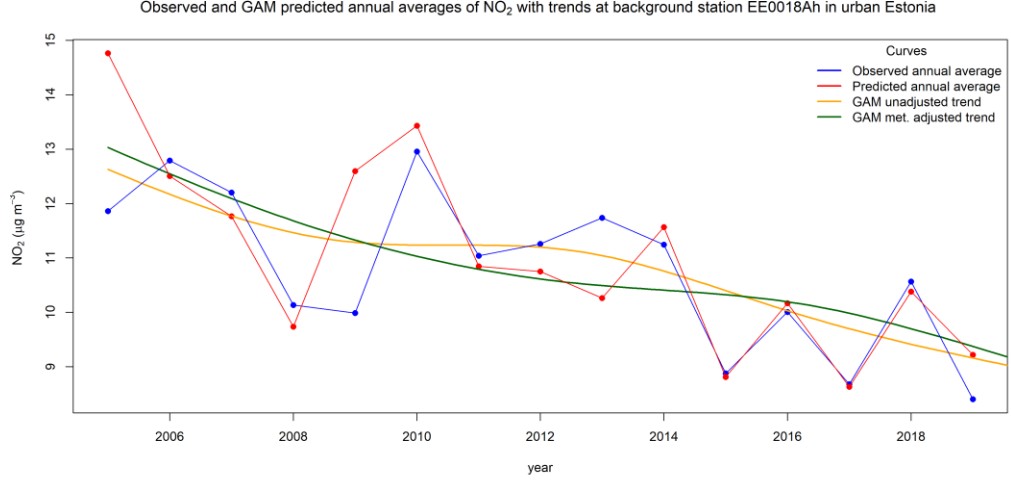

**Figure 5.** Observed (blue curve) and predicted (red curve) annual average concentrations of $NO_2$ at station EE0018Ah for 2005-2019. The
orange and dark green curves show the unadjusted and meteorology-adjusted trends, respectively.

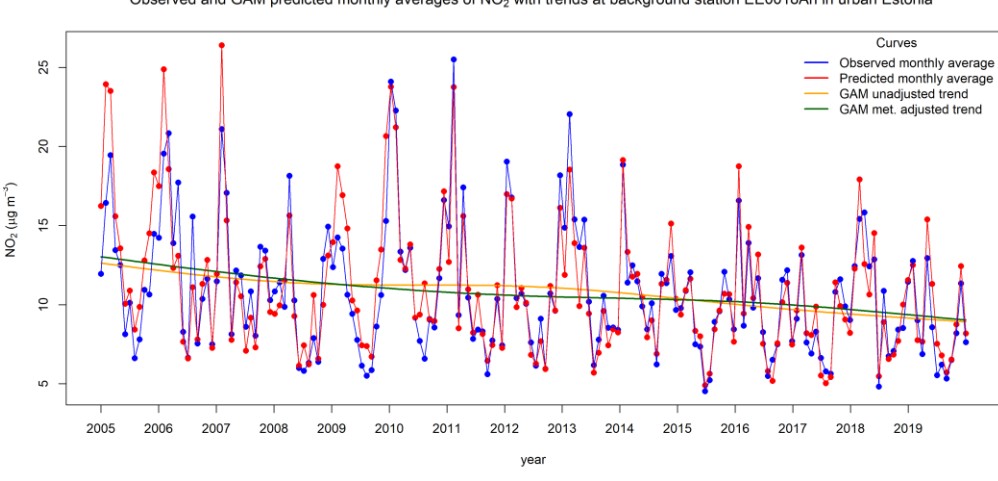

**Figure 6.** Observed (blue curve) and predicted (red curve) monthly average concentrations of $NO_2$ at station EE0018Ah for 2005-2019. Again, the orange and dark green curves show the unadjusted and meteorology-adjusted trends, respectively.

The plots show observed (blue curve) and predicted (red curve) annual and monthly average concentrations of $NO_2$ at station
EE0018Ah (`<station>`) for 2005-2019 (`<ya>-<yb>`). The orange and dark green curves show the unadjusted and meteorology-adjusted trends, respectively. In these plots, we use annual and monthly averages of the predictions from the





cross-validation for all years used for the trend estimation (`<ya>-<yb>`). Thus, the model predictions will always be from the meteorology-adjusted model.

As shown from Figs. 5-6, there is overall a good correspondence between the averaged observations and predictions for $NO_2$ at station EE0018Ah over this period.

Examples of plots of annual and monthly medians of observed and predicted concentrations are shown in Figs. 7-8. For the median plots, the trend curves are not plotted.

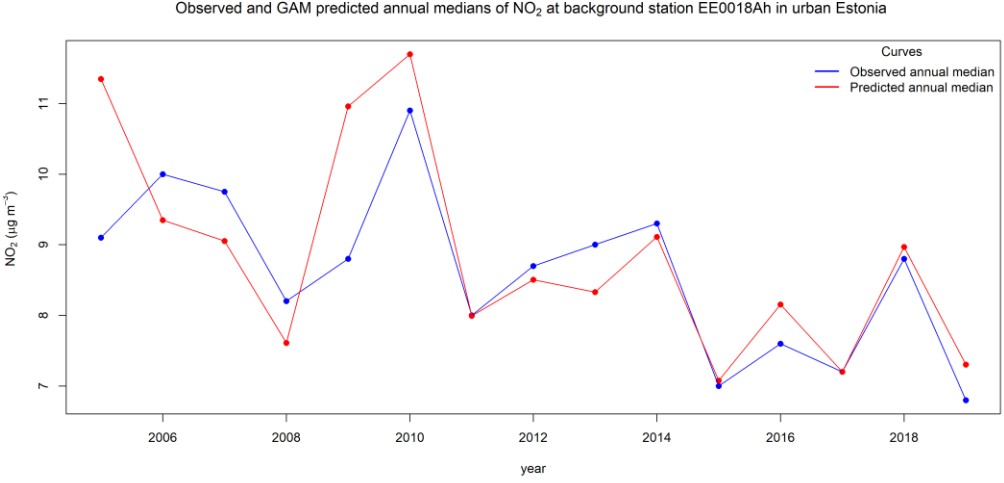


**Figure 7.** Observed (blue curve) and predicted (red curve) annual median concentrations of $NO_2$ at station EE0018Ah for 2005-2019.

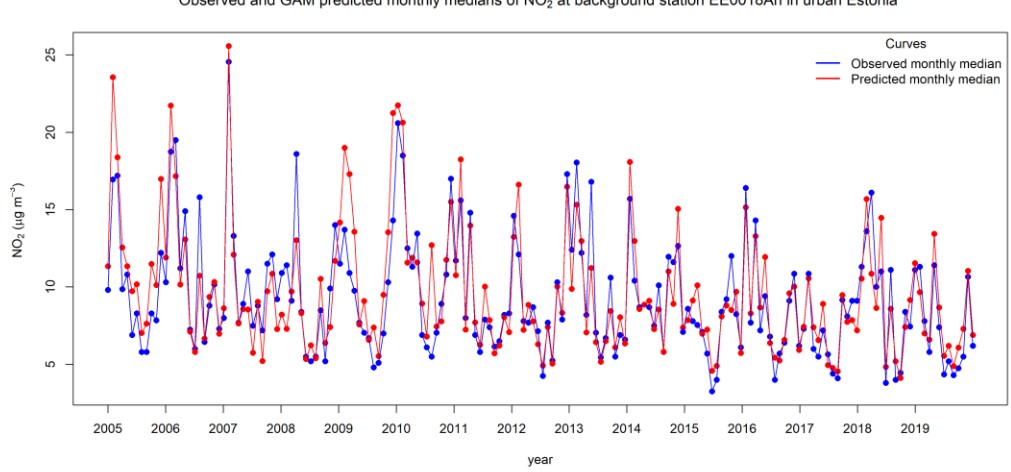

**Figure 8.** Observed (blue curve) and predicted (red curve) monthly median concentrations of $NO_2$ at station EE0018Ah for 2005-2019.





Again, there is overall a good correspondence between median values of observed and predicted concentrations of NO2 at
station EE0018Ah over this period.

### 5.1.10 Values of (sub-) annual and monthly averages and medians of observations and model predictions

The observed and predicted (sub) annual and monthly averages with trend curve values are also written to text files. The file
names are `<station>_gam.aave_<ya>_<yb>.csv` and `<station>_gam.mave_<ya>_<yb>.csv`, respectively.
Each row of the (sub) annual averages file contains the year (`year`), the observed (`obs.aave`), and predicted (`pre.aave`)
(sub) annual averages for each year, followed by the trend curve values (`trend.<adj>`), with the year ranging from `<ya>`
to `<yb>`. Likewise, each row of the monthly averages file contains the year (`year`) and month (`month`), the observed
(`obs.mave`) and predicted (`pre.mave`) monthly averages for each year, followed by the trend curve values
(`trend.<adj>`), again with the year ranging from `<ya>` to `<yb>`. Missing values (NA) are inserted in months outside the
defined sub-year period (`<ma> − <mb>`), or outside the period for cross-validation (`<yc> − <yd>`). Similarly, observed and
predicted (sub) annual and monthly medians are written to text files. In this case, the string `ave` in the filenames and the
headings in the files are replaced by `med`. No trend values are written in this case.

### 5.1.11 Processed stations

The file name is `AirGAM_stations.csv`. This is a CSV file containing a list of all stations processed by the AirGAM
model when run. Each row of the file includes the station name acronym (`name`), longitude (`lon`), latitude (`lat`), height
above sea level (`z` in m), station type (`type`), and station area characteristics (`area`). Here `type` is a text string describing
the station type (background or traffic). The `area` is a text string describing the station's surrounding area (rural, suburban or
urban). Only stations the model actively processes are listed in the file. It will thus contain a subset of the stations in the input
stations file as described in Sect. 3.4.

### 5.1.12 Program log-file

The file name is `AirGAM_log.txt`. This is a text file containing statuses and eventual warnings and errors produced by the
AirGAM model when run. Status messages include model version, time and date of the model run, details of the run
environment such as OS version, machine and user information, R and R packages versions, the working directory, top
input/output directories, most of the options used, and major milestones reached during execution. Lines with warning/error
messages contain the warning/error code, the station name acronym, the current date (year, month, day) of the data processed,
and some explanatory text. A list of the various types of warnings and errors issued by the model with each description is given
in Appendix B.





## 5.2 Deterministic model evaluation

Below we describe the result files from the deterministic model evaluation part of the AirGAM model.

### 5.2.1 Model summary

The file name is `<station>_gam.summary_<adj>_<ya>_<yb>.txt`. This is a text file containing the results of running the `summary.gam` function in the `mgcv` package (Wood, 2017) in connection with a GAM model run for the whole period for the trend analysis (`<ya>-<yb>`). This file contains first the name of the response distribution (Gaussian or Gamma), the type of link function used (identity or log), the formula used in the call to the GAM model solver (`bam` or `gam`), and the results for the intercept (estimate, standard error, t-value, and significance probability). Then for each smooth covariate in the

GAM model, the empirical degrees of freedom (`edf`), the reference degrees of freedom (`ref.df`), the F-value (`F`), and the p-value (`p`) is given, together with corresponding significance codes. Finally, the file contains the adjusted $R^2$ value, the percentage of deviance explained (`dev.expl`), the penalised likelihood final objective function value (`fREML`), the scale estimate (`scale`), the number of data values (`n`), and the residual degrees of freedom (`res.df`).

For $NO_2$ at station EE0018Ah for 2005-2019, all covariates are highly significant, with p-values very close to zero. The $R^2$ value is 0.693, which means that around 69 % of the variation in the concentrations can be explained by the covariates, which is quite good. Also, the empirical degrees of freedom value `edf` for each covariate is well below the corresponding reference degrees of freedom value `ref.df`, except perhaps for the covariate `dayofweek`, but this represents only a minor issue here. However, if `edf` should become close to `ref.df` for the trend term, one should consider increasing the number of basis

functions for the trend term, especially if one wants to capture more of the variation in the trend. This can be done through the control variable `k_years`.

### 5.2.2 Model check plots

The file name is `<station>_gam.check_<adj>_<ya>_<yb>.png`. An example of this type of plot is shown in Fig. 9.


This panel of plots shows various evaluation plots for the meteorology-adjusted model for $NO_2$ at station EE0018Ah for 2005-2019 (`<ya>-<yb>`) produced by the `gam.check` routine in the `mgcv` package. The upper left plot shows the model residual quantiles against theoretical quantiles based on a Gaussian distribution assumption for the residuals. Ideally, the black data points corresponding to the individual residual values should follow the straight red line for a good model fit. The upper right

plot shows model residuals against the model linear predictor. Ideally, the individual data points (circles) should have the same distribution along the y-axis for all x-axis values. The lower left plot shows a frequency histogram of the model residuals. Ideally, the histogram should be symmetric and Gaussian in shape. And finally, the lower right plot shows the response, i.e.





the observed concentrations, against the model fitted values. Ideally, the data points (circles) should be as close as possible to a 1:1 reference line through the origin.

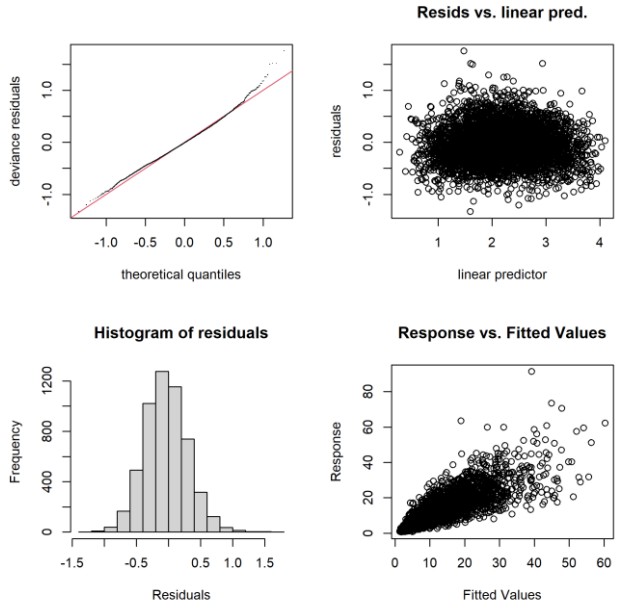

**Figure 9.** Model check plots for the meteorology-adjusted model for $NO_2$ at station EE0018Ah for 2005-2019.

As shown in Fig. 9, we see that for $NO_2$ at station EE0018Ah for 2005-2019, the model residual quantiles (upper left plot) follow the theoretical quantiles of a Gaussian distribution quite well except for the upper tail part where there is a certain deviation. The other plots in this panel show excellent results, with the ideal type of plots in all cases.

**5.2.3 Model check table**

The file name is `<station>_gam.check_<adj>_<ya>_<yb>.txt`. This text file contains additional data output from the `gam.check` routine in `mgcv`. This file includes details from the convergence of the numerical solution method used for the GAM model. The most important output, however, is a table which for each smooth covariate function shows the number of basis functions minus one (`k'`), the empirical number of degrees of freedom (`edf`), the k-index value (`k-index`), and the associated p-value (`p-value`). The user should check this table for any low `p-value` ($< 0.05$) with `k-index` $< 1$ to ensure that the `edf` value is not too close to the `k'` value.

For $NO_2$ at station EE0018Ah for 2005-2019, the model converged in 14 iterations with an objective function gradient close to zero and a positive definite Hessian matrix. The basis dimension checking results are all ok for the covariates with high p-values, except for the trend term, where the p-value is very small ($< 2 \cdot 10^{-16}$). Note that the p-values are not associated with the





significance of covariates here and should be high for all variables. However, for the trend term, `edf`=3.38 and well below `k'`=4, and thus, the number of basis functions is sufficiently large also for this variable.

### 5.2.4 Model evaluation

The file name is `AirGAM_gam.eval_<yc>_<yd>.csv`. This is a CSV file containing the results of evaluating the model
predictions against observations from the cross-validation period (`<yc>-<yd>`) using the routine `modStats` from the `openair` package in R. This is also a file common to all stations with a header line and one row of results per station. Each row contains the station name acronym (`name`), the number of cases (days) used for the model evaluation (`n`), and then the following model evaluation statistics: fraction of predictions within a factor of two (`fac2`), mean bias (`mb`), mean gross error (`mge`), normalised mean bias (`nmb`), normalised mean gross error (`nmge`), root mean squared error (`rmse`), Neyman-Pearson
correlation coefficient (`r`), coefficient of efficiency (`coe`), and index of agreement (`ioa`). The manual pages for the `modStats` routine in `openair` contains a detailed description of what these parameters represent and how they are calculated.

For $NO_2$ at station EE0018Ah for 2005-2019, the evaluation results are based on 5398 cases (days) in total. In around 94 % of
these days, the predicted concentrations are within a factor of two of the observations (fac2=0.944). Further, the mean bias (`mb`) is only around 0.25 $\mu gm^{-3}$ with a normalised mean bias (`nmb`) of only 0.023, which is quite good. The root mean squared error (`rmse`) is also relatively low, with a value of 4.8 $\mu gm^{-3}$. Also, the Pearson correlation coefficient (`r`) and index of agreement (`ioa`) are pretty good, with values of 0.82 and 0.75, respectively. Finally, the coefficient of efficiency (`coe`) also shows a decent value of around 0.5 for this compound and station.

### 5.2.5 Concurvity analysis

The file name is `AirGAM_gam.ccuv_<adj>_<ya>_<yb>.csv`. This is a CSV file containing so-called concurvity values for each smooth covariate in the AirGAM model based on all years used for the trend estimation (`<ya>-<yb>`). This is also a common file for all stations with a header line and one row of results per station. Concurvity is to GAM modelling as collinearity is to multiple linear regression; it describes the degree to which covariates can be viewed as independent of each
other. More specifically, for GAM models, the concurvity value for a given smooth covariate indicates to what degree this covariate is superfluous and could be replaced by a linear or nonlinear combination of the remaining smooth covariates in the model. It is thus important to check for this as part of the modelling. Concurvity values are calculated using the `concurvity` routine in the `mgcv` package and range from 0 (best value) to 1 (worst value). Each row of the result file contains the station name acronym (`name`), type of concurvity value (`type`), followed by a concurvity value for each smooth covariate
(`ccuv.beta<i>`) for `<i>`=1,2,… .





A concurvity value of type `worst` from the `concurvity` routine below 0.8 (approximately) is often taken to indicate that the corresponding smooth covariate is probably not severely dependent on the other smooth covariates (Ross, 2022). A higher value is more troublesome and suggests that it might be redundant and replaced by a linear or nonlinear combination of the

other smooth covariates. In this case, the covariate response function will be challenging to estimate appropriately due to identifiability problems. However, this is a relatively pessimistic measure of concurvity according to the help pages for the `concurvity` routine in `mgcv`. There the `estimate` type of concurvity is presented as somewhat better balanced than the other two, i.e. `worst` and `observed`, since "It does not suffer from the pessimism or potential for over-optimism of the previous two measures", even though, as also stated, that it is "less easy to understand". Thus, due to this better balance, we

apply this measure of concurvity in AirGAM rather than the overly pessimistic one. However, we reduce the limit to 0.4 to indicate potential problems with identifiability. For values above this, a warning is issued to the log file. If seasonal conditioning is used (`use_season_cond=1`), separate concurvity values will be output to this file for each season. This will be indicated in the header line.

For $NO_2$ at station EE0018Ah for 2005-2019, the concurvity values for the various smooth covariates are all small (below 0.4), which is good and indicates that they are all reasonably independent of each other.

### 5.2.6 Autocorrelation and partial autocorrelation function plots

The file name is `<station>_gam.acf_<adj>_<ya>_<yb>.png`. An example of this type of plot is shown in Fig. 10.

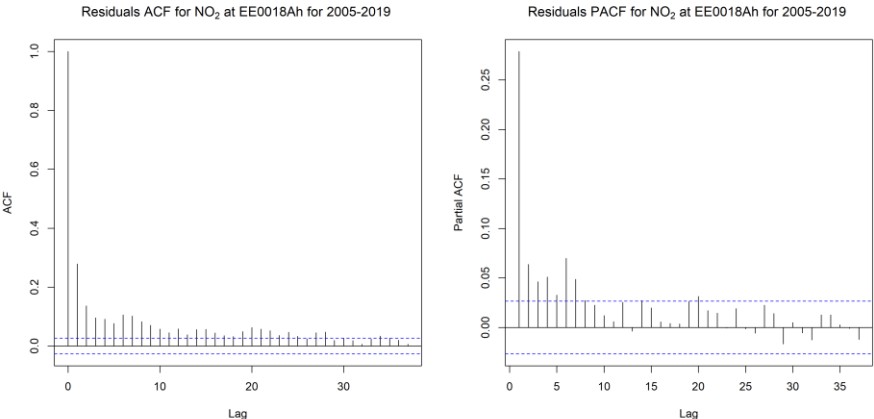

**Figure 10.** A plot of the autocorrelation function (left) and partial autocorrelation function (right) for the meteorology-adjusted model error residuals for $NO_2$ at station EE0018Ah for 2005-2019.

This figure shows plots of the autocorrelation function (left) and the partial autocorrelation function (right) for the meteorology-adjusted model error residuals for $NO_2$ at station EE0018Ah for 2005-2019. Ideally, the GAM model residuals should be independent random variables, and thus, the autocorrelation function values should be close to zero for all positive time lags.





The same applies to the partial autocorrelation function values; they should also be close to zero for all positive time lags. The level below which the autocorrelation values are non-significant (close to zero) is indicated by the horizontal dashed line(s).

As shown in Fig. 10, for this compound and station, autocorrelation values are significantly positive from time lag one and onwards, decaying slowly with the time lag. For the partial autocorrelation, the lag-1 value is most significant with a value of 1140     around 0.25, while the other values are much smaller (although a few significantly different from zero).

Running the `gamm` routine, in this case, using the option `incl_ar1=1`, handles autocorrelations by including an AR(1) model for the residuals. This results in (nearly) non-significant correlations at all time lags, as shown in the plots to the right in Fig. 11.

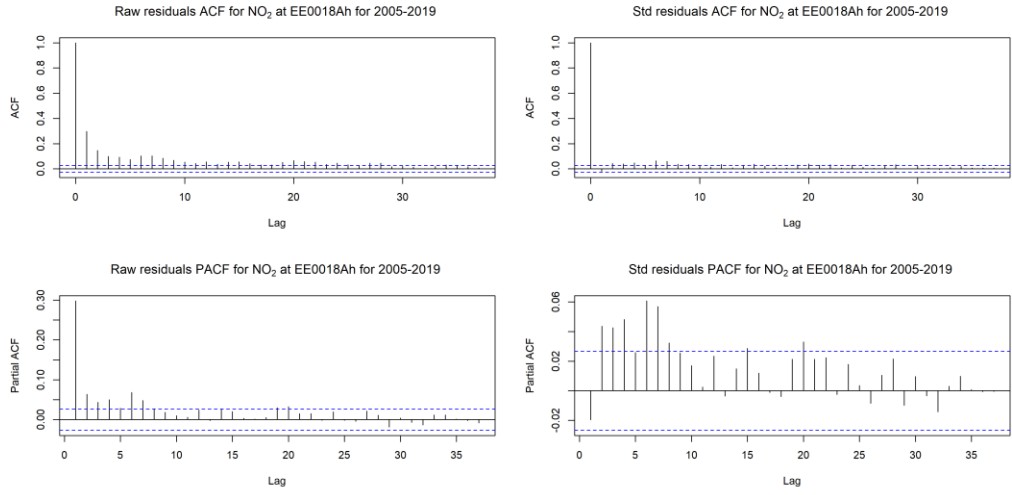


**Figure 11.** The left plots are as in Fig. 10, while the right plots show the effect of including an AR(1) model for the residuals.

### 5.2.7 Autocorrelation and partial autocorrelation function values

The file name is `AirGAM_gam.acf_<adj>_<ya>_yb>.csv`. This is a CSV file containing the autocorrelation and partial autocorrelation values for the residuals for the first ten lags (days) based on the fitted model for all years used for the trend 1150     estimation (`<ya>-<yb>`). It is a file common to all stations with a header line and one row of results per station. Each row of this file contains the station name acronym (`name`), lag-1 to lag-10 autocorrelation values (`acf.1`,…, `acf.10`), and lag-1 to lag-10 partial autocorrelation values (`pacf.1`,…, `pacf.10`). Ideally, all these values should be zero or close to zero, corresponding to independent or nearly independent model error residuals.

### 5.2.8 A conditional quantile plot

The file name is `<station>_gam.cond_quant_<yc>_<yd>.png`. An example of this type of plot is shown in Fig. 12.



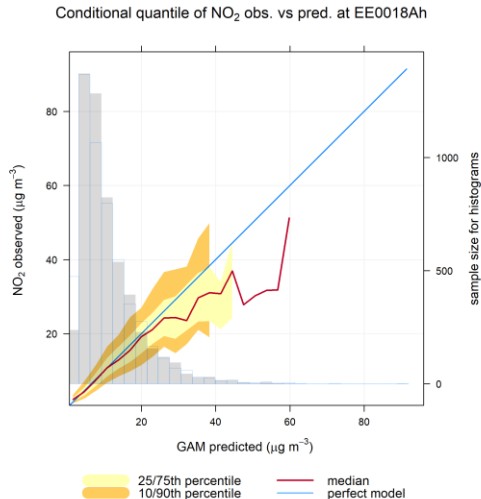

**Figure 12.** Conditional quantile plot for the meteorology-adjusted model for NO$_2$ at station EE0018Ah for 2005-2019.

This is a so-called conditional quantile plot for the model here shown for NO$_2$ at station EE0018Ah for the cross-validation years 2005-2019 (`<yc>-<yd>`). It is produced by the `conditional.Quantile` routine in `openair`. The plot shows the meteorology-adjusted model prediction quantiles against the observed concentration quantiles. The median of the model quantiles is shown as the dark red curve, while 25/75 and 10/90 percentiles are shown as the light yellow and orange-brown shaded regions. Ideally, the dark red curve should perfectly follow the straight light blue line. The background shows histograms of the model predictions in dark grey and histograms of the observations in light blue. Ideally, these two histograms should be identical.

As shown from Fig. 12, the median of model predicted quantiles follows the observed ones almost perfectly up to around 25 µgm$^{-3}$ before the two start to deviate. But the 25/75 percentile light-yellow region of the model predicted quantiles still contains the observed concentration quantiles (straight light blue line) for all values up to around 40 µgm$^{-3}$, which is good. For the higher concentrations, the quantiles deviate more. We also note that the two histograms are pretty similar, which is good.

**5.2.9 Taylor diagram plot**

The file name is `<station>_gam.taylor_<yc>_<yd>.png.`. An example of this type of plot is shown in Fig. 13.

This is a so-called Taylor diagram plot for the model here shown for NO$_2$ at station EE0018Ah for the cross-validation years 2005-2019 (`<yc>-<yd>`). The TaylorDiagram routine produces it in `openair`.

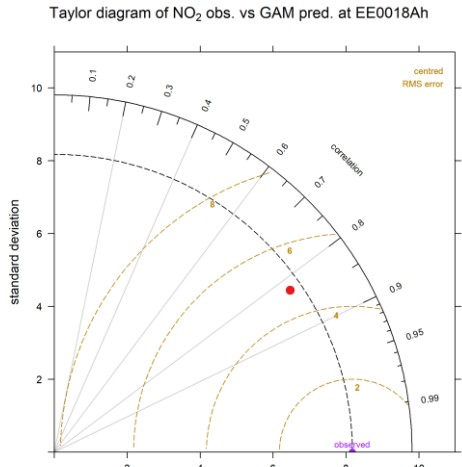

**Figure 13.** Taylor diagram plot for the meteorology-adjusted model for NO₂ at station EE0018Ah for 2005-2019.

As shown in Fig. 13, the model point (red dot) is not very far from the ideal observed point (purple dot). More specifically, the model point is in the sector between correlation levels of 0.8 and 0.9 and is quite close to the dashed black curve (circle) emanating from the observed point. Further, the dashed orange-brown lines indicate an RMSE value between 4 and 6.

**5.3 Probabilistic model evaluation**

Since model predictions from AirGAM are point predictions and come with an associated probability distribution for each predicted concentration value, it is also possible to evaluate the model against observations using probabilistic tools and concepts. Gneiting et al. (2007) and Wilks (2019, Ch. 9) contain a thorough description of this way of assessing a prediction model. Below we describe the result files from AirGAM for this type of model evaluation. Note that these results are only being produced from the meteorology-adjusted model predictions in AirGAM.

**5.3.1 PIT histogram**

The file name is `<station>_gam.pit_hist_<yy>_<yy>.png`. An example of this type of plot is shown in Fig. 14.

This is a so-called PIT (Probability Integral Transform) histogram plot for the model shown here for NO₂ at station EE0018Ah for 2019 (`<yy>=2019`). The plot shows a histogram of the observed concentrations compared with the model probabilistic predictions converted into corresponding probability values between 0 and 1. The PIT histogram can be viewed as a continuous limit of the rank histogram where the latter is based on a finite set of samples from the predictive distribution (Gneiting et al., 2007; Wilks, 2019, Ch. 9). In the figure, the predictions are based on training the model on all years for the trend estimation (2005-2019) except for the plotted year (2019).

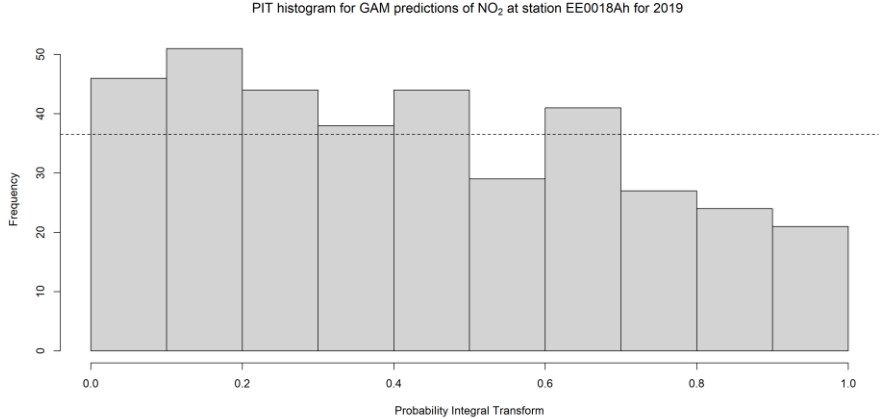

**Figure 14.** PIT histogram plot for the meteorology-adjusted model for NO$_2$ at station EE0018Ah for 2019.

Since the predictive distributions cannot be represented analytically (see Sect. 2.4), the calculations of the PIT values in AirGAM are generally based on taking a sufficiently large number of samples from the unconditional predictive distribution for each day (currently 100) and calculating an empirical cumulative distribution probability value using the corresponding observed value. Ideally, the PIT histogram should be uniform if the model's predictions are properly probabilistically calibrated relative to the actual observations. If the model predictions are too low, the histogram will be biased (skewed) to the right; if

they are too high, it will be biased (skewed) to the left. Also, if the predictions are too narrow (too low prediction uncertainty), the histogram will be U-shaped, while it will have an inverse U-shape if the predictions are too broad (too high prediction uncertainty).

As shown from Fig. 14, the histogram is somewhat biased and skewed to the left, i.e. there are lower PIT-values than high.
This means that the model predictions for the station EE00a8Ah for 2019 seem to be somewhat too high compared with the observations.

The plot always shows PIT values on the x-axis and frequency on the y-axis, and the horizontal dashed line corresponds to a uniform histogram. According to Gneiting et al. (2007), 10-20 bins used to define a PIT histogram seems to be sufficient for
most purposes. We apply ten bins in our implementation generally.

### 5.3.2 Empirical CDF of PIT values

The file name is `<station>_gam.pit_ecdf_<yy>_<yy>.png.`. An example of this type of plot is shown in Fig. 15.

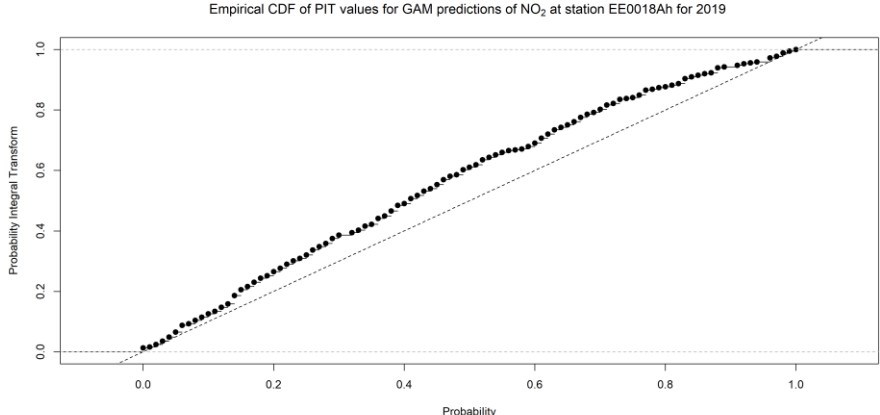

**Figure 15.** The plot of the empirical cumulative distribution function of the PIT values for the meteorology-adjusted model for $NO_2$ at station EE0018Ah for 2019.

This is a plot of the empirical cumulative distribution function (CDF) of the PIT values for the model here shown for $NO_2$ at station EE0018Ah for 2019 (`<yy>=2019`). The plot always shows theoretical cumulative probability values on the x-axis and PIT cumulative probability values on the y-axis. Ideally, the empirical CDF of the PIT values should stay close to the ideal 1:1 dashed reference line if the predictions from the model are correctly probabilistically calibrated relative to the actual observations (Gneiting et al., 2007). If the model predictions are too low, the CDF values will tend to lie below the 1:1 line, while if predictions are too high, the CDF values will tend to lie above the 1:1 line.

From Fig. 15, the CDF values all lie above the 1:1 line. This indicates that the model predictions are too high compared to the observations for this station and year.

### 5.3.3 Marginal empirical CDFs of observations and predictions

The file name is `<station>_gam.marg_ecdf_<yy>_<yy>.png.`. An example of this type of plot is shown in Fig. 16.

This is a plot of the marginal empirical CDFs of observed (blue curve) and predicted (red curve) values for the model here shown for $NO_2$ at station EE0018Ah for 2019 (`<yy>=2019`). Ideally, the two marginal empirical CDFs should stay close together overall if the model's predictions are properly marginally calibrated relative to the actual observations (Gneiting et al., 2007). The plot always shows concentration values on the x-axis and marginal CDF probability values on the y-axis.



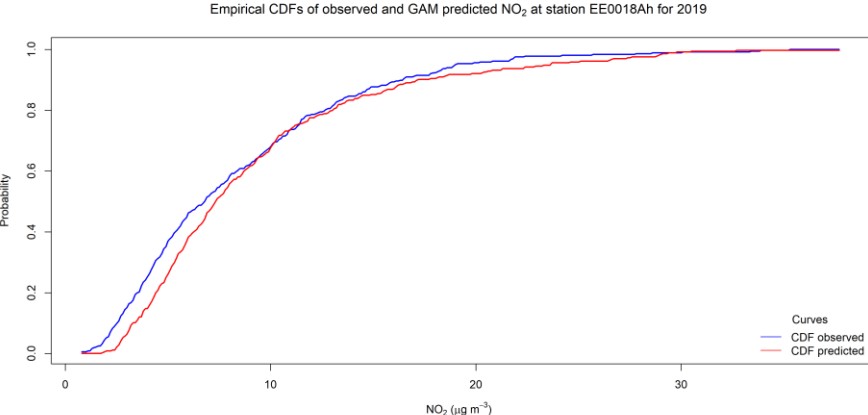

**Figure 16.** The plot of the marginal empirical CDFs of observed and predicted values for the meteorology-adjusted model for $NO_2$ at station
EE0018Ah for 2019.

From Fig. 16, we can see that the marginal CDF probabilities for the observations are generally higher than the marginal CDF
probabilities for the predictions for all concentration levels except for concentrations above around 30 $\mu gm^{-3}$, where the curves
are pretty close. This again shows that the model predictions are too high compared with the observations, also marginally, for
this station and year, except for the highest concentrations.

### 5.3.4 Sharpness diagram

The file name is `<station>_gam.sharp_<yy>_<yy>.png`. An example of this type of plot is shown in Fig. 17.

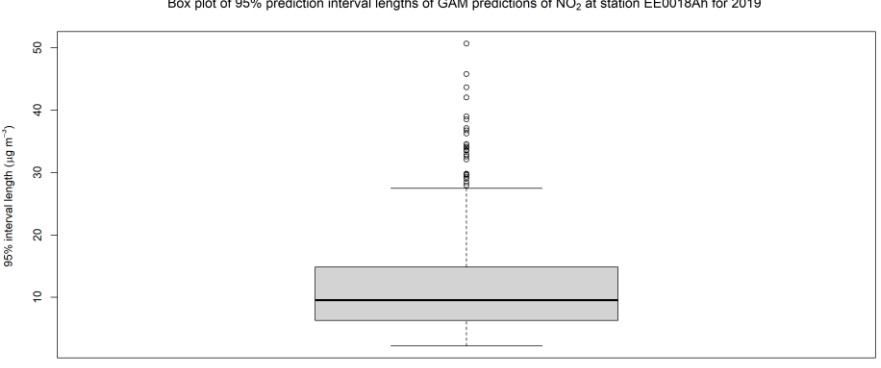

**Figure 17.** A sharpness diagram box plot of 95 % uncertainty intervals of predictions for the meteorology-adjusted model for $NO_2$ at station
EE0018Ah for 2019.

This is a so-called sharpness diagram in the form of a box plot of 95 % uncertainty interval lengths of predictions from the
model here shown for $NO_2$ at station EE0018Ah for 2019 (`<yy>=2019`). The plot always shows concentration values on the





y-axis. Ideally, such box plots should be relatively tight if the model produces sharp predictions, i.e. predictions with low uncertainties (Gneiting et al., 2007).


From Fig. 17, we can see that the model predictions have 95 % uncertainty intervals of length around 10 $\mu gm^{-3}$ on average, with 50 % of the interval lengths between 8-13 $\mu gm^{-3}$. Only occasionally are the interval lengths above around 28 $\mu gm^{-3}$. Thus the model predictions seem to be reasonably sharp overall for this station and year.

### 5.3.5 CRPS scatter plots

The file name is `<station>_gam.crps_<yy>_<yy>.png`. An example of this type of plot is shown in Fig. 18.

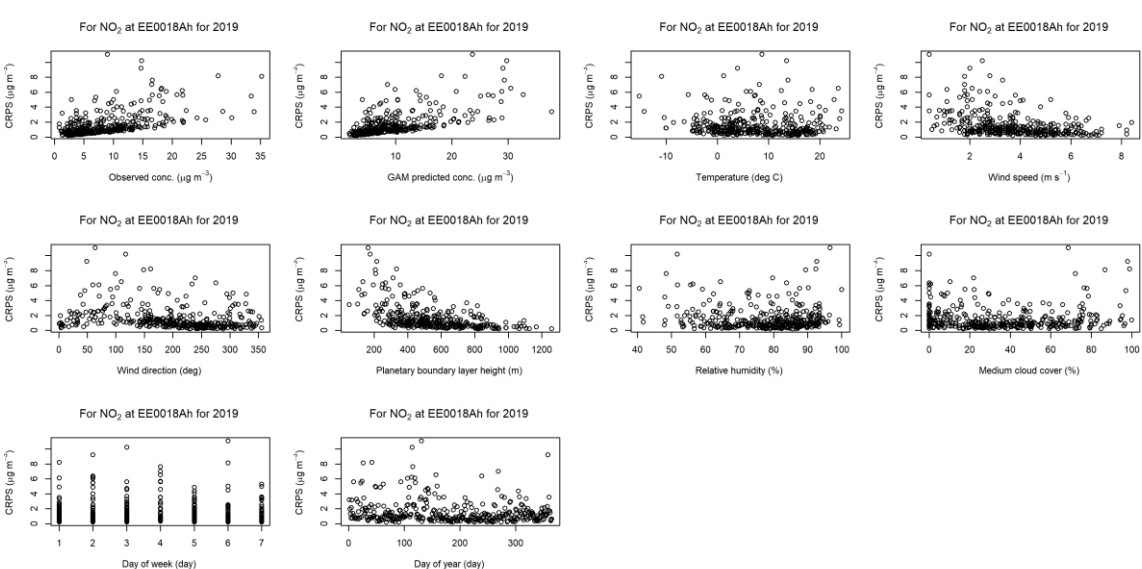

**Figure 18.** Scatter plots of daily CRPS values against observations, model predictions and covariates for the meteorology-adjusted model for NO$_2$ at station EE0018Ah for 2019 (`<yy>=2019`).

The figure shows scatter plots of daily CRPS (Continuous Ranked Probability Score) values against daily values of
observations, model predictions and covariates for the model for NO$_2$ at station EE0018Ah for 2019 (`<yy>`= 2019).

The Continuous Ranked Probability Score (CRPS) (Gneiting and Raftery, 2007; Wilks, 2019, Ch. 9) is a numerical measure of a model's predictive performance considering both calibration and sharpness. It is calculated from the model's predictive distribution and daily observed values. For CRPS, smaller values are better, the optimal value being zero, which corresponds
to a predictive distribution placed precisely at the observation value with no spread. Smaller values of CRPS correspond to predictive distributions being close to the observations, with a small spread. In comparison, larger values indicate the opposite, either through a significant bias between the prediction and the observation (poor calibration) or the predictive distribution has a large spread around the observation (poor sharpness). The CRPS always has the same unit as the concentrations, i.e. $\mu gm^{-3}$.





As shown from Fig. 18, the daily CRPS values generally increase with increasing observations and model predictions (first two scatter plots in the top row). We also see that the CRPS values are pretty even with temperature (following plot in the top row). Further, CRPS is highest for the lower wind speeds with wind directions from the east (50°-100°). This is related to situations with relatively low planetary boundary layer heights (below around 300 m). There is no clear pattern for relative humidity and medium cloud cover, but the highest CRPS values seem to occur during wintertime and spring of 2019. Thus,

during these conditions, the model has more difficulties accurately predicting observed concentrations of $NO_2$ at this station.

### 5.3.6 CRPS box plots

The file name is `<station>_gam.crps_<yc>_<yd>.png`. An example of this type of plot is shown in Fig. 19.

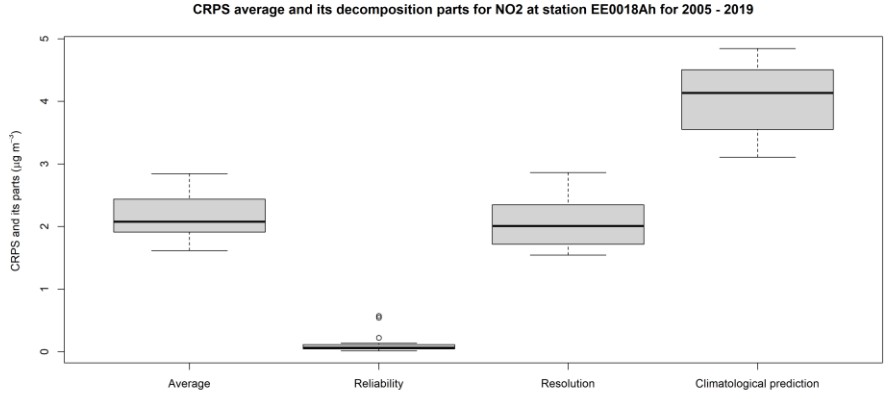

**Figure 19.** Box plots of CRPS averages and their reliability, resolution and climatological prediction uncertainty parts for the meteorology-
adjusted model for $NO_2$ at station EE0018Ah for 2005-2019.

The figure shows box plots of CRPS annual averages with their reliability, resolution and climatological prediction uncertainty parts for the model predictions of $NO_2$ at station EE0018Ah for the cross-validation years 2005-2019 (`<yc>-<yd>`).

According to Hersbach (2000), an average CRPS over a given period (e.g. a year) can be partitioned into a reliability part, a
resolution part, and a climatological prediction uncertainty part as follows:

$$CRPS_{aver} = Reli - Reso + CRPS_{clim}.$$

Here the reliability part is closely connected to the probabilistic calibration condition, i.e., the uniformity of rank or PIT-
histograms. In contrast, the resolution and climatological prediction uncertainty are related to the sharpness of the predictive distributions (average spread or width). The climatological prediction uncertainty part is the value of the $CRPS_{aver}$ if we only use the overall observed climatology based on the observations as the predictive distribution for each time instance, e.g. in our





case day. In this case, there will be zero reliability and resolution. The reliability part is a nonnegative quantity, with $\mathrm{Reli} = 0$ only for a perfectly reliable system, i.e., a system that is probabilistically calibrated with a uniform rank or PIT-histogram,

which will be the case for predictions based on the above-observed climatology. However, such a predictive system will have zero resolution, i.e. $\mathrm{Reso} = 0$, i.e., no sharpness, since all predictions will be based on the same (average) climatology.

We may, however, achieve lower values of $\mathrm{CRPS}_{\mathrm{aver}}$ for predictive systems with $\mathrm{Reli} - \mathrm{Reso} < 0$. The optimal case will be obtained if we use perfect deterministic point predictions. In this case, the system will still be perfectly reliable, i.e. $\mathrm{Reli} = 0$,

corresponding to a uniform rank or PIT-histogram, but in contrast to the climatological system, it will have an optimal positive resolution (sharpness) in the sense that $\mathrm{Reso} = \mathrm{CRPS}_{\mathrm{clim}}$, with a resulting value of $\mathrm{CRPS}_{\mathrm{aver}} = 0$. Generally, we will obtain values of reliability and resolution between the above two extremes, i.e. $-\mathrm{CRPS}_{\mathrm{clim}} \leq \mathrm{Reli} - \mathrm{Reso} \leq 0$, and thus $0 \leq \mathrm{CRPS}_{\mathrm{aver}} \leq \mathrm{CRPS}_{\mathrm{clim}}$. An excellent predictive system is hence characterised as one having a small (positive) value of reliability, and a high (positive) value of resolution, resulting in a small (positive) value of $\mathrm{CRPS}_{\mathrm{aver}}$.


As shown in Fig. 19, the model predictions are highly reliable for most years, with reliability values close to zero except for a few cases. Also, for most years, we have a reasonably high degree of resolution (around 2), reducing the climatological prediction uncertainty from about 4 to about 2 for the CRPS average for the predictive model at this station for the whole period 2005-2019. Note that all data in the box plots have the same unit as for concentration, i.e. $\mu\mathrm{gm}^{-3}$.

**5.3.7 CRPS box plots data**

The file name is `<station>_gam.crps_<yc>_<yd>.csv`. The data from each box plot in Sect. 5.3.6 is written to this file. Each row of the file contains the year, followed by the CRPS average, reliability, resolution and climatological prediction uncertainty parts of the CRPS average for that year. There is one line of data for each year in the cross-validation period `<yc>-<yd>`, and all numbers have the same unit as for concentration, i.e. $\mu\mathrm{gm}^{-3}$.

**5.3.8 CRPS box plots median values**

The file name is `AirGAM_gam.crps_<yc>_<yd>.csv`. The median values from each box plot in Sect. 5.3.6 are written to this file. Each row of the file contains the station name, followed by the median values of the CRPS averages and the reliability, resolution, and climatological prediction uncertainty parts of the CRPS averages over the years for the cross-validation `<yc>-<yd>`. Again, the numbers' units are the same as for concentration, i.e. $\mu\mathrm{gm}^{-3}$.


For $NO_2$ at station EE0018Ah and cross-validation years 2005-2019, we obtain the values of 2.08, 0.06, 2.01 and 4.12 as median values of CRPS average, reliability, resolution, and climatological prediction uncertainty, respectively. Thus, at the



median, the model predictions are highly reliable (reliability value close to zero), with a relatively high degree of resolution
(2.01), reducing the climatological prediction uncertainty from 4.12 to 2.08 for the predictive model at this station for the
whole period 2005-2019.

## 6 Results for a European trend study 2005-2019

Here we give some results with the AirGAM model based on data from the latest EEA trend study 2005-2019 (Solberg et al.,
2021a). Our experience from previous similar studies (Solberg et al., 2018a, 2018b, 2019) is that the model overall seems to
perform best for NO2, followed by O3 and PM10, and worst for PM2.5. Sections 6.1-6.4 below present results for the
compounds NO2, O3, PM10 and PM2.5, respectively.

Seasonal conditioning was not used for model runs, i.e. `use_season_cond=0`. Thus, only a single set of smooth relations
between the concentrations and the meteorological and time covariates were estimated and used by the model. The trend type
was set to nonlinear (`trend_type=nonlinear`), and the number of basis functions to be used by the trend term was set to
missing (`k_years=NA`), which implies that five basis functions (15 years/3) were used to represent the trend term in the
model. Introducing a basis function for the trend every three years was considered an appropriate setting in this long-term
trend study since we did not want to focus on, or model, too much of the short-term variations in the trend at individual stations
but rather focus on the more main features and more long-term variations in the trend.

The `bam` routine in the `mgcv` package was always tried before the `gam` routine (`incl_bam=1`) (incidentally, no `gam`-calls
were executed), and automatic model selection was turned on (`incl_select=1`). The AR(1) model was not invoked for
these runs (incl_ar1=0) to reduce the computational time, which means autocorrelation in the time series was not considered.
Thus, the focus is not accurately estimating individual trend curves' uncertainties. In the cross-validation, the "limit-covariates"
approach was used to obtain robust predictions (`rob_pred=limcov`), i.e. covariate values outside the training interval were
set to the nearest value in this interval before being used in the predictions. For all compounds except for $O_3$, all months in
each year were used to estimate the trend (`subyear=jan-dec`). For $O_3$, only the summer period (April-September) were
used for a summer-trend study (`subyear=apr-sep`).

### 6.1 NO₂

For NO2, there are 1485 stations in the Airbase/AQER database for 2005-2019, fulfilling the data coverage criteria for this
compound (75 % coverage for individual years and 75 % coverage of years in the period) excluding the industrial stations.
This forms the basis for the trend study for this compound. Due to the large number of stations for NO2, we refrain from
showing any individual station results here. Input data and results for all stations for this compound can be found in the model's
data repository (Walker and Solberg, 2022b-c).





However, the results for station EE0018Ah (Õismäe) are shown in Sect. 5, which describes the output results from the AirGAM model. This is a background station in an urban part of Tallinn, Estonia, with coordinates 52.41417°N and 24.64946°E, and is at 6 m a.s.l. This station was chosen to illustrate the results since it is the exact median station of all stations based on the cross-validation correlation results for NO$_2$. Thus, it is neither the best nor the worst station but may be viewed as "typical" for the results for this compound.


Figure 2 shows the primary trend results in a meteorology-adjusted trend for 2005-2019. Figure 3 shows plots of all smooth functions of meteorology and time explanatory variables based on all training data for the same period. A plot of model predictions from the cross-validation for 2019 based on data for the left-out years 2005-2018 is presented in Fig. 4. Figs. 5-8 shows plots of observed and predicted annual and monthly average concentrations with trend curves and yearly and monthly

median concentrations. Further, for this station, you may find plots of all evaluation results in the individual sub-sections of Sects. 5.2-5.3. All results are commented upon in the respective sub-sections of Sects. 5.1-5.3.

### 6.1.1 Results for all stations

Figure 20 shows a panel of three maps of stations over Europe. The maps are made for three categories of stations: (1) Background stations in rural areas (left); (2) Background stations in urban/suburban areas (middle); and (3) Traffic stations in

any area (right). In each map, we present the estimated meteorology-adjusted trend for NO2 as a relative trend for each station, i.e., a percentage change in the concentrations from 2005 to 2019 relative to the initial levels in 2005. The stations plotted in each map are the stations for which the 2005-2019 cross-validation gave a correlation between observed and model-predicted values above 0.65. This resulted in 205, 742 and 409 stations, respectively, for the three types of stations, 1356 in total.

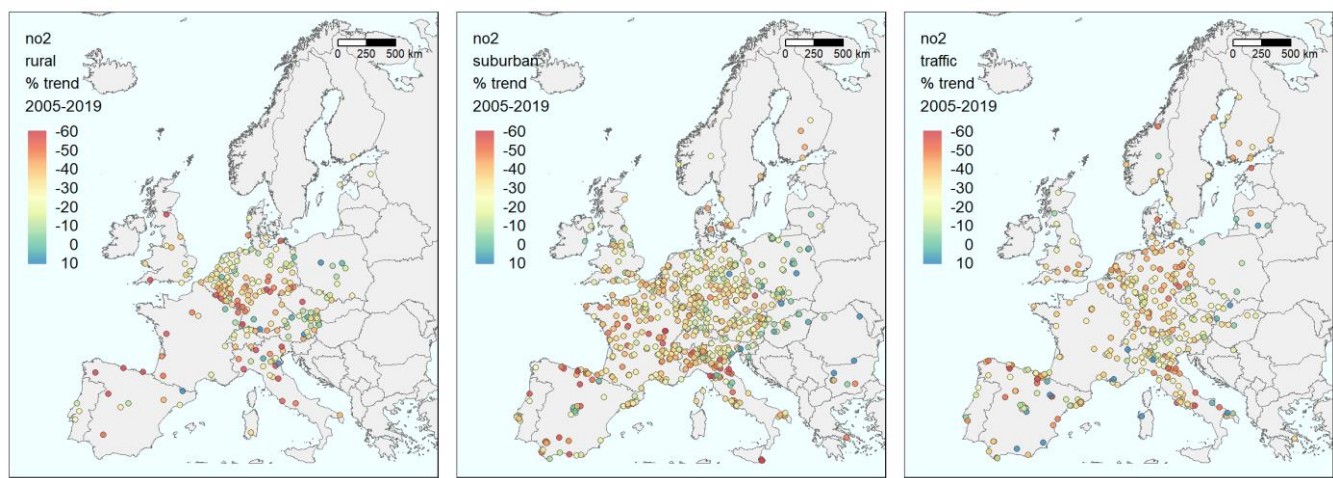

**Figure 20.** Maps of stations in Europe with the meteorology-adjusted trend for NO$_2$ as a percentage change in the concentrations from 2005 to 2019 relative to the initial level in 2005. Left: Background stations in rural areas; Middle: Background stations in urban/suburban areas; and Right: Traffic stations.





The maps indicate a weak west to east gradient with more substantial declines in the west and smaller in the east. This is particularly marked for the urban/suburban background stations (middle panel), with the densest geographical coverage. The

result is mixed with countries showing both substantial reductions and sites with no trend or even increasing levels for the traffic sites. This reflects that the roadside stations are more heterogeneous and subject to changes in the local urban environment (roads, buildings, etc.). Additionally, the $NO_2/NO_x$ ratio in tailpipe emissions will strongly influence these sites, depending on the fleet of vehicles (fraction of diesel cars) and the ambient ozone level. These issues will be reduced and smoothed out for background stations due to atmospheric mixing and the $NO_2/NO_x$ concentration ratio approaching the photo-

stationary steady state determined by solar radiation, temperature, and ozone level.

Figure 21 shows box plots of the same trends for the same three categories of stations. The left three plots show the relative trend in per cent as in Fig. 20, while the right three plots show the absolute trend as changes in concentration levels.

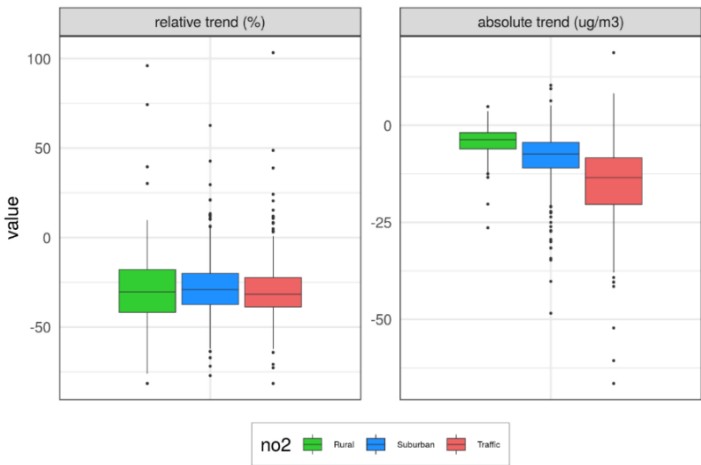

**Figure 21.** Box plots of trend values over the same three categories of stations as in the map plots in Fig. 20. The left panel shows percentage changes as in Fig. 20, while the right panel shows changes in absolute concentrations from 2005 to 2019. Units: % and $\mu gm^{-3}$.

For each station, the change in concentration level is calculated based on the physically interpretable trend curves $y_{trend}(t)$ as output from the model at the two time points $t_1$ and $t_1$ corresponding to the start and end of the trend estimation period 2005-2019, respectively, i.e. 1 Jan 2005 and 31 Dec 2019. Thus, the absolute trend is calculated as $y_{trend}(t_2) - y_{trend}(t_1)$, and the

relative trend in per cent is calculated as $100 \times (y_{trend}(t_2) - y_{trend}(t_1))/y_{trend}(t_1)$. Section 2.1.1 describes how these trend curves are calculated based on the output from the GAM model.

The results in Fig. 21 show that the $NO_2$ concentration has decreased approximately at the same rate at all stations categories during 2005-2019. Median reductions of 29 % are found for the rural and urban/suburban stations and 31 % for the traffic

stations, with corresponding decreases in concentrations of 4-13 $\mu gm^{-3}$.

Figure 22 shows box plots of some selected statistical evaluation parameters based on the cross-validation for 2005-2019. Again, the box plots are made for the same three categories of stations as in Figs. 20-21.

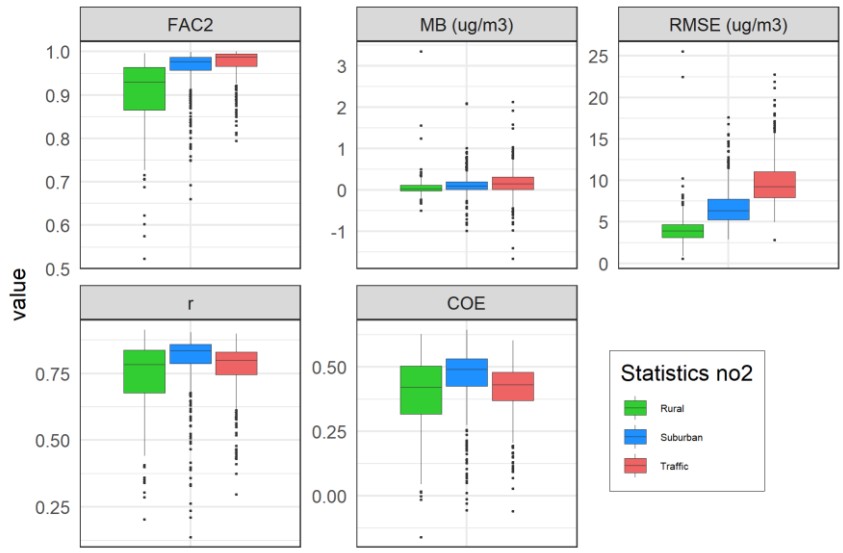

**Figure 22.** Box plots of some selected evaluation parameters from the cross-validation for 2005-2019, again for the same three categories of stations.

As seen from Fig. 22, the predictive performance of the AirGAM model for $NO_2$ concerning correlation (r) and coefficient of efficiency (COE) is somewhat better for the urban/suburban background stations than for the traffic and rural ones. However, the median values of r and COE are pretty decent for all three types. Regarding FAC2 (fraction of days with predictions within a factor two of observations) and RMSE (root mean squared error), traffic and rural sites are best, respectively, whence we have a somewhat mixed picture. Nevertheless, FAC2, r and COE are slightly poorer, i.e. lower, for rural stations, which is expected since the $NO_2$ levels at these sites depend less on the local meteorological conditions than suburban and urban sites. Also, note that the median bias is close to zero for all three types of stations, which is good.

### 6.2 $O_3$

For $O_3$, there are 1175 non-industrial stations in the Airbase/AQER database for 2005-2019, fulfilling the data coverage criteria for this compound (75 % coverage for individual years and 75 % coverage of years in the period). For this compound, the air quality data consist of maximum daily running 8-h average (MDA8) concentrations for each day. Again, no individual station results are shown here, but all data and results for this compound for all stations can be found in the model's data repository (Walker and Solberg, 2022b; Walker and Solberg, 2022d). Figures 23-25 show the same type of results as for $NO_2$.





The stations plotted in each map in Fig. 23 and as data values in Fig. 24 are the stations for which the 2005-2019 cross-validation for O₃ gave a correlation above 0.65. This resulted in 303, 594 and 44 stations, respectively, for the three types of stations, 941 in total. However, for the evaluation in Fig. 25, all 1175 stations are used, 368, 729 and 78 in each category.

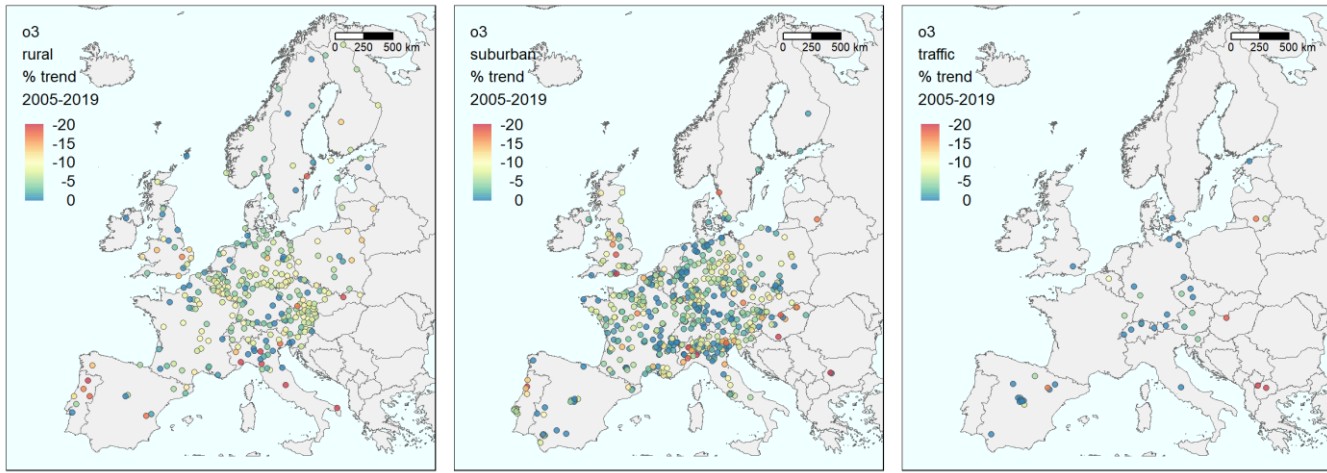

**Figure 23.** Maps of stations in Europe with the meteorology-adjusted trend for O₃ as a percentage change in the concentrations from 2005
to 2019 relative to the initial level in 2005. Left: Background stations in rural areas; Middle: Background stations in urban/suburban areas;
and Right: Traffic stations.

The geographical distribution of the ozone-summer-time trends of mean MDA8 shows no clear patterns (Fig. 23). The rural stations offer reductions (yellow-green colours) over most areas, with more substantial decreases at some stations, mainly in Portugal and Italy. The changes at urban/suburban sites are closer to zero at many locations, but several stations also show
marked reductions.

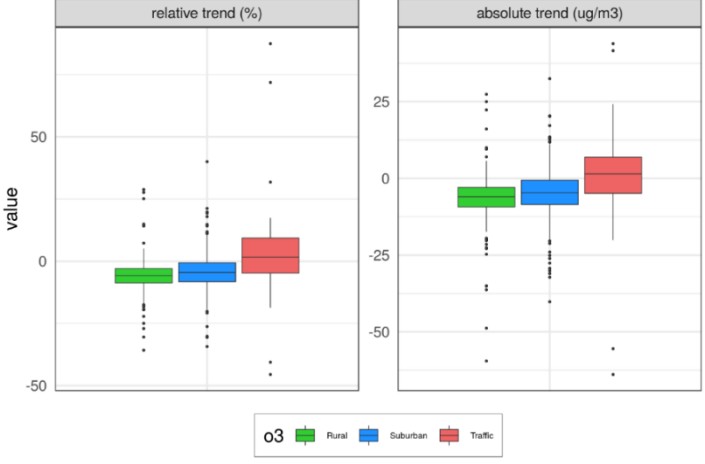

**Figure 24.** Box plots of trend values over the same three types of stations as in the map plots in Fig. 23. The left panel shows percentage changes, while the right panel shows changes in absolute concentrations from 2005 to 2019. Units: % and μgm⁻³.


As shown in Fig. 24, the calculated changes in the mean summer half-year of MDA8 during 2005-2019 are substantially
       smaller than the changes found for $NO_2$ over the same period. For the rural and urban/suburban stations, a median reduction
       of 6 % and 5 % is found, respectively, while a slight increase of 2 % is seen for the traffic sites. The corresponding changes in
       concentrations are from -6 to 2 $\mu gm^{-3}$ from 2005 to 2019.

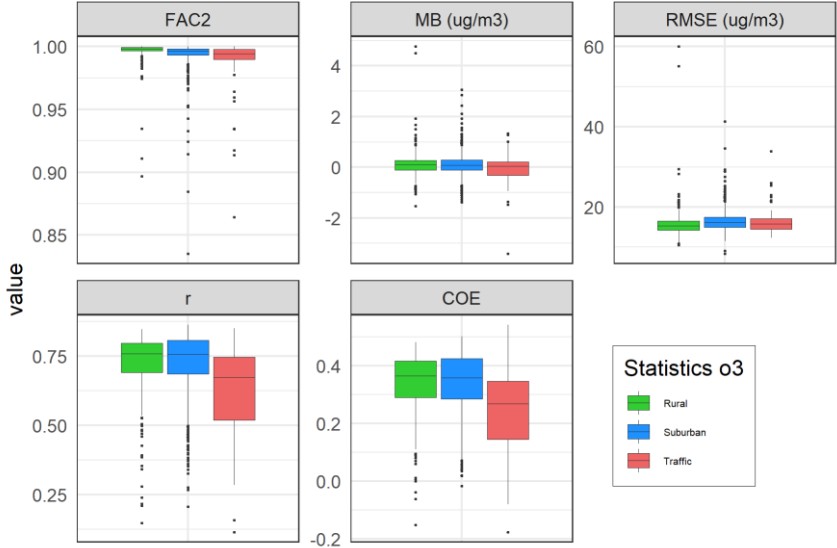

**Figure 25.** Box plots of evaluation parameters from the cross-validation for 2005-2019, again for the same three types of stations.

Otherwise, the results for ozone (Fig. 25) show the opposite compared to $NO_2$ regarding model performance; the best
performance (high FAC2, r and COE) is seen at rural stations and the poorest at traffic sites. However, the model performance
for the urban/suburban category is very close to the rural one.

**6.3 PM$_{10}$**

For PM$_{10,}$ there are 1243 non-industrial stations in the Airbase/AQER database for 2005-2019, fulfilling the data coverage
       criteria for this compound (75 % coverage for individual years and 65 % coverage of years in the period). No individual
       stations are shown, but all data and results can be found in the model's data repository (Walker and Solberg, 2022b; Walker
       and Solberg, 2022e).

Figures 26-28 show the same type of results as for the previous compounds. The stations plotted in each map in Fig. 26 and as
       data values in Fig. 27 are the stations for which the 2005-2019 cross-validation for PM$_{10}$ gave a correlation above 0.55. This
       resulted in 176, 627 and 351 stations, respectively, for the three types of stations, 1154 in total. For the evaluation in Fig. 28,
       all 1243 stations are used, 204, 658 and 381 in each category.



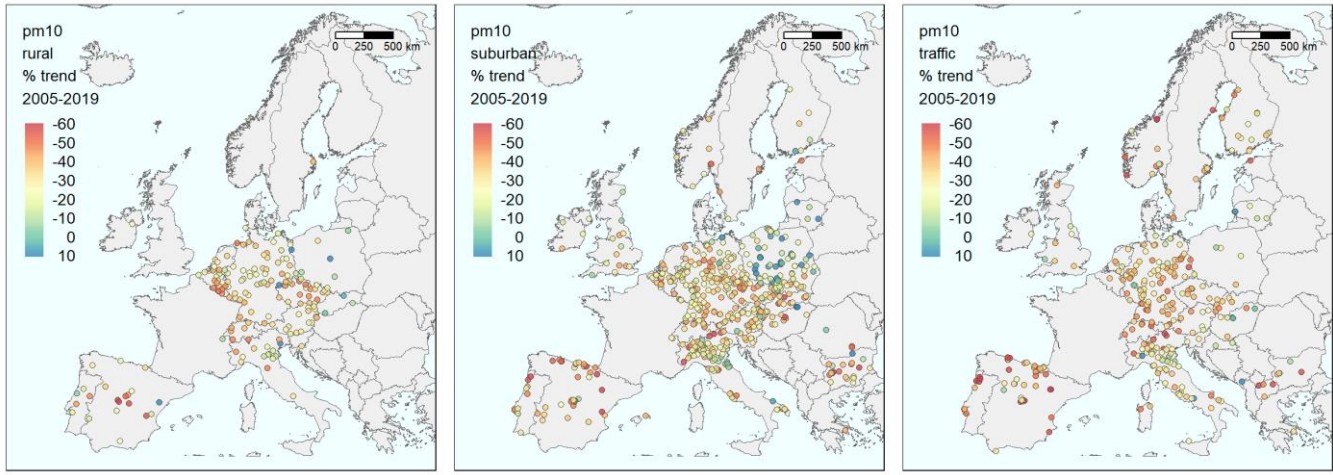

**Figure 26.** Maps of stations in Europe with the meteorology-adjusted trend for PM$_{10}$ as a percentage change in the concentrations from 2005 to 2019 relative to the initial level in 2005. Left: Background stations in rural areas; Middle: Background stations in urban/suburban areas; and Right: Traffic stations.

For PM$_{10}$, AirGAM estimates marked reductions during 2005-2019 with indications of a west to east gradient (Fig. 26) as was found for NO$_2$. Many Polish and Baltic sites have no change or even increased levels. Due to the shift from daily based
sampling to hourly in France, no trends could be calculated for sites there. Median reductions of 31 % are found for the rural and urban/suburban stations and 37 % for the traffic stations (Fig. 27), with corresponding decreases in concentrations of 7-13 µgm$^{-3}$.

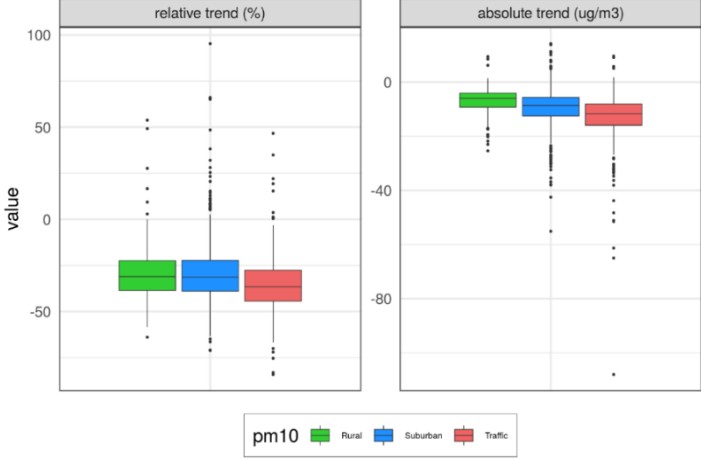

**Figure 27.** Box plots of trend values over the same three types of stations as in the map plots in Fig. 26. The left panel shows percentage
changes, while the right panel shows changes in absolute concentrations from 2005 to 2019. Units: % and µgm$^{-3}$.



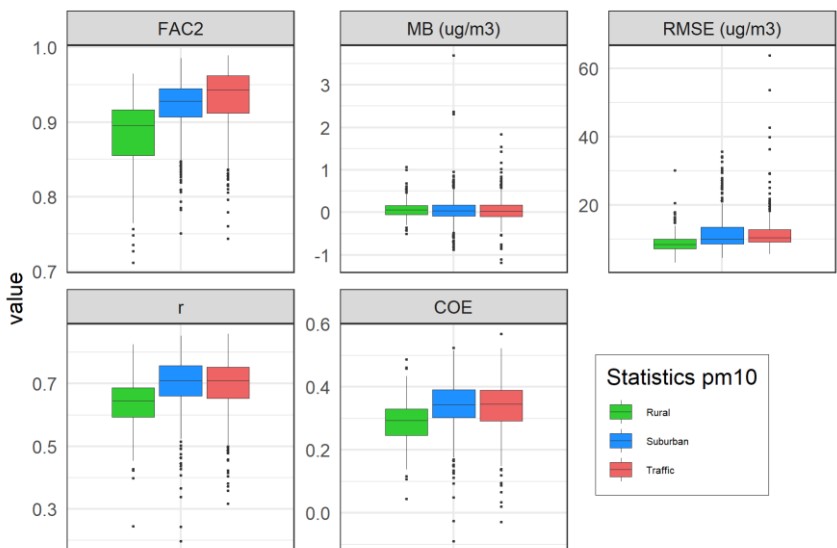

**Figure 28.** Box plots of evaluation parameters from the cross-validation for 2005-2019, again for the same three types of stations.

The AirGAM performance is best at the traffic and urban/suburban sites, with slightly poorer results for the rural ones (Fig. 28). This is as expected for $PM_{10}$ (as for $NO_2$), mainly due to these stations being exposed to more direct local emissions. In contrast, the $PM_{10}$ at rural sites is more influenced by long-range air mass transport and external processes not captured so well by the AirGAM model, such as windblown dust, forest fires, agricultural fires, etc.

**6.4 $PM_{2.5}$**

For $PM_{2.5}$, there are 354 non-industrial stations in the Airbase/AQER database for 2005-2019, fulfilling the data coverage criteria for this compound (75 % coverage for individual years and 65 % coverage of years in the period). No individual stations are shown, but all data and results can be found in the model's data repository (Walker and Solberg, 2022b; Walker and Solberg, 2022f).

Figures 29-31 show the same type of results as for the previous compounds. The stations plotted in each map in Fig. 29 and as data values in Fig. 30 are the stations for which the 2005-2019 cross-validation for $PM_{2.5}$ gave a correlation above 0.55. This resulted in 59, 186 and 80 stations, respectively, for the three types of stations, 325 in total. For the evaluation in Fig. 31, all 354 stations are used, 67, 201, and 86 in each category.



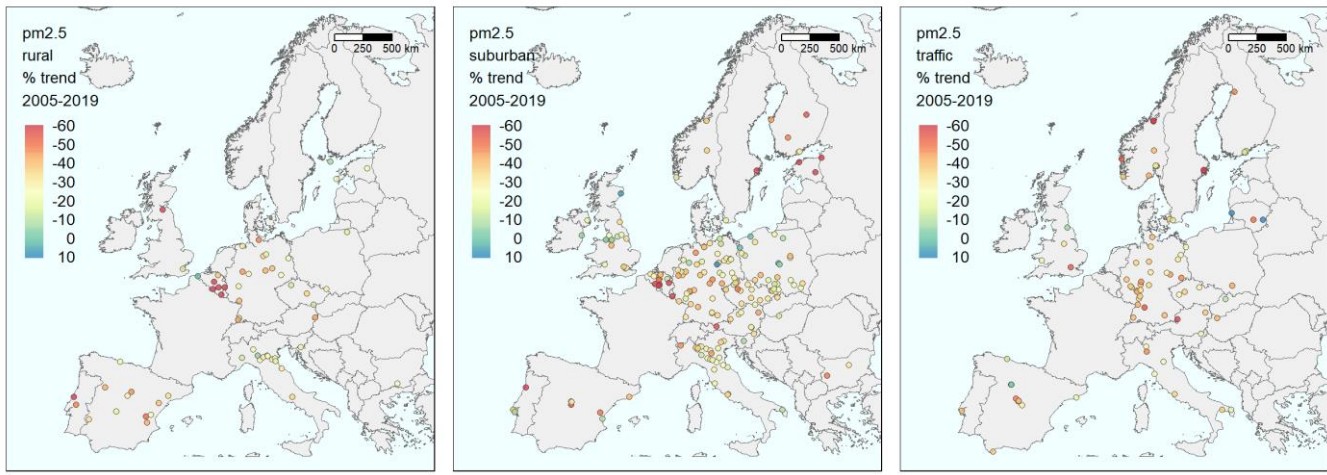

**Figure 29.** Maps of stations in Europe with the meteorology-adjusted trend for PM$_{2.5}$ as a percentage change in the concentrations from 2005 to 2019 relative to the initial level in 2005. Left: Background stations in rural areas; Middle: Background stations in urban/suburban areas; and Right: Traffic stations.

The number of stations for PM$_{2.5}$ is substantially lower than for the other compounds, and thus the interpretation of the results become more uncertain. However, the sites with sufficient monitoring length show a marked reduction in concentration level (with a few exceptions) during 2005-2019 (Fig. 29). The geographical coverage is too sparse to conclude the spatial pattern in the trends. As a median over the sites, we find relative reductions of 28 %,  31 % and 37 % at rural, urban/suburban and traffic sites, respectively (Fig. 30), with corresponding reductions in the concentration of 4-7 µgm$^{-3}$.

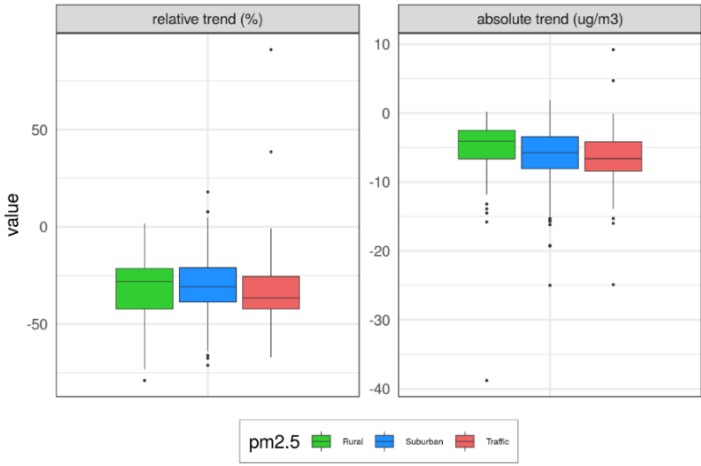

**Figure 30.** Box plots of trend values over the same three types of stations as in the map plots in Fig. 29. The left panel shows percentage changes, while the right panel shows changes in absolute concentrations from 2005 to 2019. Units: % and µgm$^{-3}$.



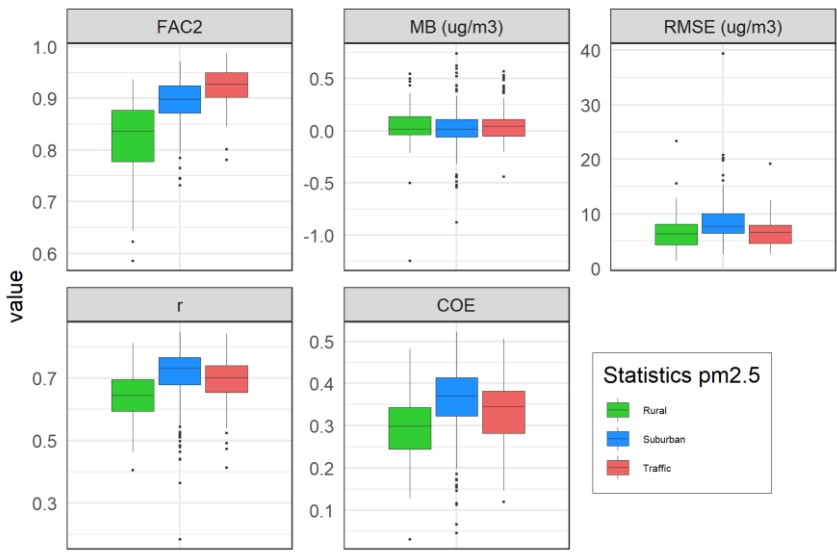

**Figure 31.** Box plots of evaluation parameters from the cross-validation for 2005-2019, again for the same three types of stations.

The AirGAM performance is best at the traffic stations concerning FAC2 but nearly the same for MB and RMSE (Fig. 31). For correlation r and COE, the urban/suburban and traffic stations are slightly better than the rural ones.

## 7 Comparison with the random forest method in `rmweather`

Another approach for discounting the effect of meteorology when estimating trends in air quality data is the random forest (RF) method (Ho, 1995; Breiman, 2001) as implemented, e.g. in the R package `rmweather` (Grange et al., 2018; Grange and Carslaw, 2019). We will briefly describe the main similarities and differences between GAM and RF and then compare them using the data from the present trend study.

Both methods attempt to solve a nonlinear regression with concentrations as response and a given set of meteorological and time variables as covariates. However, the nature of their solution methods is quite different. In GAM, we estimate a set of nonlinear relations between the response and the covariates by regularisation, i.e. by maximising a penalised likelihood. This produces smooth and non-wiggly estimated relations between the response and each covariate resulting in a model that avoids overfitting and generalises well to new data, not in the training set.

The RF approach uses decision trees as its main building block for relating the response to the covariates. Several datasets are then created randomly by bootstrapping, i.e. random sampling of new data with replacement from the available data, including randomly selecting various subsets of the available covariates. Then decision trees are fitted individually to these data. Finally, an ensemble of such fitted trees defines the nonlinear relations and predictions. The latter are produced as mean values over



the predictions from individual trees in the forest. This results in a model that avoids overfitting and generalises well to new
data. Although they cannot be classified as robust methods per se, GAM and RF are not very sensitive to outliers. They can
also handle missing data well.

Both methods result in interpretable models. We can quickly inspect the estimated nonlinear relations between the response
and each covariate to see if an association makes physical sense. In GAM, for this purpose, we use the set of estimated smooth
functions, including the smooth function for the trend, while in RF, we use a similar set of so-called estimated partial
dependencies. In the latter method, a trend is estimated by calculating meteorologically normalised concentrations over time,
i.e. mean concentrations predicted from the RF model using "average meteorology". In GAM, uncertainties in the smooth
functions are output directly as a bi-product of the estimation. For RF, however, bootstrapping, using randomly sampled data
and repeated estimations, must generally be used to estimate uncertainties in the partial dependencies and the meteorologically
normalised concentrations.

A nice feature of the RF approach is that concurvity or collinearity between the covariates is handled efficiently and is thus
not an essential issue for these models, while this can be detrimental for GAM models and needs to be checked. However, our
present study's experience shows this to be a pretty minor issue, with only a few cases of potential problems, despite many
stations. RF also has model selection built-in since the method will ignore a variable not contributing to predicting a response.
GAM also has model selection built-in when we turn on the `select=TRUE` option in the calls to the `bam` and `gam` routines.

Also, more assumptions are built into a GAM model relative to an RF model. E.g. in GAM, we need to assume a specific
probability distribution for the response given the covariates, and we need to consider transformations of the former. In
contrast, no distributions or transformations must be specified in an RF. Further, GAM assumes a smooth and continuous
underlying relation between the response and each covariate. In our case, smooth and continuous relations are often found
between air pollution and related meteorological and time variables. However, such smooth and continuous relations are not
considered in an RF approach, and they can be non-smooth and discontinuous. Even if perhaps not directly discontinuous,
more abrupt changes in the trend, e.g. may happen if policy changes or mitigation measures lead to changes in emissions and
subsequent concentration levels at a station over a relatively short period. Such sharp transitions will typically be more
smoothed out in a GAM model unless we use a higher number of basis functions around the time of the events.

A trend analysis based on hourly values of $NO_2$ for the traffic station GB0682Ah – London Marylebone Road – for 1997-2016
was conducted in Grange and Carslaw (2019) using the RF method in `rmweather`. Their paper focuses on the impact on the
trend due to various interventions imposed on road traffic in London during this period. These interventions aim to reduce
primary $NO_2$ emissions from vehicles, leading to lower $NO_2$ concentrations. To compare AirGAM with RF, we have applied
a similar trend analysis here for this station but 2005-2019 and using daily mean values of $NO_2$ as input rather than hourly





data. In our study, we use the same meteorological and time variables in RF as in AirGAM, i.e., we use meteorology from
ECMWF ERA-5 for this station rather than data from Heathrow Airport, which was used in their paper. The meteorological

covariates are the same in both studies, except for planetary boundary layer height and cloud cover, which is used here for
both models, and atmospheric pressure, which is not used. Otherwise, we run with the same hyperparameters in RF as in their
paper, using 300 trees in the forest, a minimal node size of five, and the default number of variables split at each node, three
in our case. The seed number in the calls to the routines in `rmweather` was set to 1234. A default of 300 predictions was
used to produce the meteorologically normalised concentrations.


As for AirGAM, we run with the same set-up as for the other runs in this paper, but we introduce a somewhat more agile GAM
model by increasing the number of basis functions for the trend from the default five to ten. This is the smallest number of
basis functions considered sufficient according to the `gam.check`-routine (k-1 - edf > 0.5) – so introducing just the right
amount of model complexity for the trend in GAM for this more detailed trend analysis. We also consider auto-correlation in

the time-series by including an AR(1) model for the model residuals using the option `incl_ar1=1` in AirGAM.

Figure 32 shows the meteorologically normalised trend from the RF model (blue curve) as monthly averages and the
meteorology-adjusted smooth trend from AirGAM (dark green curve) for NO$_2$ at London Marylebone Road for 2005-2019. In
the figure, the dark green dashed curves forms a 95% confidence region for the trend from AirGAM.

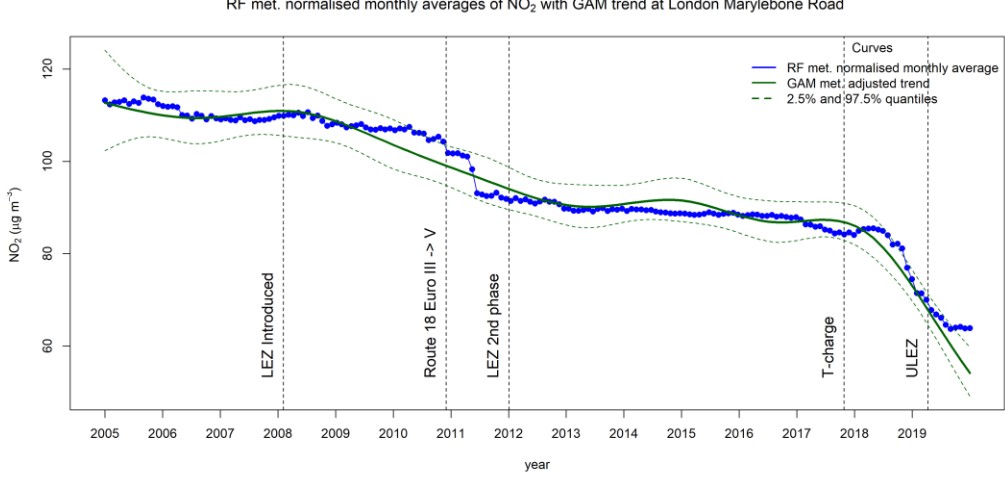


**Figure 32.** Meteorologically normalised trend (monthly averages) from the RF model (blue curve) together with a meteorology-adjusted
smooth trend (dark green curve) from AirGAM with a 95% confidence region (dashed curves) for NO$_2$ at London Marylebone Road for
2005-2019. Unit: $\mu g m^{-3}$.

The vertical dashed lines in the figure show air quality interventions in London during this period, the first two of which were

highlighted in Grange and Carslaw (2019) as identified breakpoints from a time-series breakpoint detection analysis conducted
there. These are associated with introducing the Low Emission Zone (LEZ) in London on 4 April 2008 and the change from





Euro III to Euro V type of vehicles on Route 18 at the end of 2010. The following two interventions (2nd phase of LEZ on 3 January 2012 and the introduction of the toxicity surcharge (T-charge) on 23 October 2017 were also considered in their paper but not identified as actual breakpoints for the trends in their analysis. The last intervention shown in the figure introduces the

Ultra-Low Emission Zone (ULEZ) in London on 4 April 2019. For a more thorough description of these interventions, see Grange and Carslaw (2019). A detailed description of the history and development of various congestion charges introduced in London to curb the levels of air pollution from the mid-90s to the present is given in Wikipedia (2022).

As shown in Fig. 32, the shape of the trend curve from the GAM model resembles the trend point values from the RF model

with some noticeable differences. The GAM curve is smooth and not too wiggly by construction, falling gently in several phases with more flat in-between parts before decreasing sharply at the end from around the introduction of the T-charge in 2017 and towards 2020. The concentration level is reduced from 113 µgm⁻³ in 2005 to 54 µgm⁻³ in 2020, a total reduction of 59 µgm⁻³. The trend values from RF are more variable, presumably due to the more adept nature of this method. Here the level is reduced from 113 µgm⁻³ in 2005 to 64 µgm⁻³ in 2020, a total of 49 µgm⁻³, ten µgm⁻³ less than for GAM, mainly due to the

differences in the trends at the very end of the period. Also noticeable is the sharp decrease in the RF trend around the time of the Route 18 bus fuel changes late in 2010 to the middle of 2011, which the GAM model does not reproduce. Instead, GAM estimates a smooth and gentle reduction in the concentrations over a much more extended period from just after the introduction of the LEZ in 2008 to the middle of 2013.

Note that despite differences in the shapes of these trend curves, the RF trend is well contained within the 95% confidence region for the GAM trend, except for five months in 2010-2011 and six months at the end of the period, a total of 11 months. If the RF point values were the true trend, we would expect around nine monthly values (5 % of 180 months) outside this interval. However, they are also estimated and not identical to the true trend. It is challenging to state the most realistic trend – the smooth and non-wiggly trend from GAM or the more variable and detailed trend from RF. The sharp declines in the RF

trend around the introduction of the Route 18 bus fuel changes may well indicate that the RF trend in this period is the most realistic. On the other hand, the gentle reduction in the GAM trend from the LEZ introduction in 2008 towards 2011, a period where the RF trend is more constant, is also interesting and may point to cuts in NO₂ emissions from traffic affecting the station in this period. Thus, it may be beneficial to use both methods in more detailed trend analyses to obtain a more diverse picture of and insight into the possible nature of the actual trend.


Comparing the trends produced from AirGAM and the RF method in `rmweather` for other stations and compounds (not shown here) gives in the majority of cases the same picture as above. The GAM approach in AirGAM produces trend curves that are smooth and non-wiggly, while the RF approach tends to create more variable trends with more details. All in all, however, we found the trends to be similar in most cases. It would be interesting to study how well the two methods estimate





trends using controlled experiments with simulated but realistic data to know the underlying trend. We hope to pursue such a
        study in a forthcoming paper.

        Table 2 shows for the various compounds the prediction accuracy of the GAM models in AirGAM versus the RF method in
        `rmweather` from the cross-validation for 2005-2019. Here, the evaluation parameters, i.e. prediction performance metrics,
are the same as those used in Sect. 6. Each metric in the table is calculated as a mean over the $n$ individual stations for each
        compound. The corresponding number in parentheses is an estimated standard deviation of this mean value obtained from
        bootstrapping using 5000 replications of the individual values in each case.

**Table 2.** Prediction accuracy from the cross-validation for 2005-2019 of AirGAM vs RF from `rmweather`. The prediction performance
metrics are the same as those used in Sect. 6. Numbers in bold mark the best performing model for each compound and metric.

| Compound | Model | n | FAC2 (sd) | MB (sd) | RMSE (sd) | r (sd) | COE (sd) |
|---|---|---|---|---|---|---|---|
| NO$_2$ | GAM | 1485 | **0.957 (0.001)** | **0.121 (0.007)** | 7.223 (0.080) | 0.787 (0.003) | 0.439 (0.003) |
| | RF | 1485 | 0.954 (0.001) | 0.279 (0.007) | **7.031 (0.075)** | **0.797 (0.002)** | **0.444 (0.003)** |
| O$_3$ | GAM | 1175 | 0.995 ($3\cdot10^{-4}$) | 0.094 (0.014) | 16.301 (0.091) | 0.719 (0.003) | 0.335 (0.003) |
| | RF | 1175 | 0.995 ($3\cdot10^{-4}$) | **0.023 (0.012)** | **16.113 (0.086)** | **0.726 (0.003)** | **0.347 (0.003)** |
| PM$_{10}$ | GAM | 1243 | **0.918 (0.001)** | **0.059 (0.009)** | **11.195 (0.136)** | **0.687 (0.002)** | **0.330 (0.002)** |
| | RF | 1243 | 0.909 (0.001) | 0.576 (0.016) | 11.274 (0.138) | 0.683 (0.002) | 0.314 (0.002) |
| PM$_{2.5}$ | GAM | 354 | **0.884 (0.003)** | **0.035 (0.011)** | 7.751 (0.187) | 0.691 (0.005) | **0.341 (0.005)** |
| | RF | 354 | 0.874 (0.003) | 0.404 (0.017) | 7.759 (0.193) | 0.690 (0.005) | 0.328 (0.005) |

        As shown from Table 2, the two methods are pretty similar in predictive performance concerning the various evaluation
        parameters shown. For some compounds and performance metrics, GAM is best; for others, RF is best, but the difference
        between the two is slight, except perhaps for the mean bias (MB) where GAM is clearly better than RF for all compounds
except for O$_3$, where RF is slightly better. Bootstrapping the differences in the metrics between the models results in the bold
        numbers in the table, where one model is statistically significantly better than the other at the 1 % level. In terms of the
        concentration level independent metrics r and COE, both models seem to perform best for NO$_2$, followed by O$_3$ and then PM,
        although GAM is fairly good regarding COE for PM$_{2.5}$. Overall, both methods perform pretty well for all metrics shown, but
        GAM seems to have an edge in PM while the opposite is true for NO$_2$ and O$_3$.






Interaction between covariates means that a covariate's influence on the response depends on the level of one or more of the other covariates. RF models, such as those produced by `rmweather`, can potentially include complex interactions between the covariates. Strictly additive GAM models used by AirGAM do not have any interactions between covariates. Since they are very similar in predictive performance for all compounds in this comprehensive trend study, there seem to be few or no
interactions between the covariates that need to be modelled to get good predictive performance, at least for these data. Thus, a purely additive model with no interactions appears to be sufficient.

## 8 AirGAM as a tool for data quality investigations

A spin-off from the AirGAM model is the ability to detect measurement data of dubious quality. This is easily explained as AirGAM is based on finding long-term systematic relationships between reported levels of air pollutants and meteorological
and temporal data, on the other hand. When this approach fails, it is either due to the station being dominated by long-range transport of air pollutants (whereas AirGAM relies on local meteorological data) or a result of artefacts in the monitoring data such as significant shifts in the concentration level or the seasonal cycle or other kinds of spurious effects. As described in previous reports (EEA, 2017; EEA, 2019), screening the AirGAM results on the lowest correlation coefficients (and high values for NMGE, the normalised mean gross error) has proven a valuable tool for detecting errors in the measurement data.
EEA (2017) and EEA (2019) give various examples of time series of dubious quality identified by screening the AirGAM results. The examples include time series with a substantial offset in specific years and ozone data given in a faulty unit (ppb vs $\mu gm^{-3}$) during parts of the period. Although such errors could have been identified with basic statistical tools, other types of artefacts would have been harder to detect. This includes time series of $PM_{10}$ at certain stations and years that turned out to be displaced by one day. AirGAM predicted the daily concentrations fairly accurately for these time series, whereas a
systematic shift of one day was seen compared to the observational time series. Further investigations confirmed that the timestamp of the measurements was indeed wrong. This type of error in the monitoring data would have been tough to detect by more basic statistical methods.

## 9 Summary and conclusions

This paper presents the AirGAM model – an air quality trend and prediction model developed at NILU in cooperation with
EEA over the years 2017-2021. The model is based on solving a nonlinear regression using generalized additive modelling (GAM) of daily observed concentrations at individual air quality monitoring stations with corresponding meteorological and time related explanatory variables. It has been developed primarily for $NO_2$, $O_3$, $PM_{10}$ and $PM_{2.5}$. Since the concentrations are conditioned on local meteorology in the regression, the trend estimated by the model may be viewed as a meteorology-adjusted trend – i.e. a trend in concentrations discounting for the effects of time variations and trends in the meteorological data.






The model can also produce unadjusted trends, i.e. trends using the same regression set-up but only including the time variables. These can then be compared with the meteorology-adjusted trends to see the effect of the meteorological adjustment. Unadjusted trends show changes in actual concentrations with time. In contrast, meteorology-adjusted trends show changes in concentrations mainly due to changes in emissions or physio-chemical processes not induced by meteorology.


The meteorological and time covariates used in AirGAM have been carefully selected on physical grounds for each pollutant as part of the model development. Generally, they were statistically significant both in our earlier studies and in our present study involving EEA Airbase/AQER stations in Europe for 2005-2019. Thus, we believe they are reasonable explanatory variables for the concentration variations. However, performing model selection is vital as good practice in statistical

regression with many covariates. Due to the large number of stations to be handled, more traditional model selection techniques of including or excluding individual covariates in a step-wise fashion was found to be intractable to implement in the model. Instead, a form of automatic model selection is introduced via extra penalisations in the GAM solver routines, forcing any non-essential or superfluous covariate to be pushed towards a zero flat function and thus "removed" from the regression. Based on this in our present study, most covariates were found to be significant at the 5 % level, with only a few found non-significant

at some stations – mostly cloud cover for $NO_2$ and precipitation for PM.

A concurvity analysis performed in the present study show all covariates to be relatively independent of each other for all compounds, with concurvity values of type `estimate` for the most part below 0.4. Higher values than this occurred only in 0.55 %, 0.09 %, 0.31 % and 0.53 % of the cases (stations and covariates), for $NO_2$, $O_3$, $PM_{10}$ and $PM_{2.5}$, respectively. For $NO_2$

and PM, all values were below 0.5, while for $O_3$, only three were in the interval [0.5, 0.6]. This generally indicates good statistical identifiability of the model variables, implying a good estimation of the smooth nonlinear relations, including the trend. In AirGAM, as a default, a basis function is introduced in the trend term every three years with data, which typically estimates the trend's main features and long-term properties quite well in most cases. However, the user may choose a higher number of basis functions for the trend if it is essential to capture more details and short-term variations.


Our present trend analysis in Europe for 2005-2019 show that the $NO_2$ concentration has decreased approximately at the same rate at all station categories during this period. Median reductions of 29 % are found for rural and urban/suburban stations and 31 % for traffic stations, with corresponding decreases in concentration levels of 4-13 $\mu gm^{-3}$. For $O_3$ at the rural and urban/suburban stations, median reductions of 6 % and 5 % are found, respectively, while a slight increase of 2 % is seen for

the traffic sites. Corresponding changes in concentrations are from -6 to 2 $\mu gm^{-3}$. For $PM_{10}$, median reductions of 31 % are found for the rural and urban/suburban stations and 37 % for the traffic stations, with corresponding decreases in concentrations of 7-13 $\mu gm^{-3}$. And finally, for $PM_{2.5}$, we find median reductions of 28 %, 31 % and 37 % at rural, urban/suburban and traffic sites, respectively, with corresponding decreases in concentrations of 4-7 $\mu gm^{-3}$. Thus, these are our estimated changes in concentration levels due to changes in emissions or physio-chemical processes, not due to meteorology during this period.




Cross-validation at the stations in Europe for 2005-2019 shows that the model works well and can predict concentrations with reasonably good accuracy in this period for most stations, with correlations ranging from 0.69 for PM to 0.79 for $NO_2$, and RMSE ranging from 7.2 µgm$^{-3}$ for $NO_2$ to 16.3 µgm$^{-3}$ for $O_3$. Comparison with other approaches for estimating meteorology-adjusted trends based on nonlinear regression, such as the random forest (RF) method from the `rmweather` package, show

our GAM to be on par with RF regarding prediction accuracy, e.g. in terms of RMSE and correlation while having somewhat better results regarding mean bias. Thus, despite their very different nature of construction, both methods produce models that avoid overfitting and generalises quite well towards new data, not in the training set.

Interaction means that a covariate's influence on the response depends on the level of one or more of the other covariates. RF

models, such as those produced by `rmweather`, can potentially include complex interactions between the covariates. Strictly additive GAM models implemented in AirGAM do not have any interactions between covariates. Since these two types of models have very similar predictive performance for all compounds in this comprehensive trend study, this indicates that there seem to be few or no interactions between the covariates that need to be modelled to get good predictive performance, at least for these data. Thus, purely additive GAM models with no interactions appear to be sufficient.


In AirGAM, we assume a smooth and continuous underlying relation between the response concentration and each of the meteorological and time variables of the model – including that for the trend term. In air pollution modelling, such smooth relations are natural to assume since they are, for the most part, also smooth in reality. However, more abrupt changes or steep trends in concentrations at a station may happen if, e.g. policy changes or mitigation measures are introduced, leading to

emission changes influencing the station over a relatively short period. In such cases, RF methods and similar tree-based techniques could be more appropriate since they generally allow the relationship between the concentrations and the total time variable to be non-smooth and even discontinuous. Such sharp changes in concentration levels will typically be more smoothed out in a GAM model unless we introduce more basis functions or adaptive basis functions around the time of such events. However, it is less of a problem if our focus is on the trend's main features and long-term properties.

**10 Code and data availability**

The current version of the AirGAM model is available on Zenodo (Walker, 2022a) under the GPL-2 licence. The exact version of the model (2022r1) used to produce the results used in this paper is archived on Zenodo (Walker, 2022b), as are input data and scripts to run the model and produce the plots for the results presented in this paper (Walker and Solberg, 2022a-b). The results for all individual stations and compounds can also be found on Zenodo (Walker and Solberg, 2022c-f).



## Appendix A: Installing R, R packages and the AirGAM model


Here we describe installing R, the necessary R packages and the AirGAM model for Windows (Sects. A.1-A.3) and Linux (Sects. A.4-A.6).

### A.1 System requirements for Windows

Currently, R can be used on various versions of Windows (R Core Team, 2021). We recommend using Windows 10 or 11,
preferably the 64-bit version and a computer with at least 16 GB of RAM. There is no specific requirement regarding disk space except that it should be sufficient to store R and its packages and all files for the model. The latter depends on the number of stations, the number of years defined for the trend calculation and cross-validation, and the selected amount of output.

### A.2 Installing R and R packages for Windows

R for Windows can be downloaded from http://www.r-project.org. We recommend using the latest version and installing it in
the default location `C:\Program Files\R\R-<ver>\`, where `<ver>` is the version number. The newest version as of this writing is 4.1.2. In addition to R, the AirGAM model relies upon the following R packages:

- `mgcv`
- `openair`
- `sandwich`

We recommend installing the latest version of these packages. The installed version of R contains both 32-bit and 64-bit versions installed on a 64-bit version of Windows. The model may be run in either version of R. Usually, we run it using the 32-bit version. This is usually as fast as running it using the 64-bit version. Also, there is usually no need for the extra address space of R in the 64-bit version.

### A.3 Installing AirGAM for Windows

The latest version of the model can be downloaded from Zenodo (Walker, 2022a). The exact version used to produce the results in this paper can be downloaded from the same site (Walker, 2022b). The model is installed simply by copying the AirGAM R script to the same directory as the run scripts. The latter can be downloaded from Walker and Solberg (2022a) together with input $NO_2$ data for station EE0018Ah and results for this and median stations for the other compounds.

### A.4 System requirements for Linux

R can be used on various versions of Linux (R Core Team, 2021). We have good experience running it on Ubuntu and Red Hat (CentOS). Again, the computer should have at least 16 GB of RAM, and there is no specific requirement regarding disk space except that it should be sufficient to store R and its packages and the data files for the model.





### A.5 Installing R and R packages for Linux

R for Linux can be downloaded from http://www.r-project.org. Again, we recommend using the latest version and installing
it in the default location for the specific version of Linux being used. It may be necessary to compile from source code as part
of the installation if you want the latest version, depending on what is available as precompiled. The AirGAM model relies
upon the same R packages for Linux as for Windows; see Sect. A.2. If the Linux version is 64-bit, it may be necessary and
preferable to run the model using the 64-bit version of R.

### A.6 Installing AirGAM for Linux

Installing the model on Linux is identical to installing it on Windows (see Sect. A.3). The same model R script, options file
and input data files are used for both systems. However, the script files used to run the model differ as described in Sect. 3.

### Appendix B: List of warning and error codes and messages

### B.1 Warning codes and messages

The following is a list of possible warning codes from the program with a short explanation. Together with the station name
acronym, these codes, the data's current date (year, month, day), and some explanatory text are written to the program log-file
`AirGAM_log.txt`.

**Warning #1a**: Insufficient data coverage for years.

This warning is issued if there are insufficient data for a station relative to the data coverage criterion `perc2` for years. It is
given early as part of building the global list of stations. The station will not be added to the global list and will not be processed.

**Warning #1b**: Insufficient data coverage for years.

This warning is also issued if there are insufficient data for a station relative to the data coverage criterion `perc2` for years.
Still, it will be given only after reading all data for the station and considering the `perc1` data coverage criterion for each
year. The station will not be processed.

**Warning #1c**: Negative concentration detected and replaced by the value 0.1.

This warning is issued if the station data contain zero or negative concentrations for the compounds $NO_2$, $PM_{10}$ and $PM_{2.5}$,
which are log-transformed by the model. Such concentrations are replaced by the value 0.1.




**Warning #2a**: Covariate not significant.

This warning is issued from the trend estimation if a covariate gets a p-value higher than 0.05. The smooth function associated
with the covariate is then considered not significantly different from a flat zero function at the 5 % level.

**Warning #2b**: Number of basis functions perhaps too low.

This warning is issued from the trend estimation based on the check routine `k.check` in `mgcv`. Suppose for a given covariate,
the p-value from the `k.check` output table is smaller than 0.05, and the corresponding k-index is smaller than one. In that
case, a warning will be issued if the value of $k' - edf$ from the table is smaller than 0.5, where $k'$ is the theoretical and `edf`
the empirical number of degrees of freedom for this covariate, respectively. This indicates that the number of basis functions
for the covariate might be too low. Here $k' = k - 1$, where $k$ is the number of basis functions defined for the covariate on input.
Also, note that the output from the routines `k.check` and `gam.check` (producing the output to the *gam.check* files) are
usually different since the two routines use different seed values.


**Warning #2c**: Covariate might be dependent.

This warning is issued from the trend estimation based on concurvity values of the `estimate` type from the `concurvity`
routine in `mgcv`. It is triggered if a concurvity value of a given covariate is higher than 0.4. This is then taken to indicate that
the covariate might depend on one or more of the other covariates, either linearly or nonlinearly.


**Warning #3a**: A covariate value used in prediction is outside the interval of values from the training.

This warning is issued from the cross-validation part. It is only triggered when the control variable is `rob_pred=limcov`
or `rob_pred=outmiss`. In the former case, the covariate value is adjusted to the training set's nearest value before
predicting. In the second case, the covariate value is not altered, but warning #3b is also issued with potential action as
described below. See Sect. 3.3.15 for more description of the `rob_pred` variable.

**Warning #3b**: This warning is issued from the cross-validation part. A covariate value used in prediction is outside the interval
of covariate values from the training. In addition, the predicted concentration value is outside the whisker fences of a
generalized box plot of concentration values from the training data. In this case, the predicted concentration is considered a
potential outlier. The covariate value is not adjusted, but the predicted concentration is set to the missing value (NA).

**B.2 Error codes and messages**

The following is a list of possible error codes from the program with a short explanation. Together with the station name
acronym, these codes, the data's current date (year, month, day), and some explanatory text are written to the program log-file
`AirGAM_log.txt`.






**Error #1a**: Error when reading the station list file for a given year.

Something went wrong when reading the station list file for a given year. The user should inspect this file to find the reason for the error.

**Error #1b**: A necessary column is not in the station list file for a given year.

The program checks if all required columns with names `name`, `lon`, `lat`, `z`, `type`, `area`, and `country` are in the station data frame as read from the station list file. If this is not the case, errors are issued stating which columns are missing.

**Error #1c**: Error when reading the station data file for a given year.

Something went wrong when reading the station data file for a given year. The user should inspect this file to find the reason for the error.

**Error #1d**: A necessary date column, concentration column, or meteorological covariate column is not in the station data file.

The program checks to see if all required columns with data are present in the station data file. If this is not the case, errors are

issued stating which columns are missing.

**Error #1e**: No data were found for a given station.

No data were found when reading the station data for a given station. The user should inspect the data directory to find the reason.


**Error #2a**: Error when trying to run the `bam` routine in the `mgcv` R package.

Something went wrong when calling the `bam` routine for solving the GAM model equations as part of the trend calculations. The user should inspect the station data for the whole period of the trend estimation to find the reason for the error. The same error code may also be issued when running the `gamm` routine in the trend calculations (`incl_ar1=1`). The text message

makes it clear which routine is involved.

**Error #2b**: Error when trying to run the `gam` routine in the `mgcv` R package.

Something went wrong when calling the `gam` routine for solving the GAM model equations as part of the trend calculations. The user should inspect the station data for the whole period of the trend estimation to find the reason for the error.






**Error #2c**: Error when trying to run the `bam` routine in the `mgcv` R package during cross-validation.

Something went wrong when calling the `bam` routine for solving the GAM model equations as part of the cross-validation.

The user should inspect the station data for the whole period of the trend estimation minus the current year for cross-validation to find the reason for the error. The same error code may also be issued when running the `gamm` routine as part of the cross-validation calculations (`incl_ar1=1`). The text message makes it clear which routine is involved.

**Error #2d**: Error when trying to run the `gam` routine in the `mgcv` R package during cross-validation.

Something went wrong when calling the `bam` routine for solving the GAM model equations as part of the cross-validation. The user should inspect the station data for the whole period of the trend estimation minus the current year for cross-validation to find the reason for the error.

**Appendix C: AirGAM model Q & A**

Q: Can I run for other compounds than $NO_2$, $O_3$, $PM_{10}$, and $PM_2$?

A: It should be possible to run for $NO_x = NO_2 + NO$ and $O_x = O_3 + NO_2$, in the same way as for $NO_2$ and $O_3$, respectively, since these behave somewhat similar to these compounds. Recently we applied the model for VOC (n-butane) at two rural background stations in Europe with encouraging results – showing good agreement between observed and predicted concentrations (Solberg et al., 2021a, Sect. 3.2.1). Including other compounds should be possible but may need additional work to investigate which meteorological variables to use.


Q: Why are not hours being used in the system?

A: We think it is most sensible to stick to daily values since one of the programs aims is to estimate trends over a more extended period, several years. Diurnal variations of air quality and meteorology are not essential to consider or model in this respect.

Q: How can I run only for a subset of stations, e.g. just for the Danish sites?

A: You may edit the static station files (Sect. 3.4) to include only the stations you wish to run for, e.g. just stations with DK in the station name. Note that you need to edit these files for each year. This may be a little tedious when there are many years and files, so we plan to develop a more automatic procedure based on filtering in later versions of the model.

Q: In the station input files, do the columns need to be in a specific order?

A: No, the files are read into R as data frames with headers, so the column order is irrelevant. However, the header names must be as specified in Sect. 3.4.





Q: Can there be missing data for air quality or meteorology in the station input files?

A: Yes, you can have any number of missing data as long as you have enough complete cases, i.e., days with a complete set of data, to comply with the data coverage criteria. You do not need meteorology to run the model for unadjusted trends.

Q: Do I need to have data for all years in the period selected for trend estimation?

A: No, the model tolerates missing years in the input. However, you must have enough years to comply with the data coverage percentages in the AirGAM options file. E.g. if you use 75 % as the data coverage for years (perc2), you need to have at least eight years with data if running for ten years.

Q: Can I use a different missing data value on input, e.g. -9900 or another unique number?

A: No, not currently, but this may be introduced later. For now, you can only use the two- and three-letter combinations NA (missing value in R) and NaN ("not a number" in R).

Q: What happens if my data contains zero or negative concentrations?

A: For $O_3$, nothing happens; the data are used as-is. For the other compounds due to the logarithmic transformation, zero or

negative concentrations are replaced by 0.1. A warning is written to the AirGAM_log.txt file with station name, year, month, day and the initial negative concentration for each such case.

*Author contributions.* Conceptualisation: S.E.W., S.S.; Data curation: S.S., S.E.W.; Formal analysis: S.E.W., S.S., P.S.; Investigation: S.S., S.E.W.; Methodology: S.E.W., S.S.; Resources: S.S.; S.E.W.; Software: S.E.W., S.S., P.S.; Validation: S.E.W., S.S., P.S.; Visualisation: S.E.W., S.S., P.S.; Writing—original draft preparation: S.E.W., S.S.; Writing—review and

editing: S.E.W., S.S., P.S., C.G. All authors have read and agreed to the published version of the manuscript.

*Competing interests.* The authors declare that they have no conflict of interest.

*Acknowledgements.* EEA is acknowledged for providing all atmospheric measurement data. The EEA measurement data are based on non-aggregated time-series data downloaded from the web page https://aqportal.discomap.eea.europa.eu/index.php/users-corner/ in April 2021. Data for 2005-2012 were extracted from Airbase and 2013-2019 as validated (E1a)

from the Air Quality e-reporting database. ECMWF is acknowledged for providing ERA5 data. The meteorological data were generated using Copernicus Atmosphere Monitoring Service information 2021. Parts of the data contain modified Copernicus Atmosphere Monitoring Service information 2021. Neither the European Commission nor ECMWF is responsible for any use





that may be made of the Copernicus information or data it contains. Sabine Eckhardt at NILU is acknowledged for extracting the meteorological data from ECMWF.

*Financial support.* The authors are grateful for the European Environment Agency's financial support to the European Topic Centre on Air Pollution, Transport, Noise, and Industrial Pollution (ETC/ATNI) for this work and the national contribution provided by the Norwegian Ministry of Climate and Environment.

*Review Statements.*

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
