# Peer review of "The AirGAM 2022r1 air quality trend and prediction model"

_Geoscientific Model Development, 2022_

## Author Response (AR1)

**Reply to the referees on paper gmd-2022-70**

We wish to thank both referees for their comments on the current manuscript.

Referee #1 states that the paper "currently provides too many unnecessary (and repeated) details and is quite tedious, so it must be restructured and shortened".

We agree that the paper has become quite lengthy and contains as part of the main text too many details describing the input data, how to set up and run the model, and description of results files. Therefore we plan to move Sections 2.2-3 and 3-5 to a supplement. This will shorten the paper considerably and keep all the details of running the model separate from the main text.

This is also in line with the suggestion from Referee #2 to move Sects. 3-5 to a supplement. We will try to form this part into a more user-friendly guide to the model, keeping only the overall description of the model and the results in the main paper.

Referee #1 states that we "should place the focus on convincing the reader their code is more efficient and convenient than directly working with the package mgcv in R interface".

We believe it is quite a substantial gap from only having access to R and the mgcv library to perform the level of computations enabled by our current model script. The mgcv library is a library of many different statistical routines. It is not so easy for a user not well-versed in statistics and GAM modelling to use the right set of routines properly. Many people interested in trend modelling, we believe, may not possess the proper level of knowledge and training in statistics and GAM modelling to apply this package correctly, nor have the time or wishes to acquire such knowledge. For example, the proper computations of 95% confidence intervals for predictions performed by our model are not in the mgcv library and need a separate development, even though the details are known to statistical experts in GAM modelling. Having R and the mgcv library was our starting point when we wanted to do trend modelling with GAMs, yet it took us considerable time and effort with trial and error to develop our current model system. Thus, we believe our model code should be of interest to the geophysical community. At least, it will take a person interested in GAM trend modelling significant time and effort to achieve the same level of computations as performed by our model script using only R and the mgcv-library.

Even though our current script assumes a specific structure of input data requirements, we believe it does contain several advantages for an end user, such as automating the sequence of steps needed to perform trend analysis for whole or user-defined seasonal parts of years for a large number of stations with automatic data checking and production of results files of both unadjusted and meteorology adjusted trends with all result files automatically produced with easy identification of outcomes. It also enables the automatic creation of deterministic and probabilistic model evaluation results, cross-validation analysis, and predictions for a selectable number of years. In addition, we provide scripts enabling a user to perform the model runs in parallel on Windows and Linux in the case of a large number of stations. These are some of the benefits for an end user using our script. The model code is well commented on and can be used, with the advantage, as an advanced starting point for others to develop further, to extract code from or alter as they wish.

We can add some text in the paper's introductory sections, emphasising the benefits of our current model script.

Referee #1 also states that "For people who are new to R, this code is not straightforward enough for new users."

We believe that users not well-acquainted with R don't need to know much about R to run our model since separate batch scripts are provided to run the model on both Windows and Linux. The user only needs to provide input data, define the parameters in the options file, and run the batch scripts. Thus, using the model should require minimal knowledge of R and GAM modelling – since the model and the run scripts automate all the necessary steps.

Referee #2 states that there is a "huge potential for improving the usability and user-friendliness of the model" if "the model is provided as a R-library with usage examples".

We agree that producing an R-library with user-callable routines from the current script code is an interesting and exciting idea to further the model's scope and flexible use. We wish to look into pursuing this going forward with the code. However, this will require considerable time and effort, which we believe is clearly beyond the scope of our present paper documenting our developments and results so far.

**Answers to Referee #1 line specific comments**

**Referee comment:** 1. l62, a typo for Chang et al. (2021)

**Author's response:** Corrected.

**Author's changes:** Corrected at line 62 in the updated main paper.

===

**Referee comment:** 2. l236-l242, this paragraph is repeated, it can be removed or point the readers to additional references using spatial features in GAM.

**Author's response:** Removed the paragraph.

**Author's changes:** Removed the paragraph in the updated main paper.

===

**Referee comment:** 3. Section 2.2.1, the difference between bam & gam should be mentioned in the first place.

**Author's response:** Rephrased this section's initial sentences to highlight the differences between bam and gam.

**Author's changes:** Rephrased initial sentences at line 164 in the supplement.

===

**Referee comment:** 4. Sections 2.2-2.4 can be completely moved to supplement since these are basically the same info as Section 3.

**Author's response:** Agree; we will move these sections to the supplement.

**Author's changes:** Moved the sections to the supplement starting at line 157 there.

===

**Referee comment:** 5. l348, daily autocorrelation is expected to have a small impact on long-term trends.

**Author's response:** We have added a sentence about this at the end of the paragraph.

**Author's changes:** Added a sentence at the end of section 2.2.1 at line 185 in the supplement.

===

**Referee comment:** 6. l395, is the day of week also a cyclic function?

**Author's response:** Wind direction is the only variable which is cyclic. Day of week and day of year are not defined as cyclic variables since Sunday and Monday, and 31 Dec and 1 Jan, may differ considerably. We have added a sentence in the first paragraph of Section 2.2.4 to clarify this.

**Author's changes:** Added a sentence at line 231 in the supplement.

===

**Referee comment:** 7. l1399, it looks like the absolute/relative trend only accounting for the starting and ending points. What if the nonlinear trends changed abruptly only over the last few years, such as Fig 32? Is the relative trend estimated by starting/ending points or linear fit for the whole period?

**Author's response:** The absolute/relative trends are calculated as described in the first paragraph after Fig. 21, i.e. by just using the start and end values of the trend. We agree that such absolute/relative trend values are most representative as a trend over the whole period when the trend is linear and less so when the trend changes abruptly near the end of the period. The calculation of absolute and relative trends was done only to summarise the results since a curved function is difficult to summarise. A linearisation of the results to make them easier to present was also done in the original paper (Camalier et al., 2007), which was our work's original background.

**Author's changes:** No changes made to the manuscript.

===

**Referee comment:** 8. l1544-l1545, this is not true because chemicals have a lifetime in the atmosphere. Even if the emission is cut-off, the chemicals will not disappear all of a sudden.

**Author's response:** This sentence in Sect. 7 discusses the Random Forest (RF) model vs GAM. The primary purpose of the GAM is to find the long-term trends and to split this into a meteorologically driven trend vs all other drivers. The RF model has some advantages over GAM and vice versa, and the possibility of detecting sudden breakpoints in the data is one of these. In principle, the referee is correct that an

immediate cut in the emissions in a source region will sometimes not be reflected in a simultaneous drop in the air concentrations at a station downwind, but this is a common problem for both RF and GAM in that they will not be able to detect the emission changes in such circumstances immediately.

As shown clearly in Fig. 32, the GAM trend closely follows the RF trend, but since the GAM trend tends to be more smooth and less wiggly than an RF trend, sudden changes in the emissions are more challenging to estimate. However, applying the GAM for the Covid lockdown period in Europe (Solberg et al., 2021) did show a marked impact on the atmospheric concentrations following the significant cuts in emissions, in line with many other studies. The day-to-day perturbation was perhaps not 100% exact in time, but this slight difference did not influence our main results very much.

**Author's changes:** No changes made to the manuscript.

**Answers to Referee #2 line specific comments.**

**Referee comment:** [L. 22] 'thought to be'? Does it mean that the emissions and background concentrations (e.g., inventory data?) can be derived implicitly from the non-linear relationship between meteorology, time, and concentrations? I am curious to which extent this statement is true. The emissions data are often provided at monthly/annual intervals. This could be another reason for not including the emissions data into the model.

**Author's response:** We deliberately used the term "sought" here rather than "thought" since the latter is a little stronger, indicating that we always will succeed in bringing in the level and variations of emissions and background concentrations through meteorology and time variables in the model. However, the fact that the GAM regression models as predictor models are often quite good at predicting the concentration levels indicates that the model often succeeds in "bringing in" the actual emissions and background concentrations in the stations' surroundings. We believe this is often to a certain degree true but usually not entirely.

However, it is unfortunately not possible to unentangle the emission and background concentration parts from the constant term and non-linear relations estimated by the model. E.g. the constant term in the regression is related to the average concentration level throughout the estimation period and thus to the combined effect of average levels of emission and background concentrations. Also, the time variables day of week and day of year are related to the weekly and annual variations in emissions and background concentrations, but again as a combined effect of both.

**Author's changes:** No changes made to the manuscript.

===

**Referee comment:** [63] Typo. The citation might be 'Chang et al. (2021)'

**Author's response:** Corrected.

**Author's changes:** Corrected at line 62 in updated main paper.

===

**Referee comment:** [L.116]" these are thought"?

**Author's response:** See answer to [L. 22].

**Author's changes:** No changes made to the manuscript.

===

**Referee comment:** [L. 211, Table 1] Does UT mean "universal time"? As the AirBase/AQER stations are scattered across several time zones, I would doubt whether it is appropriate to use variables at 18 UT. The atmospheric stability may differ substantially for stations located in different time zones.

From the context, it is not clear which ERA5 reanalysis data were used in the model. But based on the references, it is the "ERA5 hourly data on single levels from 1979 to present" that were used. As far as I know, ERA5 hourly data on single levels do not provide humidity variables directly. If the humidity variables are derived using 2m air temperature, 2m dewpoint and surface pressure, please describe the formulae used. 10m wind direction is not directly given in the dataset neither (at least not for land areas).

**Author's response:** The measurement data that this work is based on covers three time zones: CET-1h, CET, and CET+1h, CET = Central European Time = UTC + 1. We believe the 1h difference compared to the met data from ECMWF that are given in UTC is negligible and that the effect of this on the results is well within all other uncertainties.

We are not entirely sure what type of data set description the referee asks for. We refer to the Acknowledgement, stating: "The meteorological data were generated using Copernicus Atmosphere Monitoring Service information 2021."

The referee is correct that wind direction as such is not provided in the ECMWF data, but the horizontal wind components (u and v at 10 m) are provided, and we used these data and the R routine windDirSpd to compute the direction.

The referee is also right that relative humidity is not given in the ECMWF data, but absolute humidity is. We used the data for absolute humidity, surface temperature, surface pressure and height of the monitoring station to calculate the relative humidity based on formulas given in Vaisala (2013) (this report can be obtained from us if needed).

In detail:

The saturation pressure of water vapour in the air:

$P_{ws} = A*10^{[m*T/(T+T_n)]}$

where

$A = 6.116441$

$m = 7.591386$

$T_n = 240.7263$

The partial pressure of water vapour in g/kg is calculated as:

Pw = h2o*Ps/(h2o+B)

where

h2o = q*1000. where q = the absolute humidity given in the ECMWF data

Ps = estimated atmospheric pressure at the height of the station:

PS = 0.01*(MSL-1.2*9.81*Height of station)

where MSL = mean sea level pressure and

B = 621.9907

Then the relative humidity in per cent, RH, is calculated by:

RH = 100.*Pw/Pws

According to Vaisala (2013), these formulas could be used for a temperature range of [-20 C, +50C]

On a few occasions, this calculation leads to RH higher than 100 % or lower than 0. We forced RH to be 100% and 0 in these cases.

We will add these formulae used in pre-processing the meteorological input data to the manuscript.

**Author's changes:** Have added Appendix C to the supplement describing how we obtain wind direction and relative humidity from the ECMWF ERA5 data. Also added a paragraph about this at line 229 in the updated main paper.

===

**Referee comment:** [Table 1] Are the cyclic variables (wind direction, dayofyear) transformed in the model? For example, day 1 and day 365 diverge remarkably in value but are very close in time.

**Author's response:** None of the covariates is transformed in the model, only the concentrations for NO2 and PM. Wind direction is the only variable which is cyclic. Day of week and day of year are not defined as cyclic variables since Sunday and Monday, and 31 Dec and 1 Jan, may differ considerably.

**Author's changes:** Added a paragraph at line 225 in the updated main paper.

===

**Referee comment:** [L. 271 - 272] it is not clear for me what the sentence "Since ….A= Y_bar" means.

**Author's response:** This relates to Eq. (3) in the paper. Suppose we set the average of the y_trend values to the average of Y values. Then since B(t) averages to zero over the time points, which is a property of GAM modelling, the average of the y_trend values will be equal to A, i.e. A will be equal to the average of the Y values. We will paraphrase this sentence in the paper using the above text, hopefully clarifying what we mean.

**Author's changes:** Added some text to this sentence at line 275 in the updated main paper.

===

**Referee comment:** [Section 2.2.2] If the select can be set as "TRUE", incorporating background emissions into the model may not be a problem. If it does not bring benefits to add emission data, GAM can detect and delete it.

**Author's response:** Yes, background emissions might be introduced in the model, and if they do not provide any explanatory power to the regression, the automatic model selection turned on by the select=TRUE will remove them, i.e. make them into zero functions.

If such background emissions data do not contain any trends, then the trend term in AirGAM will still estimate trends in local and background emissions affecting the concentrations. If there are trends in the background emissions, then the trend estimated by AirGAM will be an estimate of the remaining trend in local and background emissions after the model has accounted for these trends in the input background emissions.

However, we prefer to run AirGAM without any emissions since it is often difficult to obtain relevant background emissions over long periods and large spatial domains, such as in our present European study from 2005-2019. Not including emissions in the GAM model also makes it much simpler to interpret the trend as the trend in emissions affecting the stations since we do not need to consider eventual trends in the input emission data.

Also, regarding incorporating background emissions, the monitoring stations are affected by emissions from a large area surrounding the site, depending from hour to hour on wind speed, wind direction and atmospheric mixing. Incorporating such data into the GAM might result in something more akin to a standard CTM (Chemical transport model) based on gridded and time-varying emissions, which was not the intention and purpose of our GAM.

**Author's changes:** No changes made to the manuscript.

===

**Referee comment:** [L. 399] Please justify the usage of sandwich library and vcovHAC routine.

**Author's response:** There are three reasons for using these for the simple linear regression model to obtain the beta_linreg coefficient. First, since a linear regression model is not an exact model, in this case, we prefer to use the sandwich construction $J^{-1}*K*J^{-1}$ for the variance matrix for the linear regression coefficients rather than the simpler but more inaccurate $J^{-1}$ (J-inverse). Secondly, in this case, the regression uses time-series data (concentrations) that are auto-correlated. This is considered by the AC part of the vcovHAC routine. Thirdly, the vcovHAC routine also considers heteroscedasticity in the time series, i.e., variances vary with the level of the series (the concentrations). The benefit is that we get a better estimate of the variance of the estimated beta_linreg parameter and thus a better estimate of the p-value associated with this parameter.

**Author's changes:** Added a paragraph about this at line 239 in the supplement.

===

**Referee comment:** [Section 3] For better readability I would suggest that the description on technical details be put into the supplement (in the form of a manual). If the authors would move a step further to

demonstrate the usage of the AirGAM model and introduce various functions offered by the package, a tutorial made by Rmarkdown would be helpful.

**Author's response:** We agree and will move Section 3 into the supplement. If time permits, we will try to make a short tutorial in Rmarkdown to demonstrate the model.

**Author's changes:** Moved Section 3 to the supplement as Section 3 there.

===

**Referee comment:** [L.543] According to the definition of winter and summer made here, the entire year is divided into two parts – winter (oct-mar) and summer (apr-sep). This contradicts Lines 567-568 where each season comprises three months. Consider changing to warm-cool, hot-cold?

**Author's response:** Perhaps we could use DJF, MAM, JJA and SON for the four seasons?

**Author's changes:** Changed this at line 411 in the supplement.

===

**Referee comment:** [L. 611] "The default setting of this variable, when the trend_type is nonlinear, is NA, …"

**Author's response:** Default here means this is the typical variable setting in the input file. It will also be set to NA, i.e. missing value, if the variable is omitted from the input file.

**Author's changes:** No changes made to the manuscript.

===

**Referee comment:** [Section 4-5] These two sections could be also put into the supplement.

**Author's response:** We agree and will move these two sections to the supplement.

**Author's changes:** Moved these sections to the supplement as Sections 4 and 5 there also.

===

**Referee comment:** [L. 961] "blue curve". Please try to give quantitative measures (e.g., MSE, R2, Nash-Sutcliffe Efficiency) to support the statement of good correspondence between modelled and observe values.

**Author's response:** We have added RMSE and R2 in a sentence following the figure.

**Author's changes:** Corrected figure text to "blue curve" at line 860 and added these numbers at line 861 in the updated main paper.

===

**Referee comment:** [Section 5.3] The section on the probabilistic model evaluation needs to be expanded to include more details on the performance measures adopted. It is not intuitive how these metrics are defined, even the references are given.

**Author's response:** We will do this.

**Author's changes:** Added definition of PIT values at line 1036 in the supplement and definition of CRPS values at line 1116 in the supplement.

===

**Referee comment:** [L. 1384] Can you add an inlet figure (e.g., percentage change vs. longitude) to support the west-east decreasing gradient?

**Author's response:** We think this is perhaps unnecessary since the gradient is very weak, as we indicate in the text. Such a figure will also be tiny and difficult to read. We might decide to remove the text on this since the gradient is indeed very weak.

**Author's changes:** No changes made to the manuscript.

===

**Referee comment:** [L. 1464-1465] It's not clear why the trend analysis was not conducted for stations in France. Are the French data provided at hourly or daily interval? If the French data are hourly, they could be easily aggregated to daily to perform the trend analysis.

**Author's response:** The reason for the missing trends for the French PM data was a significant shift from daily to hourly sampling in that country from 2010. We decided not to aggregate hourly values into daily data since it was not recommended to combine these two types of data from France (cf. Augustin Collette). The reason was that a specific scaling factor should be used for the hourly data sets when considering the annual mean. So, the annual means calculated this way could be combined (from hourly and daily samplers), but not the daily means we used in our work.

**Author's changes:** No changes made to the manuscript.

===

**Referee comment:** [L. 1552] Why not using the same dataset with the same temporal coverage to compare results from the two approaches?

**Author's response:** The dataset from the study by Grange and Carslaw (2019) covers the period 1997-2016, while ours covers the period 2005-2019. Thus, we could have chosen the common period 2005-2016. This common period is part of our current study period 2005-2019. We managed to reproduce the results from Grange and Carslaw in our analysis using the RF method for 2005-2016. Thus, we thought it would be interesting for the reader to present and compare RF and GAM for our whole study period, 2005-2019, since this includes a steep trend at the end.

**Author's changes:** No changes made to the manuscript.

===

**Referee comment:** [L.1859] typo, 'PM2.5'

**Author's response:** Corrected.

**Author's changes:** Corrected this at line 746 in the updated main paper.

===

**Referee comment:** [Appendix A] I do not think the details on how to install R would be necessary. As far as I can see from the Zenodo repository, the AirGAM model has not been wrapped into an independent library, but it is run like a normal script in R/Rstudio. For better user friendliness, it would be nice if the model is provided as a R-library with usage examples.

**Author's response:** We can remove the descriptions of the R installation and focus on the specifics related to AirGAM and its libraries.

Making an R-library of AirGAM with user examples is an exciting proposition. We also see that a user may need it as a tool to be able to run parts of it in an interactive R-session. Our focus in the current development was to make it into a model script with precise input data requirements and a set of optional parameters to perform the calculations and supply the user with run scripts to be able to run it for a (large) number of stations.

Making it into an R-library with callable routines performing parts of what AirGAM now does is doable, although it will take considerable time and effort to do this. Thus, we believe it is outside the scope of the current work. However, we will consider this going forward with the model development.

**Author's changes:** Moved Appendices A and B to the supplement. Removed the details of R installation. Also made a new Section 2.3 at line 330 in the updated main paper describing the content of the supplement with Appendices A and B. Also described pros and cons of having the model as an R script vs. an R library in the new Section 2.3 in the updated main paper.

=== END ===

---

## Author Response (AR2)

**Answers to Referee #1 comments submitted 20 Oct 2022**

**Referee comment:**

I found the final step to summarize the trends is somewhat disappointing and discouraging ("The absolute/relative trends are calculated as described in the first paragraph after Fig. 21, i.e. by just using the start and end values of the trend"). All those careful considerations and model settings regarding the nonlinearity, emissions, and meteorologies, but at the final step ONLY the start and end points (1 Jan 2005 and 31 Dec 2019) are used to summarize the trends. This is essentially implying no matter what sort of nonlinearities in the data, as long as the (fitted) start and end points are identical. they will have the same summarized trend values, regardless of the autocorrelation, data variability, sample size, etc... and all those factors deem to be important for trend detection.

"We agree that such absolute/relative trend values are most representative as a trend over the whole period when the trend is linear and less so when the trend changes abruptly near the end of the period."

This should not be a justification since the whole point of this paper is that GAM is adopted for handling the nonlinearity. As demonstrated in their results, the air quality analysts often need to deal with time series data from hundreds or thousands of monitoring stations at once. Under this circumstance it is not possible to inspect each time series in detail separately, the same model formulation will be specified for all stations and in the end merely the final linear trends will be reported (e.g., Fig 2 in the paper). How can the practitioners make sure the results are appropriate? Especially no standard errors associated with the relative trends are reported.

**Author's response:** In Section 3 of the main paper, we calculate and report the *relative change in the expected concentration level at each station from 2005 to 2019.* We do this by using the start and end values of the meteorology-adjusted non-linear trend curve over this period since these values represent the expected concentration levels at the start and end of the period. Section 2.2 in the main paper describes the precise definition of these expected concentrations in connection with the calculation of the physically interpretable trend curves. We acknowledges that the way we wrote this could easily be misunderstood to mean that we were reporting the overall trends themselves. We have therefore corrected the text in Section 3 to make this clearer.

The trends, as such, are non-linear curves over the period. These curves are shown for each station in the plot files

`<sname>_gam.trend_metadj_2005_2009.png`

where `<sname>` denotes the station name, and the trend curve values are given in the text files

`<sname>_gam.trend_metadj_2005_2009.txt.`

In addition, 2.5 and 97.5 percentiles are given in this latter file for each trend curve value, forming a 95% confidence interval for this value. Thus, they describe the uncertainty associated with the estimated trend pointwise through the period.

All these non-linear trend results can be found in the data repositories accompanying the paper. However, it isn't easy to summarize their non-linear nature concisely. This is why we decided on only reporting the overall absolute and relative changes in the expected concentrations based on the non-linear trend curve at each station over the period.

It is possible to use the 95% confidence intervals for the start and end trend values to define an approximate 95% confidence interval for the corresponding absolute or relative changes in the expected concentrations at each station. However, due to the very large number of stations, we decided to leave this out since it would have required almost doubling the number of plots.

Note that, based on the output from AirGAM it is perfectly possible to report several absolute and relative changes in expected concentrations over parts of the period, e.g., from say 2010 to 2014, or 2015 to 2019, etc. Such results over parts of the period can be easily calculated based on the output of the data for the non-linear trend curves. This would not have been possible if we only used regression with a linear trend.

That we use only the start and end of the trend curve is done just to simplify the presentation of the total change over the time period for the reader. If the non-linear trend estimated by AirGAM is undulating there is no simple way to summarize that quantitatively. One solution could be to estimate the total change over the period as done now and possibly combine that with a qualitative description of the trend. We admit that this could be a challenging task when studying a large number of stations, but we see this as a challenge for the data analyst in the post-processing due to the complexity of the trends themselves and should not be seen as a critic against the GAM method itself.

**Author's changes:** Rewritten parts of Sections 3.1.1 and 3.2-3.4 in the main paper to clarify that we are reporting the absolute and relative changes in expected concentrations based on the meteorology-adjusted non-linear trend at each station from 2005 to 2019. The changes are marked in red.

**Referee comment:** l.163 is a "linear" regression model linking... to one or more explanatory "nonlinear" variables.

**Author's response:** No, our GAM models are non-linear regression models. As explained above, we calculate the *absolute and relative change in the expected concentration level at each station from 2005 to 2019* by using the start and end points of the meteorology-adjusted non-linear trend. It does not mean that we assume in any way that the trend in itself is linear over the period. Thus, the absolute or relative change in expected concentration is not the result of performing regression assuming the trend to be linear.

**Author's changes:** No corrections made.